# Brain–phenotype models fail for individuals who defy sample stereotypes

Abigail S. Greene[1,2 ✉], Xilin Shen[3], Stephanie Noble[3], Corey Horien[1,2], C. Alice Hahn[3], Jagriti Arora[3], Fuyuze Tokoglu[3], Marisa N. Spann[4], Carmen I. Carrión[5], Daniel S. Barron[6,7,8,9], Gerard Sanacora[7], Vinod H. Srihari[7], Scott W. Woods[7], Dustin Scheinost[1,3,10,11,12] & R. Todd Constable[1,3,10,13 ✉]

Individual differences in brain functional organization track a range of traits, symptoms and behaviours[1–12]. So far, work modelling linear brain–phenotype relationships has assumed that a single such relationship generalizes across all individuals, but models do not work equally well in all participants[13,14]. A better understanding of in whom models fail and why is crucial to revealing robust, useful and unbiased brain–phenotype relationships. To this end, here we related brain activity to phenotype using predictive models—trained and tested on independent data to ensure generalizability[15]—and examined model failure. We applied this data-driven approach to a range of neurocognitive measures in a new, clinically and demographically heterogeneous dataset, with the results replicated in two independent, publicly available datasets[16,17]. Across all three datasets, we find that models reflect not unitary cognitive constructs, but rather neurocognitive scores intertwined with sociodemographic and clinical covariates; that is, models reflect stereotypical profiles, and fail when applied to individuals who defy them. Model failure is reliable, phenotype specific and generalizable across datasets. Together, these results highlight the pitfalls of a one-size-fits-all modelling approach and the effect of biased phenotypic measures[18–20] on the interpretation and utility of resulting brain–phenotype models. We present a framework to address these issues so that such models may reveal the neural circuits that underlie specific phenotypes and ultimately identify individualized neural targets for clinical intervention.

Relating individual differences in brain activity to complex phenotypes is a long-standing aim of human neuroscience, which has been advanced by the application of machine learning algorithms to neuroimaging and phenotypic data. Such work has revealed patterns of brain activity that are associated with a range of traits[1,2,5], behaviours[6–9], psychopathology[10–12], clinical risk factors[3] and treatment outcomes[4] across operationalizations, datasets, age groups and diagnoses. Together, this reflects a paradigm shift in human neuroscience research from a focus on the group to a focus on the individual, with important potential applications to clinical practice[21–23].

To deliver on this promise, however, these approaches must identify patterns of brain activity that are relevant to the phenotype of interest in a given individual—the patient sitting before their clinician, for example. Previous linear modelling work has relied on the assumptions that (1) a single brain network is associated with a given phenotype, with patterns of activity within that network varying across individuals[10,24,25]; and (2) larger, more heterogeneous samples will more accurately and reliably capture this single model[26,27]. But although many published models have demonstrated impressive generalizability[6,9,10], they do not account for brain–phenotype relationships in all individuals[13,14]. This raises the crucial question of in whom models fail, and why.

The existence of structured model failure—some individuals who are better fit by a model than others[14,24,26]—would suggest that one brain–phenotype relationship does not fit all, and that systematic bias may determine who is fit and who is not. This, in turn, may engender imprecise, misleading and in some cases harmful model interpretations. That is, a brain network that is found to be associated with a given phenotype may only apply to a specific subset of the population at large, limiting its practical utility[14,26,28], or may not

[1]Interdepartmental Neuroscience Program, Yale School of Medicine, New Haven, CT, USA. [2]MD–PhD program, Yale School of Medicine, New Haven, CT, USA. [3]Depatment of Radiology and Biomedical Imaging, Yale School of Medicine, New Haven, CT, USA. [4]Department of Psychiatry, Columbia University Irving Medical Center, New York, NY, USA. [5]Department of Neurology, Yale School of Medicine, New Haven, CT, USA. [6]Department of Anesthesiology and Pain Medicine, University of Washington, Seattle, WA, USA. [7]Department of Psychiatry, Yale School of Medicine, New Haven, CT, USA. [8]Department of Psychiatry, Brigham and Women's Hospital, Harvard Medical School, Boston, MA, USA. [9]Department of Anesthesiology, Perioperative and Pain Medicine, Brigham and Women's Hospital, Harvard Medical School, Boston, MA, USA. [10]Department of Biomedical Engineering, Yale School of Engineering and Applied Science, New Haven, CT, USA. [11]Department of Statistics and Data Science, Yale University, New Haven, CT, USA. [12]Child Study Center, Yale School of Medicine, New Haven, CT, USA. [13]Department of Neurosurgery, Yale School of Medicine, New Haven, CT, USA. ✉e-mail: abigail.greene@yale.edu; todd.constable@yale.edu

represent the phenotype of interest. Indeed, factors that interfere with adequate phenotypic characterization have been documented for many widely used neurocognitive tests[18,29], and may include the fallacy of universalism (construct bias), the application of inappropriate norms, discordance between primary and assessment language and the presence of instrument, administration (method) or interpretation bias[18–20,30,31]. Related concerns about the ethical implications of data and model bias are receiving increasing attention in the machine learning literature[32] (for example, racial disparities[33] or unrelated attribute sensitivity[34] in facial recognition, and the reflection of biased input data in algorithmic predictions, from criminal justice[35,36] to healthcare[37,38]). However, whether brain–phenotype models are affected by bias in phenotype measurement and, if so, how this bias governs model failure remain open questions. Answering them is a prerequisite for discovering precise and useful brain–phenotype relationships.

To do so, we trained models to use brain activity to classify neurocognitive test performance and investigated the failure of these models. Across a range of data-processing and analytical approaches applied to three independent datasets, we found that model failure is systematic, reliable, phenotype specific and generalizable across datasets, and that the scores of individuals are poorly classified when they 'surprise' the model, performing in a way that is inconsistent with the consensus covariate profile of high and low scorers. Together, these findings suggest that brain-based models often represent not unitary neurocognitive constructs, but rather constructs of interest intertwined with clinical and sociodemographic factors. These factors comprise a stereotypical profile that does not fit all individuals in the study sample, and may generalize worse still to the population at large[39]. Models that predict this profile will fail in those who defy it. Model failure is thus informative, both because it identifies subtypes that require distinct predictive models, and because it offers insight into data and model biases that should guide model interpretation.

## FC predicts phenotype but models frequently fail

To examine in whom models fail, we first developed a pipeline to train and test models, using functional connectivity (FC) to predict performance on tests that represent a range of cognitive domains. Primary results were derived from a new dataset collected at Yale (Supplementary Tables 1–3 and 'Datasets' in Methods) using four distinct prediction algorithms (Fig. 1a, Extended Data Fig. 1, Supplementary Table 11 and 'Phenotype classification' in Methods), and validated in two independent, publicly available datasets (UCLA Consortium for Neuropsychiatric Phenomics[17] and the Human Connectome Project (HCP)[16]; Supplementary Table 3).

FC from the best-performing condition for each measure classified scores on 14 out of 16 phenotypic measures with above-chance performance (mean accuracy = 0.51–0.88; Fig. 1b). Performance varied across in-scanner conditions that were used to calculate FC (mean accuracy across iterations and measures, using only FC calculated from the best-performing condition = 0.68; mean accuracy across all conditions, iterations, and measures = 0.60; Fig. 1b and Supplementary Table 4). FC also significantly classified phenotype in the UCLA data (Extended Data Fig. 2a and Supplementary Table 5) and HCP data (with family members assigned to the same fold, and permutations respecting family-related limits on exchangeability[40,41]; Extended Data Fig. 3a and Supplementary Table 6). However, although the classification accuracy of most phenotypic measures was significantly better than chance, many participants were misclassified (for example, 12–59% of participants across all conditions and measures in the Yale dataset; see Supplementary Tables 4–6 for model performance in each dataset). We turn next to an investigation of structure in these model failures.

## Model failure is consistent and phenotype specific

First, we examined whether misclassification demonstrates non-random structure. We reasoned that if misclassification were random, misclassification frequency (that is, the fraction of iterations on which a given participant was misclassified; see 'Internal validation analyses' in Methods) would be approximately normally distributed around a mean of 0.5. Indeed, when phenotypic labels were permuted, the mean misclassification frequency did not differ from 0.5 in 14 out of 16 cases ($P > 0.05$, FDR adjusted, by two-tailed, one-sample $t$-test for all measures except symbol search (mean = 0.505, $P_{FDR}$ = 0.02) and letter–number (mean = 0.507, $P_{FDR}$ = 0.0004)). Conversely, misclassification frequency using original (unpermuted) data has a markedly different, U-shaped distribution; for most measures, most participants were consistently correctly classified, whereas a smaller subset of participants was consistently misclassified across iterations (Fig. 2a). For every measure, the mean and median of the distribution significantly differed from 0.5 (all $P < 0.05$, FDR adjusted (32 tests), by two-tailed $t$-test and Wilcoxon signed-rank test), and the distributions of misclassification frequency for original and permuted analyses significantly differed for every measure (all $P < 0.0001$, FDR adjusted, by two-tailed, two-sample Kolmogorov–Smirnov test).

Next, we tested the consistency of misclassification across in-scanner conditions. For all neurocognitive measures except cancellation, correlation of misclassification frequency across conditions for a given measure was significantly greater than chance (that is, than correlation of misclassification frequency derived from permuted-label analyses, by paired, one-tailed Wilcoxon signed-rank test; all $P < 0.0001$, FDR adjusted; Fig. 2b), suggesting that the tendency to be misclassified is consistent regardless of the in-scanner condition during which fMRI data were acquired.

Finally, we tested the phenotype specificity of misclassification—that is, whether similar phenotypic measures yield similar sets of misclassified participants. Indeed, the more similar the phenotypic measure scores, the more similar the misclassification frequencies of participants for those measures (measure versus misclassification frequency correlation: $r_s = 0.49$, $P < 0.0001$; Fig. 2c). Hierarchical linkage demonstrates which measures have a similar misclassification frequency, revealing a logical organization by cognitive construct (Fig. 2d).

These results were replicated in the UCLA and HCP datasets (Extended Data Figs. 2 and 3). In the HCP sample, misclassification frequency did not significantly differ for individuals with and without siblings in the sample (crystallized intelligence (cIQ) $P = 0.77$, fluid intelligence (fIQ) $P = 0.36$, uncorrected, by Mann–Whitney $U$ test). Misclassification frequency and overall classification performance were comparable with additional motion controls (Supplementary Fig. 1 and Supplementary Table 7), as well as with different supervised learning algorithms, brain parcellation, cross-validation approach and phenotype binarization threshold (Extended Data Fig. 1). Together, these results show that model failure is reliable and phenotype specific.

## Models and their failure generalize across datasets

For misclassification to be a meaningful organizing principle and not simply the product of relatively small (Yale and UCLA) or idiosyncratic samples, it must generalize across datasets. To ensure that this is the case, we trained a model to use GFC[42] to classify performance on three neurocognitive measures that are included in both the Yale and UCLA datasets (letter–number, vocabulary and matrix reasoning). Models were trained on all participants, only on within-sample CCP or only on within-sample MCP, and all analyses were performed twice, once with each dataset as the training set and all participants in the other dataset as the test set ('Cross-dataset analysis' in Methods).

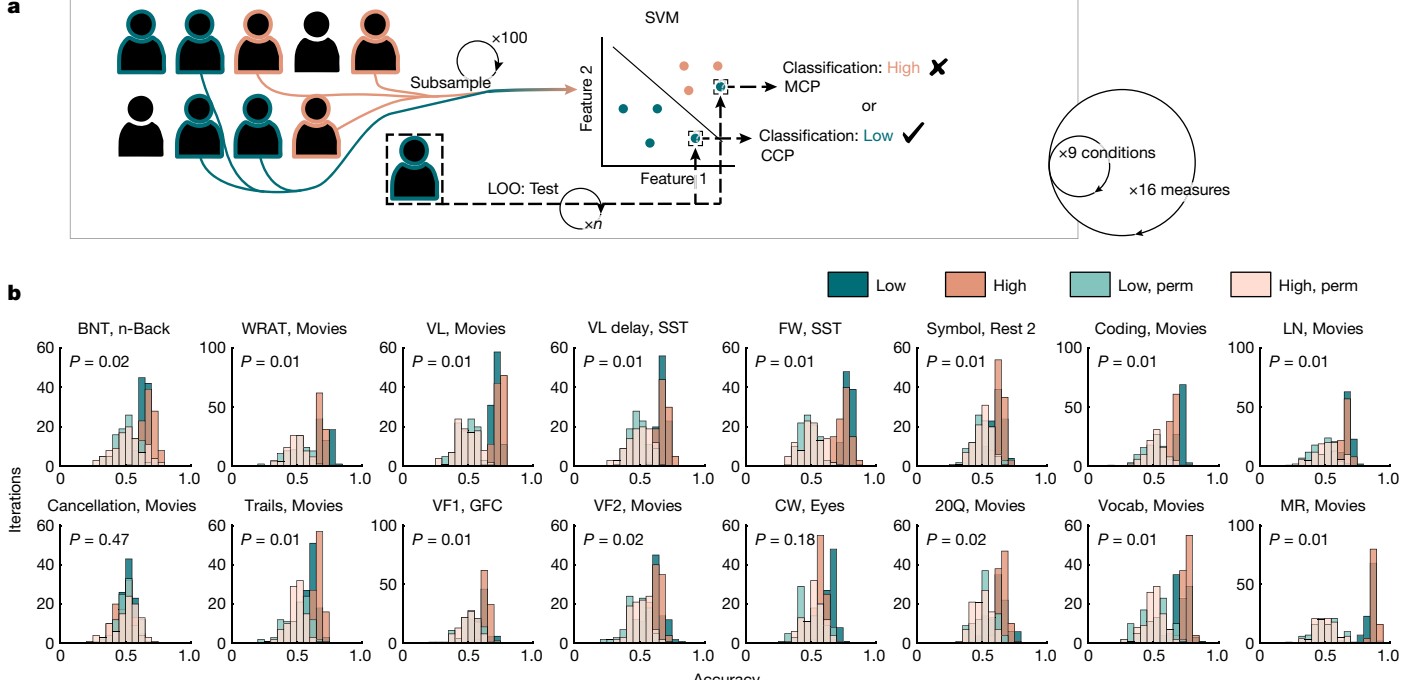

**Fig. 1 | FC can be used to classify scores on a range of neurocognitive measures. a**, Schematic illustration of the main classification pipeline. Classification was performed using leave-one-out (LOO) cross-validation. The training set was subsampled from the remaining participants to balance classes and was submitted to a linear support vector machine (SVM), using summed FC of selected edges as features. This trained model was then applied to the left-out test participant to classify their score as high or low from their FC. Participants who were successfully classified are termed 'correctly classified participants' (CCP), and participants who were misclassified are termed 'misclassified participants' (MCP). This procedure was repeated iteratively, with each participant used as the test participant, and this, in turn, was repeated 100 times with different training set subsamples selected on each iteration. This pipeline was repeated for each in-scanner condition and neurocognitive measure (numbers correspond to Yale study; comparable approach for UCLA and HCP). To ensure that the results are robust to these choices, analyses were repeated with alternative algorithms (bagging[72] and neural networks); with 10-fold cross-validation; with an alternative parcellation of functional magnetic resonance imaging (fMRI) data; with an alternative threshold for score binarization; and with continuous phenotypic measures. See Methods, Extended Data Fig. 1 and Supplementary Table 11 for comparable results. **b**, Classification accuracy for each phenotypic measure, shown separately for high and low scorers and compared to the distribution of accuracy from 100 iterations of permutation tests ('perm'). Significance was determined using the fraction of iterations on which the null classifier performed as well as or better than the median accuracy of unpermuted classifiers (across the whole sample) and resulting one-tailed *P* values were adjusted for multiple comparisons using the false discovery rate (FDR; 16 tests). Distributions and significance testing reflect accuracy across iterations for the best-performing in-scanner conditions, each noted in the plot title. For abbreviations and more on tasks and phenotypic measures, see Supplementary Tables 1 and 2. For sample sizes, see Supplementary Table 4.

First, we found that models generalize across dataset. That is, whole-sample-based models classified neurocognitive scores in the whole test sample with above-chance accuracy, as did models that were trained only on CCP (*P* < 0.0001, FDR adjusted, by nested ANOVA; Fig. 3a, 'Train: All and Train: Correct; Test: All').

Next, across all measures and both datasets, we found that classification outcome (correct versus misclassified) generalizes across dataset. CCP in one dataset were classified with above-chance accuracy by a model trained on all participants or only on CCP in the other dataset (both *P* < 0.0001, FDR adjusted, by nested ANOVA (see Supplementary Table 8 for a note on multiple comparison adjustment); Fig. 3a, 'Train: All and Train: Correct; Test: Correct'). Conversely, a model trained on MCP performed with below-chance accuracy on the other dataset's CCP (*P* < 0.0001, FDR adjusted, by nested ANOVA; Fig. 3a, 'Train: Misclassified; Test: Correct'). Similarly, MCP in one dataset were classified with above-chance accuracy only by a model trained on the other dataset's MCP (*P* < 0.0001, FDR adjusted, by nested ANOVA), but were classified with below-chance accuracy by models that were trained on the whole training dataset or only on its CCP (both *P* < 0.0001, FDR adjusted, by nested ANOVA; Fig. 3a, 'Test: Misclassified'). Notably, building group-specific models of phenotypic score (that is, training a model on CCP and applying it to CCP, or training a model on MCP

and applying it to MCP, rather than training and testing on the whole datasets) improved classification accuracy (CCP versus whole and MCP versus whole; both *P* < 0.0001 by paired, one-tailed Wilcoxon signed-rank test).

In sum, CCP-based models work on another dataset's CCP, and MCP-based models work on another dataset's MCP, but CCP-based models in one dataset do not work for MCP in the other, and vice versa. All results were replicated in analogous analyses using the HCP dataset (Extended Data Fig. 3d), and individual models tested for significance using permutation tests yield comparable trends (Supplementary Table 8). Together, these results demonstrate that what it means to be misclassified is consistent across datasets.

## Comparison across measures, datasets and groups

The significantly below-chance performance when a model was trained on CCP and tested on MCP and vice versa motivated us to further investigate model similarity across groups. First, we trained the model on CCP, as described previously, switched positively and negatively correlated edge sums in the test set and calculated the classification accuracy in test-set MCP, as before, using this 'inverted model'. This was then repeated with MCP as the training data and the

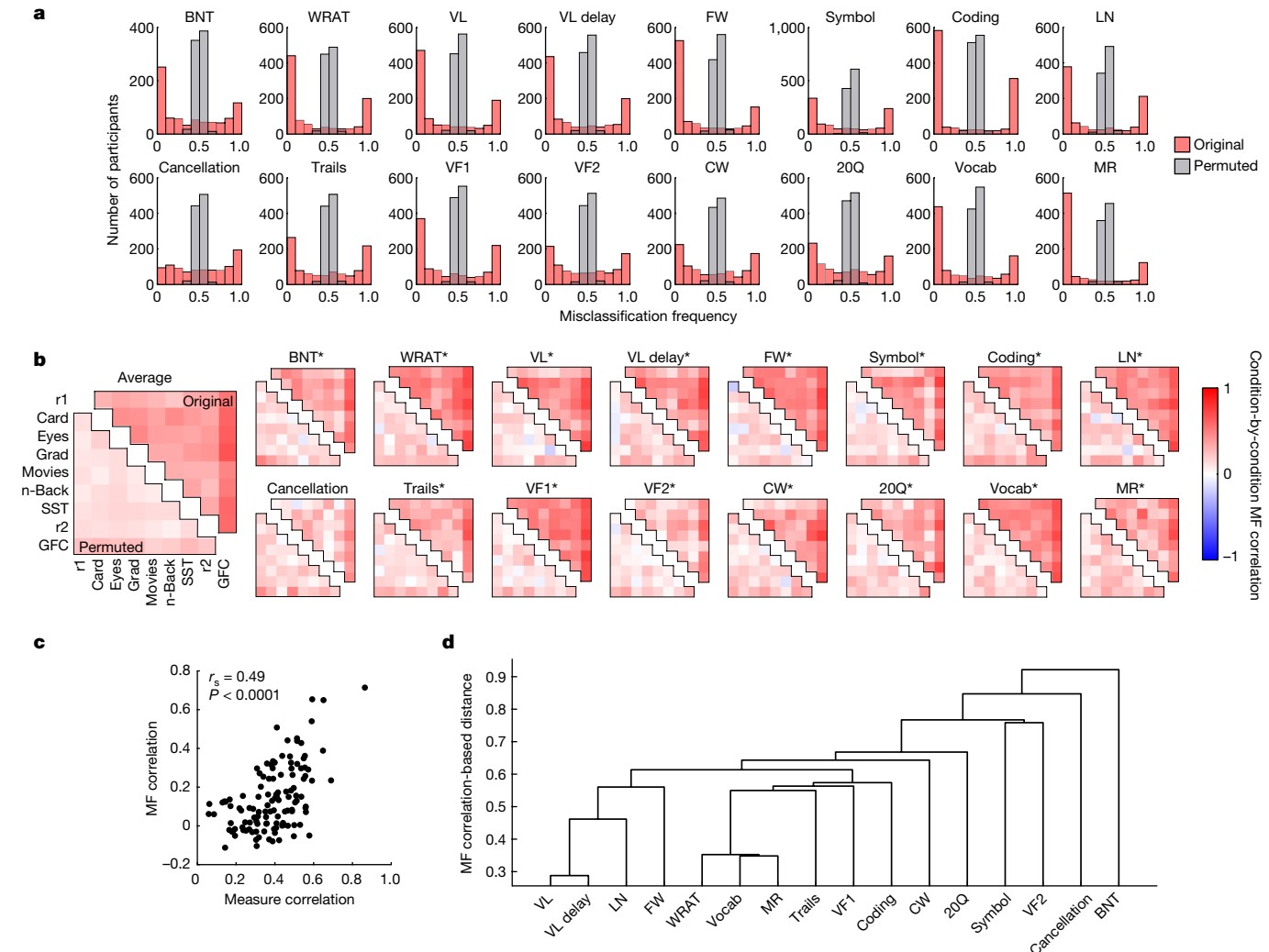

**Fig. 2 | Misclassification is consistent and phenotype specific. a**, Histogram of misclassification frequency for each phenotypic measure. Each histogram represents misclassification frequency (MF) for each participant, concatenated across in-scanner conditions and presented for analyses using original (that is, unpermuted) data (red) and permuted data (grey). **b**, Condition-by-condition correlation of misclassification frequency for analyses using original (top triangle) and permuted (bottom triangle) data, presented for each phenotypic measure. Condition order for individual phenotypic measures as in 'Average'. *Significantly different from permuted result correlations, by paired, one-tailed Wilcoxon signed-rank test; all

$P < 0.0001$, FDR adjusted (16 tests). r1, rest 1; r2, rest 2; GFC, general FC[42]; grad, gradCPT[73]. **c**, Relationship between phenotypic measure similarity (Spearman correlation) and misclassification frequency similarity (Spearman correlation). Each point represents a measure pair (given different participants excluded for intermediate, missing or outlier scores for each measure; number of correlated participants for each measure pair ranges from misclassification frequency: 63 to 114, measure: 105 to 129). **d**, Alternative visualization of misclassification frequency similarity, using a hierarchical linkage tree to reveal that measures that tap into similar constructs yield similar sets of misclassified participants.

inverted-model accuracy calculated in test-set CCP. In both Yale–UCLA and Yale–HCP analyses, the mean classification accuracy was significantly above chance in both cases (all $P < 0.0001$, FDR adjusted (18 tests), by nested ANOVA; Yale–UCLA CCP model applied to inverted MCP: overall mean (s.d.) = 0.64 (0.06); MCP model applied to inverted CCP = 0.64 (0.09); Yale–HCP CCP model applied to inverted MCP = 0.76 (0.05); MCP model applied to inverted CCP = 0.76 (0.05)). Together, these results show that edges that are positively correlated with phenotype in CCP are negatively correlated with phenotype in MCP, and vice versa.

The success of these inverted models suggested that models trained in CCP and MCP would overlap substantially for a given phenotype, albeit with opposite relationships between FC and phenotype. We also expected similarity of a given model across phenotypic measures and datasets. To test these predictions, we calculated the similarity of every cross-dataset model pair as 1 − Jaccard distance

('Cross-dataset analysis' in Methods) and visualized the resulting model-by-model similarity matrix, thresholded for significance at $P < 0.05$, by the hypergeometric cumulative distribution function (Fig. 3b). As expected, for a given phenotypic measure, significant similarity is primarily observed across datasets and measures for the same model type (for example, edges positively correlated with phenotype in CCP-based models; 'PC' in Fig. 3b), and opposite-sign edges in the other classification outcome group (for example, edges negatively correlated with phenotype in Yale CCP-based models and positively correlated with phenotype in Yale MCP-based models; UNC and UPM in Fig. 3b).

Model similarity across datasets as well as opposite relationships between FC and phenotype in correct and misclassified groups are also apparent when the selected edges incident to each model's highest-degree node are visualized (Fig. 3c and Extended Data Fig. 4). For example, node 166 is the highest-degree node in both the UCLA-test

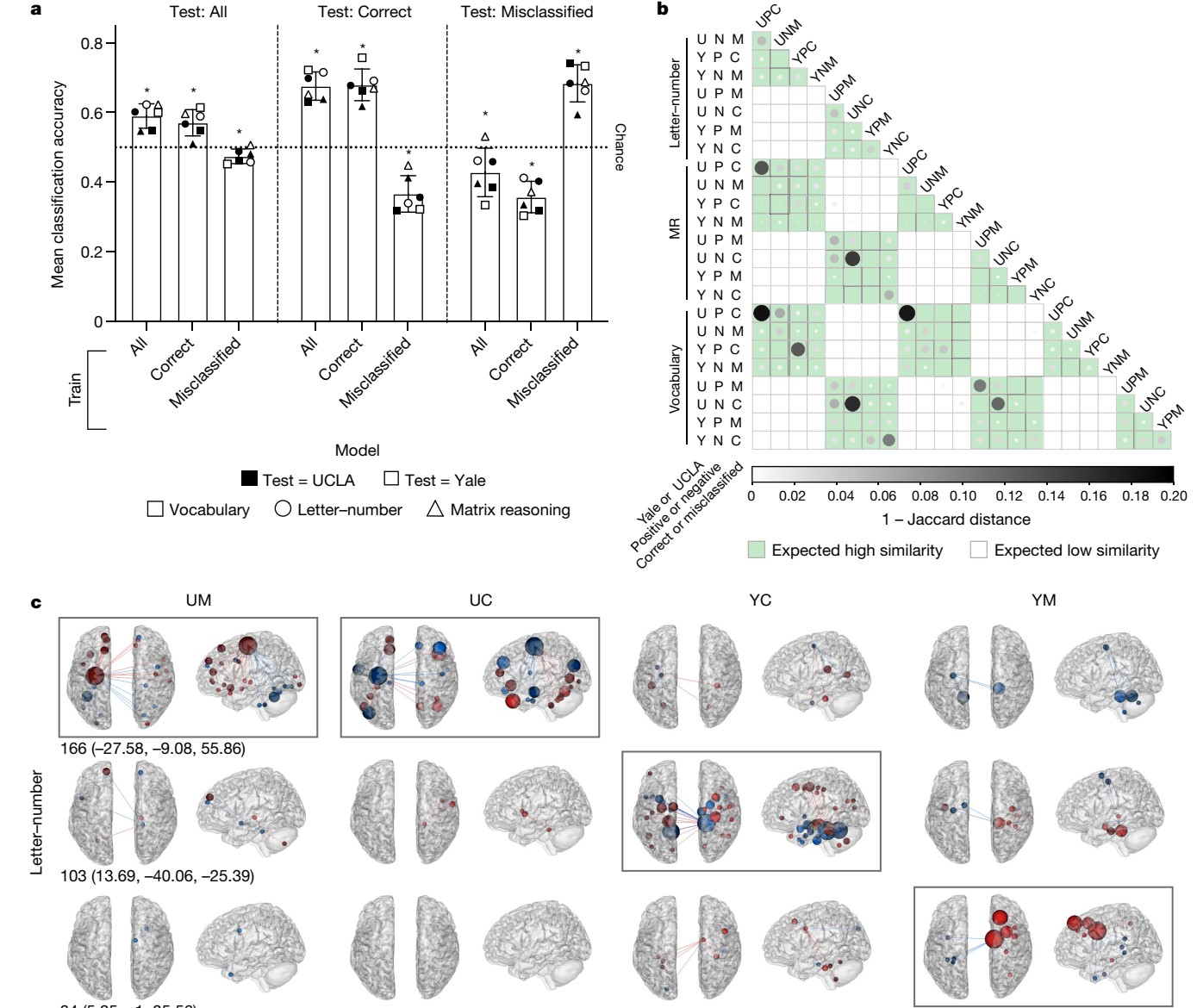

**Fig. 3 | Misclassification generalizes to an independent dataset. a**, For each of the three measures common to both datasets (LN, MR, vocabulary), six models were trained: one using all Yale participants (training set *n*: 80, 58, 58 for LN, MR, vocabulary, respectively), one using Yale CCP (50, 40, 40), one using Yale MCP (30, 18, 18), one using all UCLA participants (100, 78, 74), one using UCLA CCP (64, 48, 50) and one using UCLA MCP (36, 30, 24). Each model was applied to all high and low scorers in the test dataset (see Supplementary Tables 4 and 5 for test-set sizes), and the results are displayed as accuracy in all test participants, only in test CCP (Test: Correct), and only in test MCP (Test: Misclassified). *Significantly different from chance (mean accuracy using permuted data; dotted line presented for visualization only) by two-tailed, nested ANOVA; all *P* < 0.0001, FDR adjusted (nine tests). Bar height, grand mean; error bars, s.d. **b**, Similarity of model pairs, with similarity = 1 – Jaccard distance, thresholded at *P* < 0.05, by the hypergeometric cumulative distribution function. Models are divided into edges that are positively and negatively correlated with phenotype to facilitate interpretation. Larger, darker circles indicate increased similarity. Number of edges in each model (that is, selected on at least 75% of iterations): 30–374. Cells shaded on the basis of predicted patterns of similarity. **c**, Each model's highest-degree node and its incident edges are visualized in all models. Models for which the depicted node is the highest-degree node are enclosed in grey rectangles. Red, positive relationship with phenotype; blue, negative relationship with phenotype. Node size scales with degree, and nodes are coloured red if, of the edges incident to that node, the number of edges positively related to phenotype is greater than or equal to the number of edges negatively related to phenotype; blue otherwise. P, edges positively correlated with phenotype; N, edges negatively correlated with phenotype; UM, Yale MCP train, UCLA test; UC, Yale CCP train, UCLA test; YC, UCLA CCP train, Yale test; YM, UCLA MCP train, Yale test. Node number in the Shen atlas (MNI coordinates).

MCP and UCLA-test CCP letter–number models, and exhibits similar incident edges, but with largely opposite relationships to phenotype (for example, edges that are positively correlated with letter–number performance in CCP are negatively correlated with performance in MCP). Across datasets, similarities can also be observed. For example, similar edges between node 166 and cerebellum were selected for their negatively correlated with phenotype in MCP in both UCLA and Yale data (Fig. 3c). However, the inverted nature of the MCP and CCP models does not entirely account for their selected edges, as evidenced by differences in patterns of FC across models. This suggests that MCP and CCP models comprise a core set of overlapping edges as well as sets of distinct edges.

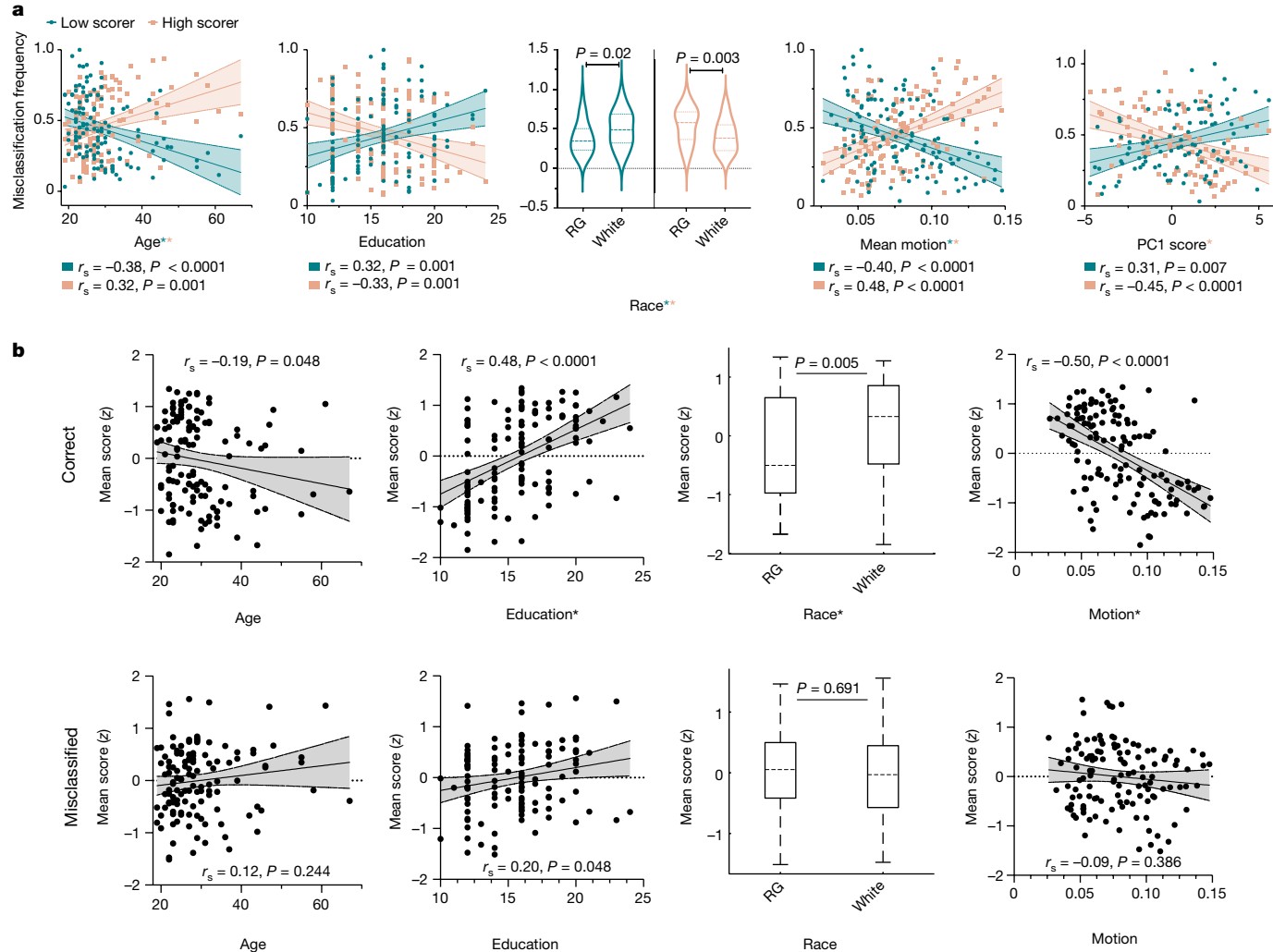

**Fig. 4 | Frequently misclassified participants defy stereotypical profiles.**
**a**,**b**, Data are shown for all covariates that were found to have significant pairwise relationships with misclassification frequency by two-tailed rank correlation and Mann–Whitney *U* test. **a**, Relationship with misclassification frequency, averaged separately across measures on which participants scored high ('high scorer') and low ('low scorer'). *Significant (*P* < 0.05) in corresponding full regression of low- or high-scorer misclassification frequency on these covariates. **b**, Relationship with mean scores, averaged separately across measures on which participants were frequently correctly (top) and incorrectly (bottom) classified. *Significant (*P* < 0.05) in full regression of mean (correct or misclassified) score on these covariates. Lines and shading: best-fit line from simple linear regression with 95% confidence bands. Violin plot lines represent median and quartiles. Box plot centre line and hinges represent median and quartiles, respectively; whiskers extend to most extreme non-outliers. All reported *P* values FDR adjusted (**a**, 30 tests; **b**: 8 tests). See Supplementary Tables 9 and 10 for relationships between misclassification frequency, mean score and all tested covariates, as well as sample sizes. RG, racialized groups.

## Model failure reveals groups that defy stereotypical profiles

Given the accumulated evidence that model failure is systematic, with a subset of participants reliably misclassified and misclassification generalizing across measures and datasets, we turn finally to the question of who these misclassified individuals are. To characterize misclassified Yale participants, misclassification frequency was related to 15 covariates (characterizing participant demographics, clinically relevant experiences, in-scanner head motion and overall cognitive ability; for a description of how these covariates were selected and their relationships to each other, see 'Exploring contributors to misclassification' in Methods and Supplementary Fig. 2). Misclassification frequency was averaged separately for measures on which participants scored low and high, to yield, for each participant, two mean misclassification frequencies. We note that many of these covariates, particularly race, are proxies that were available, but are neither biological nor causal and obscure much heterogeneity of culture, identity and experience

(see 'Causes and implications of model failure' (below) and 'Additional limitations and future directions' in Supplementary Discussion). Of these covariates, five were significantly related to misclassification frequency in both low and high scorers (Fig. 4a): age (low $r_s = -0.38$, $P < 0.0001$; high $r_s = 0.32$, $P = 0.001$), race (low group median difference = $-0.14$, $P = 0.02$; high group median difference = $0.19$, $P = 0.003$), education (low $r_s = 0.32$, $P = 0.001$; high $r_s = -0.33$, $P = 0.001$), overall cognitive ability (low $r_s = 0.31$, $P = 0.007$; high $r_s = -0.45$, $P < 0.0001$) and motion (low $r_s = -0.40$, high $r_s = 0.48$, both $P < 0.0001$; all $P$ values FDR adjusted).

Together, these covariates reflect a stereotypical profile that the model detected and used, with a high misclassification frequency in participants who defied this profile. For example, in CCP, an increased amount of education was associated with an increased mean score on the neurocognitive battery ($r_s = 0.48$, $P < 0.0001$, FDR adjusted; Fig. 4b). Correspondingly, individuals with low neurocognitive scores and high education, or with high scores and low education, were frequently misclassified. In keeping with this stereotype-defying profile,

the relationship between education and mean score was substantially diminished among MCP, and the relationships between mean score and age, race and motion—all significant in CCP (Fig. 4b, 'Correct')—were abolished in MCP (Fig. 4b, 'Misclassified'). Together, these results suggest that models reflect stereotypical demographic profiles of high and low scorers that, when violated, result in misclassifications.

Trends in the UCLA and HCP data using comparable, available covariates are similar (Extended Data Figs. 2e and 3e and Supplementary Figs. 3 and 4). Notably, in the UCLA sample, self-reported symptom severity and use of psychiatric medication were related to misclassification frequency (Supplementary Table 9), and low scorers who did not have a mental health diagnosis (assessed by diagnostic interview) were slightly, although not significantly, more likely to be misclassified than low scorers who did (misclassification frequency median difference = 0.25, $P$ = 0.07). Symptom severity, medication status and diagnosis in Yale and UCLA datasets were in many cases related to mean score in CCP, with illness tracking worse performance (Supplementary Table 10). There was no evidence of a relationship between symptom severity and mean score in the HCP sample (both $P$ > 0.1; Supplementary Table 10), which focused recruitment on healthy individuals. Together, these results reflect both the potential relevance of mental illness to stereotypical profiles and misclassification frequency, and the dataset-specificity of these profiles and who defies them (see 'Causes and implications of model failure', below). Relationships between misclassification frequency, mean score and all tested covariates are presented in Supplementary Tables 9 and 10.

Finally, we considered two additional questions raised by these findings. First, we compared the FC patterns of CCP and MCP groups to identify any group differences in functional brain organization that may explain misclassification. We found no consistent differences between groups at either the edge or the network levels (Extended Data Fig. 5). Second, these findings raise the concern that models reflect only demographic and clinical variables, not the neurocognitive constructs of interest. To investigate this, we regressed GFC edge summary scores on phenotypic scores and covariates. In all cases, the full model was preferred to a reduced model that included only score (all $P$ < 0.0005 by extra sum-of-squares $F$-test). Score was in most cases significantly associated with edge summary score after controlling for all covariates, and covariates associated with edge summary score were, as expected, highly overlapping with those that track misclassification frequency (Extended Data Table 1).

## Causes and implications of model failure

In this work, we interrogate model failure to better understand group differences in brain–phenotype relationships. Across three datasets, a wide range of predicted measures and various analysis approaches, we find notably consistent results: model failure occurs reliably in a subset of individuals, generalizes across phenotypic measures and datasets and is associated with phenotype scores that do not fit the sample's stereotypical profile for high and low scorers.

Together, these results show that one model does not fit all; model failure identifies subgroups that require distinct predictive models (see 'Model failure as a tool for subtyping' in Supplementary Discussion). Furthermore, they show us that we are often predicting not unitary outcomes, but rather outcomes of interest intertwined with constellations of covariates. This is crucial to acknowledge, both because these stereotypical profiles can teach us about the predicted construct and its potential biases, and because these profiles have practical and conceptual implications for model generalizability (stereotypes, and thus models, do not fit all individuals) and model interpretation (identified brain activity patterns may represent elements or consequences of this profile, not the phenotype of interest, per se). Model failure is thus inextricably intertwined with biases in input data, and these issues must be jointly addressed if brain–phenotype models

are to yield useful neuroscientific and clinical insights. We turn next to a discussion of these points, and close with a proposed framework for future work.

## Models predict complex score profiles

Our results suggest that brain activity-based models are often predicting complex profiles rather than unitary cognitive processes, highlighting the need to consider these profiles and the influence of sample representation on them. For example, a sample of same-age individuals would not demonstrate a relationship between age and performance, whereas targeted recruitment of groups with distinct medical health histories may render this variable relevant to performance and misclassification. In keeping with this intuition, the HCP sample, which selectively recruited healthy individuals, did not show a relationship between psychopathology and test performance, a relationship that exists in both the Yale and the UCLA samples.

Other effects of sample representation on phenotypic score profiles are suggested by many intersecting literatures. Cultural influences on task strategies and test performance are well documented[43,44], and neuropsychological test performance differs by factors such as life course epidemiology, education quality, acculturation and physical health[45–47]. Many tests are thus composite measures[20], and it is these composites that our models are predicting.

Furthermore, relationships between covariates and the outcome of interest may be complex, and differentially affect brain–phenotype relationships (see 'Covariate–outcome relationships may be varied and complex' in Supplementary Discussion). Verbal memory offers an illustrative example of group differences in covariate–outcome relationships and a use-case for the utility of subgroup-specific modelling. In native English speakers, the articulation rate of subvocal rehearsal tracks digit and word span performance. This relationship is substantially attenuated in native Mandarin speakers, who also perform better overall than English speakers on the task[48]. Together, these findings suggest that there is a meaningful difference in the cognitive processes that are associated with verbal memory across cultural groups. To account for this, brain–verbal memory modellers may build separate models for these groups. This would be likely to increase model performance for each group, consistent with our results (Fig. 3) and with the previous finding that matching training and test data for confounding relationships maximizes classification accuracy[49]. It would also reveal whether these group-specific models track distinct processes (that is, neural correlates of articulation rate in English speakers, and of other factors—such as nonverbal rehearsal processes or increased capacity of the phonological store—in Mandarin speakers[48]). By situating the interpretation of verbal memory test scores—and of corresponding brain-based predictive models—in the context of this existing literature, more appropriate models may be built for each group and a more nuanced interpretation of these models achieved.

The existence of factors that track performance in one group but not in another is consistent with our finding that, overall, the same covariates that track misclassification also track score in CCP but not in MCP. Because models are predicting profiles of which these covariates are a part, individuals who defy the profile will require a different brain–phenotype model. In all three studied samples, for example, individuals with more education tended to score higher on neurocognitive tests, but this correlation was not perfect. A substantial number of individuals with low education scored high, and vice versa, and these individuals were frequently misclassified in the Yale and HCP samples (Fig. 4a, Extended Data Fig. 3e and Supplementary Tables 9 and 10). The observation of this pattern in both the clinically heterogeneous Yale and healthy HCP samples suggests that it cannot be exclusively explained by disease processes. These cases thus present an opportunity to study potential correlates of resilience, obstacles to performance and alternative cognitive strategies.

Equally important and not mutually exclusive is the opportunity to use such cases, and the profiles they defy, to explore sources of bias encoded in input data. That is, if test scores are themselves biased, the models may be as well. Such model bias has been described in applications of machine learning algorithms ranging from criminal justice[35,36] to healthcare[37,38]. Care must be taken to interpret results accordingly. For example, African American and Hispanic or Latino American individuals tend to score lower than white Americans without Hispanic or Latino ancestry on neuropsychological tests[46]. These group differences are complex, often reductionist and non-causal; efforts to explain them have focused on differences in factors such as education quality[50], acculturation[47], neighbourhood disadvantage[51] and research methodology[18,52]. Although consensus causal explanations remain an open question, the pervasiveness of such bias in commonly used phenotypic measures[18,20] is a call to action to carefully consider what brain-based models are truly predicting. Indeed, race tracked neuropsychological test performance in all three studied samples. And despite the fact that our models had no access to information about race, race was related to misclassification frequency in the Yale and HCP samples, such that high-scoring participants who identify with racialized groups (see Methods) were frequently misclassified as low scoring, and vice versa for white participants. This finding is reminiscent of the errors made by the Correctional Offender Management Profiling for Alternative Sanctions (COMPAS, now 'Equivant') system[36] and of recent evidence for 'prediction shift' in African American individuals[14].

We seek to avoid overinterpretation of these findings and note again that race is a non-causal, non-biological proxy for unmeasured variables that obscures much heterogeneity in these samples (see 'Additional limitations and future directions' in Supplementary Discussion). What our results do reveal is unintended and easily missed bias in both model inputs (that is, phenotypic measures limited by available assessment tools[18,20], such as those comprising the NIH Toolbox[53,54]) and model outputs (that is, the profiles to which models correspond). This bias matters for two reasons: (1) it may yield the right predictions for the wrong reasons; researchers may interpret the model as the neural representation of a unitary phenotypic construct or may acknowledge the role of covariates but wrongly assume causality; and (2) it determines the limits of model generalizability, which in turn guide the practical applications of the models. Given, as we demonstrate, that models represent a composite profile, and that model generalizability will be limited to the group that fits this profile, it is critical for modellers to characterize these stereotypical profiles in their samples along multiple and intersecting dimensions (see specific recommendations in 'Limitations and future directions', below).

## Macroscale circuits associated with phenotypes

Notably, we show that misclassified individuals—those who defy the consensus score profile—do not have a distinct brain organization; rather, it is the relationship between brain and phenotype that differs between CCP and MCP. Specifically, MCP do not require an entirely distinct model to classify their phenotypes. This finding ran contrary to our expectation that edges relevant to phenotype in CCP would not be systematically related to phenotype in MCP, and that misclassification would thus identify groups with a different macroscale neural circuitry underlying a given phenotype. Rather, MCP positively correlated edges overlap significantly with CCP negatively correlated edges and vice versa, and simply inverting the model trained on one group yielded successful classification of the other, suggesting a stereotype-based ingroup–outgroup dichotomy for each model. It is possible, however, that in a more demographically homogeneous sample, the influence of these stereotypical profiles would be minimized and more nuanced group differences in phenotype-relevant circuitry would be observable.

In addition, we do not seek to suggest that these predictive models reflect only a constellation of covariates; there is likely to be variance in phenotype and shared variance between brain and phenotype that are attributable to the cognitive processes of interest. Indeed, there is an extensive psychometric literature describing the construct validity of neuropsychological measures[55], and several recent studies have demonstrated the phenotypic specificity of FC-based predictive models[8,56]. In the Yale, UCLA and HCP data, even after controlling for all included covariates, the relationship between brain and phenotype remained significant in most cases (Extended Data Table 1). The macroscale circuits revealed by FC-based predictive modelling can thus be interpreted as the neural representation of a complicated mixture of the construct of interest and a range of sample-dependent demographic and clinical variables.

## Limitations and future directions

Disentangling these relationships presents an important and broad opportunity for future research. These issues are relevant to work that relates neural and phenotypic data at all levels of analysis, and thus will not be limited to human neuroscience; as individual differences gain increasing attention in cellular and systems neuroscience[57], precise phenotypic characterization and model interpretation will be paramount. Our results thus encourage each modeller to collect the data necessary to identify and, to the extent possible, correct stereotypical profiles for a given phenotype in their sample.

Doing so must begin with study design. Given the importance of sociodemographic and clinical covariates to brain–phenotype modelling analyses, future work should further characterize score profiles, looking to best-practice guidelines to collect more expansive and inclusive demographic data[58], increase the enrolment of underrepresented groups and exchange proxies such as race for more meaningful causal or explanatory variables[59-64] (see 'Additional limitations and future directions' in Supplementary Discussion for more on the use and characterization of race in this work). In the service of result generalizability and as a proof of principle, we present the covariates that are related to model failure across all studied phenotypic measures, but such future work will permit the identification of more precise and phenotype-specific stereotypical profiles ('Additional limitations and future directions' in Supplementary Discussion). In parallel, phenotypic measures must be carefully selected and administered to maximize their validity[18,52]. These choices may be guided by tools to evaluate the risk of bias in study design (for example, PROBAST, Step 3[65]).

Then, once data are collected, they must be used. That is, modelling analyses must be adapted to ask what combination of factors our outcome of interest measures, and how we can interpret related patterns of brain activity. First, statistical tools may be used to isolate, to the extent possible, the phenotype of interest. When standardizing phenotypic measures, norms should be carefully considered to ensure appropriateness (for example, for the NIH Toolbox[66,67], but see refs. [53,54]). Furthermore, data may be corrected for identified confounds. Many approaches to confound correction rely on the assumption that a single covariate–outcome relationship holds for all individuals in the sample. If this is not the case, as we show here, then such correction will fail, and may even induce a confounding relationship where in truth none exists[68]. To address this issue, more sophisticated correction approaches that account for sample-specific stereotypes will be necessary (for example, the use of crossed-term confounds, confound-based sample splitting[68], inverse probability weighting[69] or post-hoc confound control[70]). Inevitably, however, confounds will remain. It is incumbent on the modeller to use the previously collected, comprehensive sociodemographic data to precisely characterize these persistent confounds and interpret resulting models accordingly: as group-specific neural representations of composite phenotypes. See Extended Data Fig. 6 for a summary of these steps.

To ignore these issues is to risk missing structured model failure, and the development of models that only apply to a specific—but uncharacterized—slice of the population. Only by integrating standard model evaluation criteria (for example, accuracy, sensitivity and specificity) with more thorough investigations of model failure can we hope to define the population to which each model generalizes[71] and move away from the limitations of a one-size-fits-all approach. That models pick up on and use stereotypical profiles is not always, in itself, a problem for data-driven studies of brain–phenotype relationships. However, we must characterize these profiles to identify potentially harmful biases and to know whether and how a given model applies to the individual sitting before us. Doing so opens a world of possible applications for brain-based predictive models, chief among them the identification of neuromarkers that both shed light on the biological basis of disease and guide intervention.

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

# Methods

## Datasets

Three datasets were used in these analyses. Primary analyses use data collected at Yale from February 2018–March 2021, and external validation analyses use the UCLA Consortium for Neuropsychiatric Phenomics (CNP)[17], which is of comparable size to the Yale dataset and includes individuals with mental health diagnoses, and the Human Connectome Project (HCP)[16], which includes only healthy participants and is substantially larger (thus addressing concerns about mean and variance of predictive model accuracy in small samples[74,75]; although it is still not as large as samples called for in recent work[27], we note that the concerns raised in that study are not directly relevant to our work, as the former focuses on within-sample brain–phenotype associations, whereas our analyses rely exclusively on prediction, presenting brain–phenotype relationships that generalize to unseen data). Each dataset is described below, with demographic and clinical information for each reported in Supplementary Table 3 and Extended Data Fig. 7. Together, these datasets comprise nearly 1,000 individuals, and each is of comparable size to or larger than datasets that are commonly used in brain–phenotype modelling work (for example, refs. [1,6,10]).

**Yale dataset.** Participants completed an MRI scan followed by a post-scan neuropsychological and self-report battery; the scan and post-scan battery were each approximately 2 h in length, and, to cover a broad cognitive landscape, were designed to correspond to Research Domain Criteria (RDoC)[76] domains and constructs (Supplementary Tables 1 and 2).

**Yale imaging parameters, study design and preprocessing.** All imaging data were acquired on three harmonized Siemens 3T Prisma scanners with a 64-channel head coil at the Yale Magnetic Resonance Research Center. For alignment to common space, a high-resolution T1-weighted three-dimensional anatomical scan was acquired using a magnetization-prepared rapid acquisition with gradient-echo (MPRAGE) sequence (208 slices acquired in the sagittal plane, repetition time (TR) = 2,400 ms, echo time (TE) = 1.22 ms, flip angle = 8°, slice thickness = 1 mm, in-plane resolution = 1 mm × 1 mm). Functional data were acquired using a multiband gradient-echo-planar imaging (EPI) sequence (75 slices acquired in the axial-oblique plane parallel to the AC–PC line, TR = 1,000 ms, TE = 30 ms, flip angle = 55°, slice thickness = 2 mm, multiband factor = 5, in-plane resolution = 2 mm × 2 mm).

Participants completed eight functional scans—two resting-state runs and six task runs—each 6 min, 49 s long (including an initial shim). The first and last functional scans were resting-state runs; participants were asked to rest with their eyes open, and a fixation cross was displayed. Participants completed each of the six tasks during the remaining runs, with task order counterbalanced across participants. Tasks were presented using Psychtoolbox-3[77].

For a detailed description of the design of each task, see Supplementary Table 1 and previously published work[78]. In brief, the tasks included adaptations of: (1) 2-back[79] (working memory; adapted from a previous study[80]; one block of scene stimuli and one block of emotional face stimuli[81,82]); (2) stop signal (response inhibition)[83]; (3) card guessing (reward)[84,85]; (4) gradual-onset continuous performance (gradCPT; sustained attention)[73]; (5) reading the mind in the eyes (social)[86]; and (6) movie watching tasks. Each task, with the exception of movie watching, was preceded by instructions and practice, after which the participant had an opportunity to ask questions about the task. Responses were recorded on a 2 × 2 button box. A fixation cross was displayed between tasks.

After the scan, participants completed a battery of scales selected from extensively validated neuropsychological and self-report measures[87–100] (Supplementary Table 2), as well as a structured diagnostic interview[101], all administered by research assistants and graduate students trained by a clinical neuropsychologist. All phenotypic data were hand-scored by two independent raters, and age-adjusted, normed neuropsychological test scores (IQ, scaled, T and $z$ scores, as relevant), derived from corresponding test manuals, were used in subsequent analyses. Further demographic adjustments were not used given their variable availability and utility in a research setting[18,54,102,103], but we note that such adjustments—which are often used in clinical settings—can increase sensitivity to neurocognitive deficits, control for variance associated with premorbid factors and improve performance interpretation, making them an important area for future investigation, particularly in the study of acquired cognitive changes[67] (see 'Limitations and future directions' in 'Causes and implications of model failure'). A small subset of participants was recruited for a related study of bipolar disorder; their study sessions were comparable with the exception of an additional in-scanner task (not included in these analyses) and the use of the Structured Clinical Interview for DSM-5[104] instead of the MINI.

Standard preprocessing procedures were applied to imaging data. Structural scans were skull stripped using an optimized version of the FMRIB's Software Library (FSL)[105] pipeline (optiBET)[106]. Motion correction was performed in SPM12[107]. Nonlinear registration of the MPRAGE to the MNI template was performed in BioImage Suite[108], and linear registration of the functional to the structural images was performed using a combination of FSL and BioImage Suite linear registration tools to optimize registration quality. All additional preprocessing steps were performed in BioImage Suite, and included regression of mean time courses in white matter, cerebrospinal fluid and grey matter; high-pass filtering to correct linear, quadratic, and cubic drift; regression of 24 motion parameters[109]; and low-pass filtering (Gaussian filter, $\sigma = 1.55$). All registered data were visually inspected to ensure whole-brain coverage, adequate registration and absence of anomalies, artefact or other data quality concern. Subsequent analyses and visualizations were performed in BioImage Suite, MATLAB (Mathworks), GraphPad Prism and R[110] (packages: ggpairs[111], corrplot[112] and R.matlab[113]).

**Yale participants.** Participants were recruited through broadly distributed community advertisements and referrals from Yale clinics, with an emphasis on recruiting a clinically naturalistic and demographically diverse sample. That is, participants experienced a range of symptom severity and frequently had multiple psychiatric diagnoses, ages were distributed broadly across the lifespan and the sample was enriched for mental illness and identification with racialized groups (Supplementary Table 3 and Extended Data Fig. 7). Note that we use this phrase to describe participants who did not identify as white, in keeping with the literature describing 'racialization' as the process of categorizing, marginalizing, or regarding according to race with socialized tendencies to view white race as the default, and in recognition that such categorization is without meaningful biological distinction[114]. All participants provided written informed consent in accordance with a protocol approved by the Yale Institutional Review Board.

We restricted our analyses to those participants who completed all fMRI scans (six task, two rest), whose grand mean frame-to-frame displacement was less than 0.15 mm and whose maximum mean frame-to-frame displacement was less than 0.2 mm. Several additional participants were excluded because they were found to have an anatomical anomaly that interfered with registration or because of technical issues during their session that compromised data quality. Several participants did not complete all neuropsychological and demographic measures; they were included in the sample but excluded from specific analyses for which they were missing data. Similarly, for several participants, a slightly shorter protocol (25 s shorter than the standard protocol) was used for all or a subset of functional scans; given the known explanation for the missing time points, these participants were not excluded. In total, the sample includes 129 participants (Supplementary Table 3 and Extended Data Fig. 7).

**CNP dataset, study design and preprocessing.** Participants completed both resting-state and task-based fMRI scans, as well as an out-of-scanner battery of neurocognitive and self-report measures. In this study, we used imaging data from all cognitive tasks and rest, data from three of the out-of-scanner neurocognitive measures[98] and sociodemographic and clinical covariates[115,116]. All data were accessed through OpenfMRI (accession number ds000030).

Details of the study design and in-scanner tasks are described in previously published work[17]. In brief, imaging data were acquired on two 3T Siemens Trio scanners. From the imaging data, as for the Yale data, we used the MPRAGE (176 slices acquired in the sagittal plane, TR = 1,900 ms, TE = 2.26 ms, slice thickness = 1 mm, field of view (FOV) = 250 mm, matrix size = 256 × 256) and EPI (34 slices acquired in the oblique plane, TR = 2,000 ms, TE = 30 ms, flip angle = 90°, slice thickness = 4 mm, FOV = 192 mm, matrix size = 64 × 64) scans.

In this work, we used fMRI data acquired during seven runs: eyes-open rest and six tasks performed over two, counterbalanced scan sessions. Runs varied in length. Cognitive tasks included: (1) balloon analogue risk task (BART); (2) paired-associate memory task (2 scans, one each for the encoding (PAMenc) and retrieval (PAMret) components of the task); (3) spatial working memory capacity (SCAP) task; (4) stop signal task (SST); and (5) task-switching (TS) task[17]. Participants received training on each task immediately before the scan. Responses were recorded on a button box.

Participants also underwent a neuropsychological and clinical assessment. From these tests, we used three phenotypic measures that are the same as or comparable to measures used in the Yale dataset: WAIS-IV vocabulary, WAIS-IV matrix reasoning and WAIS-IV letter–number sequencing. Age-adjusted scaled scores obtained from the WAIS-IV manual were used for all analyses.

Preprocessing procedures were similar to those used on the Yale dataset (minor differences include use of SPM8 and the middle, rather than the first, scan as reference for motion correction; an earlier version of the FSL and BioImage Suite linear registration protocol; and a Gaussian low-pass filter with $\sigma = 1$), and have been described previously[9]. All registered data were visually inspected to ensure whole-brain coverage, adequate registration and absence of anomalies, artefact or other data quality concern.

**CNP participants.** Participants were recruited through community advertisements and outreach to clinics; the sample was enriched for mental illness, with some comorbidities being grounds for exclusion[17]. All participants provided written informed consent consistent with procedures approved by the Institutional Review Boards at UCLA and the Los Angeles County Department of Mental Health. We restricted analyses to participants with complete rest, BART, PAMenc, PAMret, SCAP, SST and TS runs, with grand mean frame-to-frame displacement less than 0.15 mm and maximum mean frame-to-frame displacement less than 0.2 mm, and without nodes entirely lacking coverage (see 'Functional parcellation and network definition'), leaving a final sample of 163 participants (Supplementary Table 3 and Extended Data Fig. 7).

**HCP dataset, study design and preprocessing.** As in the Yale and UCLA datasets, participants completed both resting-state and task-based fMRI scans, as well as an out-of-scanner battery of neurocognitive and self-report measures. In this study, we used imaging data from all tasks and rest, as well as summary scores from the NIH Toolbox assessments[117], and sociodemographic and clinical covariates matched, to the extent possible, to those used in primary analyses[95,117,118]. All data were released as part of the HCP 900 Subjects release and are publicly available on the ConnectomeDB database (https://db.humanconnectome.org).

Details of the study design, imaging protocol and in-scanner tasks have been extensively described[16,119–121]. In brief, all MRI data were acquired on a 3T Siemens Skyra using a slice-accelerated, multiband, gradient-echo, EPI sequence (72 slices acquired in the axial-oblique plane, TR = 720 ms, TE = 33.1 ms, flip angle = 52°, slice thickness = 2 mm, in-plane resolution = 2 mm × 2 mm, multiband factor = 8) and a MPRAGE (256 slices acquired in the sagittal plane, TR = 2,400 ms, TE = 2.14 ms, flip angle = 8°, slice thickness = 0.7 mm, in-plane resolution = 0.7 mm × 0.7 mm).

In total, 18 fMRI scans were conducted for each participant (working memory (WM) task, incentive processing (gambling) task, motor task, language processing task, social cognition task, relational processing task, emotion processing task and two resting-state scans; two runs per condition (one left-to-right (LR) phase encoding run and one right-to-left (RL) phase encoding run))[120,121] split between two sessions. Participants received instructions for each task outside of the scanner, as well as a brief reminder of the task instructions and button box response mappings immediately prior to each task.

Participants also completed an extensive out-of-scanner battery, including most of the NIH Toolbox measures[121]. Given the lack of exact correspondence between the NIH Toolbox and the neuropsychological tests used in the Yale and UCLA datasets, we used the Toolbox composite age-adjusted scaled scores (variables: CogFluidComp_AgeAdj and CogCrystalComp_AgeAdj), corresponding broadly to fluid cognitive functions and verbal reasoning, respectively.

Preprocessing procedures have been described previously[24]. The HCP minimal preprocessing pipeline was used on these data[122], which includes artefact removal, motion correction and registration to standard space. All subsequent preprocessing was performed in BioImage Suite[108] and included standard preprocessing procedures[1], including removal of motion-related components of the signal; regression of mean time courses in white matter, cerebrospinal fluid and grey matter; removal of the linear trend; and low-pass filtering. Mean frame-to-frame displacement was averaged for the LR and RL runs, yielding nine motion values per participant; these were used for participant exclusion and motion analyses.

**HCP participants.** Participants were recruited from families containing twins in Missouri. Participants were healthy, broadly defined, and reflective of the ethnic and racial composition of the USA as documented in the 2000 Census[16]. The scanning protocol (as well as procedures for obtaining informed consent from all participants) was approved by the Institutional Review Board at Washington University in St Louis. We restricted analyses to participants with complete rest and task runs, with complete zygosity data (necessary to respect family-related limits on exchangeability for permutation testing[40,41]), with grand mean frame-to-frame displacement less than 0.15 mm and maximum mean frame-to-frame displacement less than 0.2 mm, and without nodes entirely lacking coverage. We further excluded participants with identified quality control issues B–D or other major issues described on the HCP wiki as of October 2021 (for example, RL runs processed with the incorrect phase encoding direction), or who failed preprocessing, leaving a final sample of 664 participants (Supplementary Table 3 and Extended Data Fig. 7).

## Functional parcellation and network definition

The Shen 268-node atlas derived from an independent dataset using a group-wise spectral clustering algorithm[123] was applied[1] to the preprocessed Yale, UCLA and HCP data. After parcellating the data into 268, functionally coherent nodes, the mean time courses of each node pair were correlated and correlation coefficients were Fisher transformed, generating eight 268 × 268 connectivity matrices per Yale participant, seven 268 × 268 connectivity matrices per UCLA participant (one per fMRI run), and nine 268 × 268 connectivity matrices per HCP participant (averaged across RL and LR runs for each in-scanner condition).

To ensure that results are robust to parcellation choice, we repeated analyses using the Shen 368-node atlas, derived using a combined

approach: data-driven parcellation of cortical areas, anatomic delineation of subcortical regions and a cerebellum parcellation based on the Yeo 17-network parcellation[124] (Extended Data Fig. 1).

### Phenotype classification

A modified version of connectome-based predictive modelling[25]—which crucially tests models on previously unseen data (unlike explanatory models[13,15])—was used to classify phenotypic scores as high or low using FC. First, phenotypic scores were binarized to generate unambiguous classification outcomes and avoid common sources of bias in comparisons of observed and predicted outcomes[125]. To ensure that all participants understood and performed the neurocognitive tests as intended, we used normative means and standard deviations for each measure to exclude outlier extremely low scorers (score ≤ mean − 3 × s.d.)[126], as is common and recommended practice[127–129]. Scores less than or equal to the normative mean − 1/3 × s.d. were considered low (label = −1); scores greater than or equal to the normative mean + 1/3 × s.d. were considered high (label = 1); thresholds were rounded as relevant. To ensure that results are robust to this thresholding choice, we repeated analyses using the normative mean as the cut-off (that is, high if greater than mean; low if less than mean). For results, see Extended Data Fig. 1 ('Phenotype mean split'). Leave-one-out cross-validation was used, such that one participant was left out as test data and training data were selected from the remaining participants (for results using 10-fold cross-validation, see Extended Data Figs. 1 and 3). Only high and low scorers were classified, and from the training set, only high and low scorers were considered. From that subset, the larger class was pseudorandomly undersampled to enforce balanced classes[15], and the resulting subset was used to train the classifier.

To do so, each edge across all training participants was correlated with their labels for the given phenotypic measure. Using a significance threshold ($P < 0.05$, uncorrected[25]), two sets of edges were selected—one positively correlated with labels, and one negatively correlated with labels. Those edges were separately summed to derive two edge summary scores for each training participant, one each for the positively and negatively correlated edge sets. These scores were in turn normalized ($z$-scored) and submitted to a linear SVM to classify low versus high scores in the training set. Selected positively and negatively correlated edges were then summed and normalized in the test data, and the trained classifier was applied to these scores to classify the test participant(s) as high or low scoring. To ensure that results are robust to classification algorithm, we repeated these analyses using two additional, commonly used algorithms for supervised learning that, together with linear SVM and linear regression (see below), reflect the full range of model interpretability and complexity[130]: an ensemble of weak learners and a fully connected neural network, both implemented in MATLAB (Mathworks). In both cases, we used a subset of phenotypic measures for hyperparameter optimization (ensemble learners: ensemble aggregation method, number of learners, learning rate (where relevant) and minimum leaf size; neural network: activation functions, standardization, regularization term strength and layer sizes). We used all available high and low scorers' GFC and phenotypic data, undersampled to balance class size, in a leave-one-out manner for each optimization analysis, and used the best consensus hyperparameters (ensemble learners: method = bagging, number of learners = 150, minimum leaf size = 1; neural network: activation function = none; standardization = true; lambda = $1.34 \times 10^{-5}$; layer sizes = 8, 200 and 8) to classify all 16 phenotypic measures as in main analyses, using all selected edges (correlation $P < 0.05$, as in main analyses) as features. See Extended Data Fig. 1 for results.

Test participants were considered misclassified (MCP) versus correctly classified (CCP) if their predicted label did not match their observed label. Classification was repeated iteratively, with each participant excluded once, and overall accuracy was calculated as the number of correctly classified participants divided by the number of classified participants. Accuracy was also calculated separately for high and low scorers to show that performance was comparable in both groups.

This process was repeated 100 times, with distinct subsampling on each iteration (for 10-fold analyses, 50 iterations of subsampling were performed for each 10-fold partition with 20 partitions, yielding 1,000 iterations). The entire pipeline was repeated for every combination of in-scanner condition (rest, tasks and GFC[42] (without regression of task structure[131])) and phenotypic measure.

The significance of classifier performance was assessed by 100 iterations of permutation testing (or, in the case of HCP 10-fold analyses, 1,000 iterations; see above). That is, the classification pipeline was repeated with one modification: high and low phenotypic scores were permuted on each iteration. Permutations were fixed across in-scanner conditions, and respected family-related limits on exchangeability for the HCP dataset[40,41]. $P$ values were calculated as: $P_i = \#\{a_{i,\mathrm{null}} \geq a_{i,\mathrm{median}}\} + 1/$ (no. iters + 1), in which $i$ indexes the phenotypic measure and $a_i$ is the classification accuracy for measure $i$, with $\#\{a_{i,\mathrm{null}} \geq a_{i,\mathrm{median}}\}$ indicating the number of iterations on which the permutation-based classifiers performed as well as or better than the median accuracy of unpermuted data-based classifiers. The resulting $P$ values were FDR adjusted for multiple comparisons (number of tests = number of phenotypic measures in the given dataset; one in-scanner condition per measure; for all conditions, see Supplementary Tables 4–6).

Finally, to ensure that dichotomization of continuous neurocognitive scores did not bias results, leave-one-out cross-validated connectome-based predictive modelling was performed[1,25], using FC from each in-scanner condition to predict performance on each of the 16 Yale phenotypic measures. The difference between predicted and observed scores was deconfounded by regressing it on observed scores[125,132], and this residualized model fit metric, averaged across in-scanner conditions and phenotypic measures, was related to covariates in place of mean misclassification frequency (see 'Exploring contributors to misclassification'). Results were comparable to those from main analyses and are presented in Supplementary Table 11.

### Internal validation analyses

The consistency of MCP was explored at several levels of analysis. First, misclassification frequency for a given phenotypic measure and in-scanner condition was calculated for each classified participant as the number of misclassifications divided by the number of iterations. For each phenotypic measure, the original and permuted distributions of misclassification frequency, concatenated across all in-scanner conditions, were compared (Fig. 2a).

The results for each in-scanner condition were then examined separately by rank correlating misclassification frequency for each condition pair for a given phenotypic measure (Fig. 2b). Given the directional hypothesis, original and permuted correlations were compared by paired, one-tailed Wilcoxon signed-rank test; $P$ values were FDR adjusted for multiple comparisons across the phenotypic measures (for example, 16 tests for the Yale dataset).

Finally, misclassification frequency across phenotypic measures was compared. To do so, misclassification frequency for a given in-scanner condition was rank correlated across phenotypic measures. The resulting phenotype × phenotype misclassification frequency correlation matrices were averaged across in-scanner conditions. Separately, normed scores for each phenotype pair were rank correlated, yielding a phenotype × phenotype similarity matrix. The lower triangles of these matrices were plotted and rank correlated (Fig. 2c), and misclassification frequency correlations were submitted to a hierarchical clustering algorithm (single linkage), with distance = 1 − corr (Fig. 2d). In all cases, NaN values (that is, participants with intermediate (not labelled high or low), low outlier, or missing scores on the relevant measures) were excluded.

Results were replicated in the UCLA and HCP datasets, as relevant (Extended Data Figs. 2 and 3).

## Cross-dataset analysis

To explore the generalizability of misclassification, the classification analysis was adapted such that one dataset was used to train the classifier, and another to test it. In primary analyses using the Yale and UCLA datasets, letter–number, vocabulary and matrix reasoning tests were used as phenotypic measures because they represent relatively distinct cognitive domains and are included in both datasets. GFC was used given the in-scanner task differences across datasets. As described previously ('Phenotype classification'), low outlier scores were excluded, normed phenotypic scores were binarized and training data were subsampled to ensure balanced classes. For each phenotypic measure, six models were trained using (1) all UCLA high and low scorers; (2) only CCP UCLA high and low scorers; (3) only MCP UCLA high and low scorers; (4) all Yale high and low scorers; (5) only CCP Yale high and low scorers; and (6) only MCP Yale high and low scorers. Correct and misclassified participant sets were derived from the GFC-based iteration with the median classification accuracy for the given phenotypic measure (ties settled by maximum correlation with relevant misclassification frequency). In each case, edges were selected on the basis of their correlation with binarized phenotype, and a sparsity threshold was used to facilitate comparison across models given differing sample sizes (that is, 500 most correlated edges; 500 most anticorrelated edges). As before, edge strengths were summed and normalized, and the resulting summary scores were submitted to a linear SVM. Trained models 1–3 were applied to all Yale high and low scorers; models 4–6 were applied to all UCLA high and low scorers. As in within-dataset analyses, given subsampling to balance classes, each analysis was repeated 100 times with different subsamples. Results are presented as the mean (across 100 iterations) fraction of the whole sample that was correctly classified by each model, the mean fraction of each CCP group that was correctly classified (Test: Correct) by each model and the mean fraction of each MCP group that was correctly classified (Test: Misclassified) by each model (Fig. 3a). Within each category (for example, Train: All/Test: All, Train: Correct/Test: Correct, Train: Correct/Test: Misclassified, and so on), a nested ANOVA was used to compare means to chance (mean accuracy for corresponding analyses using permuted phenotypic data), and $P$ values were FDR adjusted across the nine result categories (an adjustment that we note is complicated by the non-independence of the test sets, but which we use as a more conservative estimate of significance than uncorrected $P$ values). Each model's performance was also tested for significance using permutation tests (Supplementary Table 8), and group-specific (CCP- and MCP-based) model performance was compared to whole-sample-based model performance by paired, one-tailed Wilcoxon signed-rank test, given the directional hypothesis that group-specific models outperform whole-sample models.

Edges selected on 75–100% of iterations were compared by Jaccard distance (Fig. 3b) and visualized using BioImage Suite (edges incident to the highest-degree node for each model; illustrative example: Fig. 3c; all cross-dataset models: Extended Data Fig. 4). The above approach for identifying CCP and MCP groups was also used to explore between-group differences in FC in the Yale sample at the edge (by two-sample $t$-test) and network (by the constrained network-based statistic[133]) levels (Extended Data Fig. 5). Cross-dataset classification results were replicated using the Yale and HCP datasets (vocabulary/cIQ and matrix reasoning/fIQ measures; Extended Data Fig. 3d and Supplementary Table 8).

## Exploring contributors to misclassification

To characterize frequently misclassified participants and better understand why they are misclassified, we related 12 covariates (sex, age, self-reported race (binarized given limited sample size, with 'white' indicating that this category was selected as the participant's only racial group; see 'Additional limitations and future directions' in Supplementary Discussion), years of education, self-reported psychiatric symptoms (through the Brief Symptom Inventory[92] global severity index), self-reported stress (through the Perceived Stress Scale[93]), self-reported sleep disturbance (through the Pittsburgh Sleep Quality Index[95] global score), self-reported positive and negative affect (through the Positive and Negative Affect Schedule[94]), presence or absence of mental health diagnosis (ascertained by diagnostic interview), use of one or more psychiatric medications (antidepressant, mood stabilizer, antipsychotic, stimulant, or benzodiazepine or other sedative) and grand mean in-scanner frame-to-frame displacement), along with overall cognitive performance (by principal component analysis on phenotypic scores; top three principal components (PCs) retained given cumulative variance explained), to misclassification frequency, averaged separately for measures on which participants scored high and low. These covariates reflect key demographic features of participants, as well as a range of clinically relevant information. For the latter, we complemented an extensively validated structured diagnostic interview for the major DSM-5 disorders[101] with self-report measures that reflect participants' lived experiences with mental health concerns (for example, sleep disturbance and perceived stress–both common experiences that transcend diagnostic boundaries[134–137]). Relationships between covariates are presented in Supplementary Fig. 2.

For binary covariates (sex, race, diagnosis and medication), two-tailed Mann–Whitney $U$ tests were used to compare misclassification frequency in the two groups. The remaining continuous covariates were rank correlated with misclassification frequency. This was repeated for high- and low-score misclassification frequency. $P$ values were FDR adjusted, with number of tests = 2 × number of covariates (for example, 30 tests in the Yale dataset). Results for covariates significantly related to misclassification frequency are presented in the main text and figures; full results are presented in Supplementary Table 9.

To further explore these relationships, these covariates (excluding score PCs) were related (again by two-tailed Mann–Whitney $U$ test and Spearman correlation) to mean phenotypic scores, $z$-scored within measure and averaged across measures on which each participant was frequently (50% or more of iterations and conditions) and infrequently (less than 50%) misclassified. All $P$ values were FDR adjusted for multiple comparisons. Results and multiple comparison correction for covariates significantly related to misclassification frequency are presented in the main text and figures; full results are presented in Supplementary Table 10.

For additional insight into relationships among covariates, phenotype and misclassification frequency, covariates that were significantly related to misclassification frequency were entered into regressions of misclassification frequency on covariates (separately for high and low scores) and of mean phenotypic score on covariates (separately for frequently and infrequently misclassified groups).

These analyses were repeated in the UCLA and HCP datasets using comparable measures where available (HCP: symptom severity through the Achenbach Adult Self-Report, stress through the NIH Toolbox perceived stress survey and positive affect through the NIH Toolbox positive affect survey; UCLA: symptom severity through the Hopkins Symptom Checklist and diagnosis through the Structured Clinical Interview for DSM-IV). Relationships between covariates are presented in Supplementary Figs. 3 and 4, and results of these analyses are presented in Extended Data Figs. 2e and 3e.

Finally, to investigate whether models reflect any information beyond these covariates, we regressed the difference between positive and negative GFC edge summary scores for a given modelled phenotype (see 'Phenotype classification') on scores for that phenotypic measure, as well as all of the demographic and clinical covariates that were used for that dataset. We performed this analysis using the CCP-based models from each dataset (for Yale data, which were involved in two cross-dataset analyses, Yale–UCLA models were used), for all classified

phenotypes in cross-dataset analyses, using edges that were selected on at least 75% of iterations. The results are presented in Extended Data Table 1.

## Investigating potential confounds

Given the known effects of head motion on FC estimates[138,139] and our finding that head motion was significantly correlated with misclassification frequency (Supplementary Table 9), we repeated the main classification analyses (see 'Phenotype classification') after regressing mean frame-to-frame displacement out of each edge in the training set. The resulting residualized edges were used for model training, and corresponding regression coefficients were used to residualize the test participant's edges before model testing. The misclassification frequency for a given condition–phenotype combination was calculated as described previously, and correlated with the misclassification frequency derived from raw functional-connectivity-based classification (Supplementary Fig. 1).

## Statistical analysis

All analyses of preprocessed data were performed in BioImage Suite, MATLAB versions 2017a, 2018a and 2021b (Mathworks), R v.3.6.0 for macOS and GraphPad Prism v.9.0.1 for macOS. All statistical tests are named and described with corresponding results, and sample sizes—which differ for each phenotypic measure given exclusions (for outlier, missing and intermediate scores)—are noted where relevant (supplementary tables and figure legends). For the main classification analyses, significance was assessed by nonparametric permutation tests. Where relevant, $P$ values were adjusted for multiple comparisons using the false discovery rate. Significance testing is described and reported, along with the number of comparisons for correction, with the corresponding methods and results.

## Reporting summary

Further information on research design is available in the Nature Research Reporting Summary linked to this article.

## Data availability

The primary dataset used in these analyses represents the first set of participants in an ongoing study. Data will be released in waves through the NIMH Data Archive, Collection 3276 (https://nda.nih.gov/edit_collection.html?id=3276). The UCLA CNP data can be obtained from the OpenfMRI database (https://openfmri.org/dataset/ds000030/). The HCP data are publicly available on the ConnectomeDB database (https://db.humanconnectome.org/app/template/Login.vm). Data used to generate the parcellation can be found at http://fcon_1000.projects.nitrc.org/indi/retro/yale_hires.html. Source data are provided with this paper.

## Code availability

MATLAB code to run classification, validation and covariate analyses is available at https://github.com/abigailsgreene/modelFailure/. BioImage Suite tools used for analysis and visualization can be accessed at https://bioimagesuiteweb.github.io/webapp/.

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

**Acknowledgements** This work was supported by funding from the NIH (GM007205 and TR001864 to A.S.G., MH121095 to D.S. and R.T.C., GM007205 to C.H.). D.S.B was partially funded by the National Institute of Mental Health (5 T32 MH 19961-22). We are grateful to have access to the publicly available UCLA CNP and HCP datasets. Data were provided in part by the Human Connectome Project, WU-Minn Consortium (Principal Investigators: D. Van Essen and K. Ugurbil; 1U54MH091657) funded by the 16 NIH Institutes and Centers that support the NIH Blueprint for Neuroscience Research; and by the McDonnell Center for Systems Neuroscience at Washington University. We thank N. Turk-Browne, D. Lee, B. J. Casey, M. Nitabach, D. Barson, S. Gottlieb-Cohen, C. Kelley and F. Arias for discussions and suggestions.

**Author contributions** A.S.G. conceptualized the study, with guidance from R.T.C. and D.S. A.S.G. performed the analyses with support from X.S., S.N., D.S., C.H. and R.T.C. A.S.G. designed the Yale study with support from M.N.S. and R.T.C. A.S.G., C.A.H., J.A. and F.T. collected the Yale dataset, and C.A.H. managed the study. D.S.B. developed the bipolar disorder study, and J.A. collected data for this study. G.S., V.H.S. and S.W.W. supported all clinical aspects of the Yale study. D.S.B. preprocessed the UCLA CNP dataset and provided related support. C.I.C. provided guidance on result interpretation. A.S.G. wrote the manuscript, with contributions from R.T.C., S.N., C.I.C. and C.H., and comments from all authors.

**Competing interests** In the past two years, G.S. has served as a consultant or scientific advisory board member to Axsome Therapeutics, Biogen, Biohaven Pharmaceuticals, Boehringer Ingelheim International, Bristol-Myers Squibb, Clexio, Cowen, Denovo Biopharma, ECR1, EMA Wellness, Engrail Therapeutics, Gilgamesh, Janssen, Levo, Lundbeck, Merck, Navitor Pharmaceuticals, Neurocrine, Novartis, Noven Pharmaceuticals, Perception Neuroscience, Praxis Therapeutics, Sage Pharmaceuticals, Seelos Pharmaceuticals, Vistagen Therapeutics and XW Labs; and received research contracts from Johnson & Johnson (Janssen), Merck and Usona. G.S. holds equity in Biohaven Pharmaceuticals and is a co-inventor on a US patent (8,778,979) held by Yale University and a co-inventor on US provisional patent application no. 047162-7177P1 (00754), filed on 20 August 2018 by Yale University Office of Cooperative Research. Yale University has a financial relationship with Janssen Pharmaceuticals and may receive financial benefits from this relationship. The University has put multiple measures in place to mitigate this institutional conflict of interest. Questions about the details of these measures should be directed to Yale University's Conflict of Interest office. V.H.S. has served as a scientific advisory board member to Takeda and Janssen. The remaining authors declare no competing interests.

**Additional information**
**Correspondence and requests for materials** should be addressed to Abigail S. Greene or R. Todd Constable.

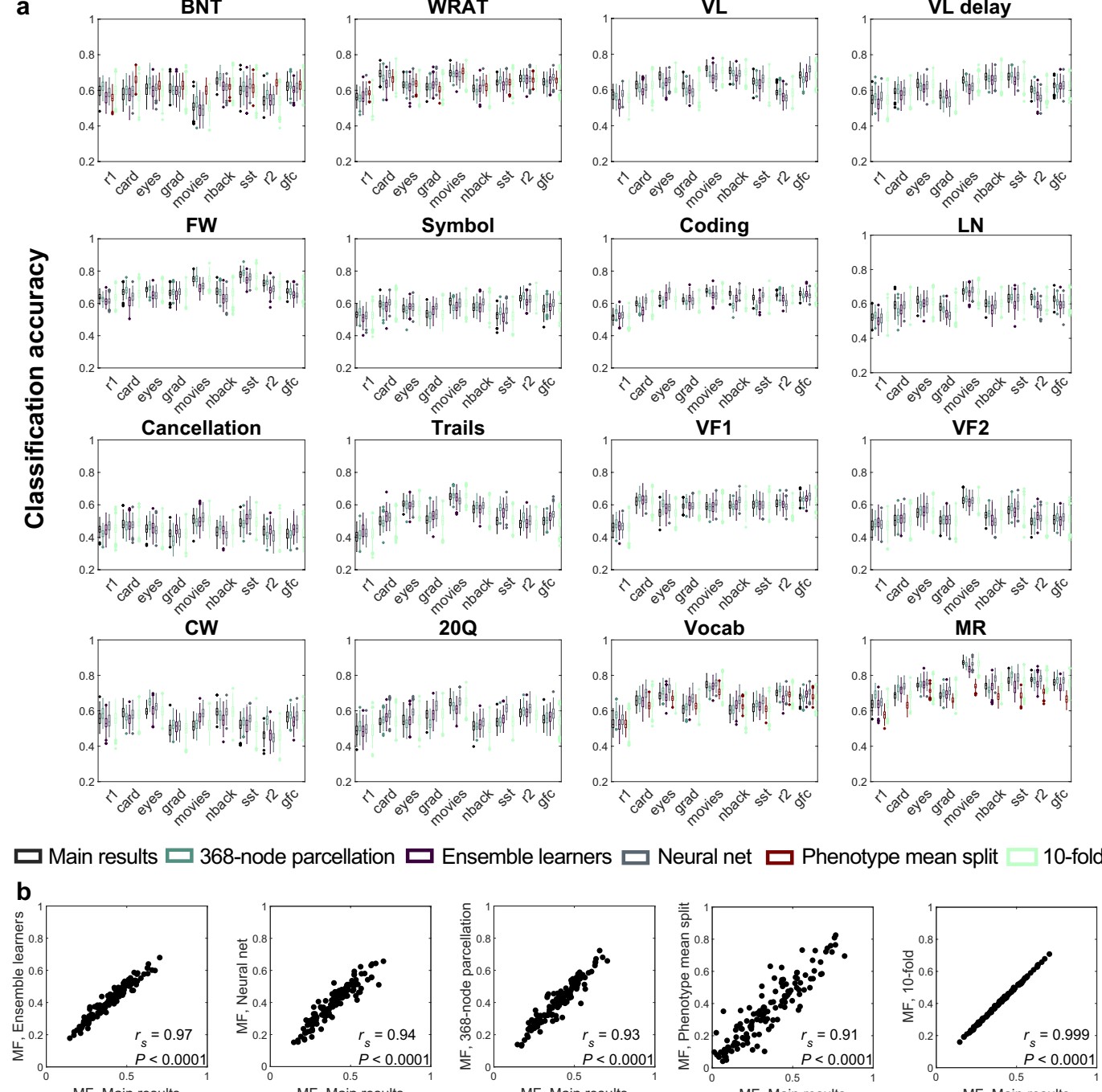

**Extended Data Fig. 1 | Model performance and misclassification frequency are robust to analysis approach.** (a) Classification accuracy for each phenotypic measure using FC calculated from all in-scanner conditions in the Yale dataset, and five different analysis pipelines: an alternative, 368-node parcellation for FC matrix generation, two alternative classification algorithms (ensemble of weak learners and neural network), an alternative phenotypic binarization threshold (mean split), and an alternative (10-fold) cross-validation approach (see Methods for additional description of each analysis). Box plot line and hinges represent median and quartiles, respectively; whiskers extend to most extreme non-outliers; outliers plotted individually (+). Number of classified individuals and size of training sample same as in main analyses (see Supplementary Table 4) for all analyses except mean split (4 measures [see below], number classified = 109-127, training sample size = 72-82) and 10-fold (number classified same as in main analyses, training sample size = 34-110). r1, rest 1; r2, rest 2; grad, gradual-onset continuous performance task; sst, stop signal task; gfc, general FC. (b)

Misclassification frequency (MF), averaged across in-scanner conditions and phenotypic measures to derive a single value per participant, compared between each alternative analysis and main-text analyses. $r_s$, two-tailed rank correlation, $n$ = 128-129, $P$ values FDR adjusted. Note that phenotype mean split is equivalent to mean ± 1/3 × s.d. for scaled scores; mean split-based model accuracy is not reported for these measures, nor are they included in the calculation of misclassification frequency. Given the limited mean split-based results, we repeated this analysis in the HCP data, with comparable results (mean misclassification frequency $r_s$ = 0.86, $P$ < 0.0001). 10-fold results reflect 1,000 analysis iterations per phenotypic measure and in-scanner condition (50 per cross-validation partition); all other analyses reflect 100 iterations. In this and all subsequent figures: BNT, Boston Naming Test; WRAT, Wide Range Achievement Test; VL, verbal learning; FW, finger windows; LN, letter–number sequencing; Trails, trail making; VF, verbal fluency; CW, colour–word interference; 20Q, 20 questions; Vocab, vocabulary; MR, matrix reasoning.

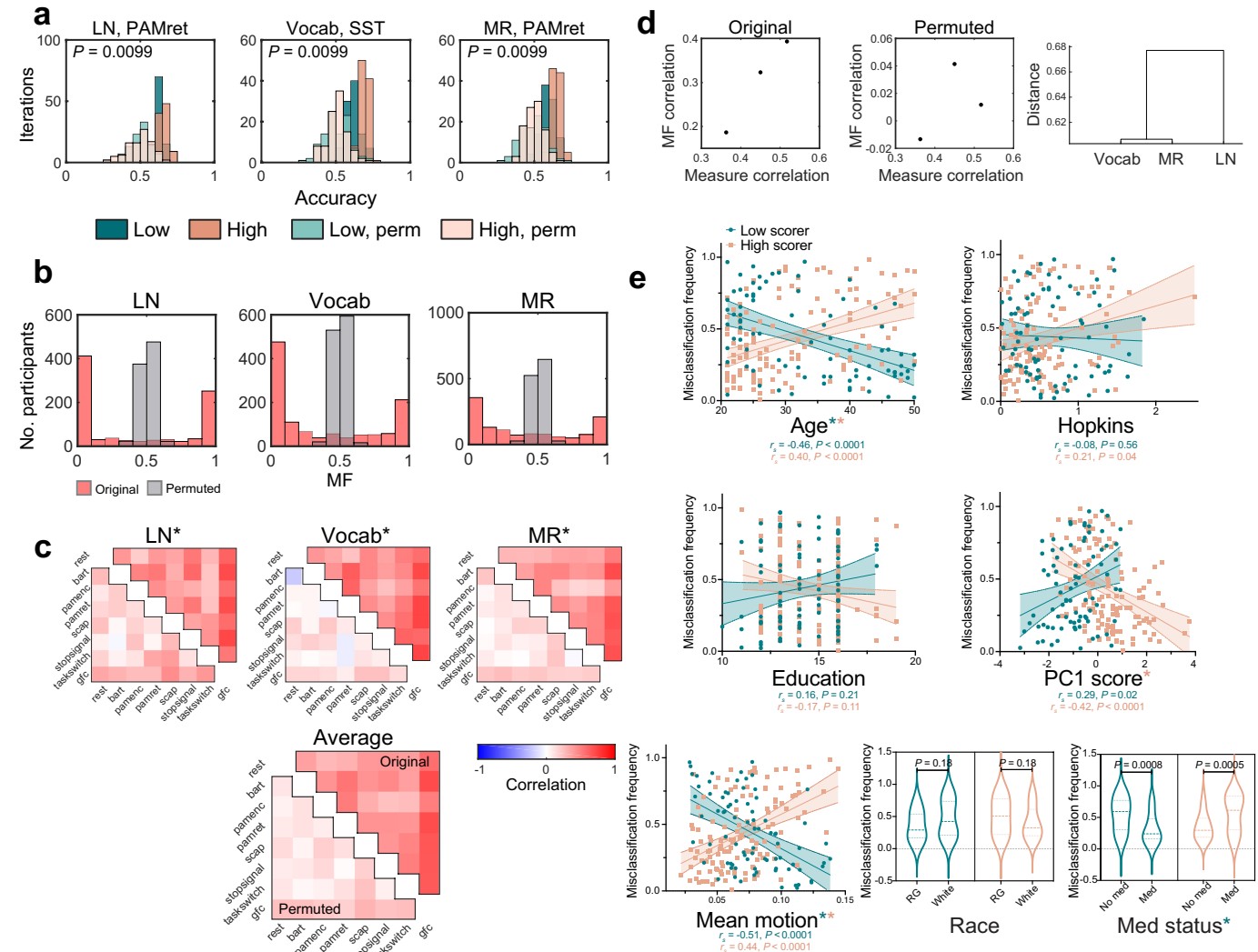

**Extended Data Fig. 2 | Replication of classification and internal validation results in the UCLA CNP dataset.** Results as presented in Fig. 1b, Fig. 2, and Fig. 4a. (a) Significance via one-tailed permutation testing (as in Fig. 1b); $P$ values FDR adjusted (3 tests). For sample sizes, see Supplementary Table 5. (b) As in Yale data, mean of permuted distribution did not significantly differ from 0.5 (all $P > 0.05$, FDR adjusted [3 tests]), mean and median of original data-based distribution significantly differed from 0.5 (all $P < 0.0001$, FDR adjusted [6 tests] via two-tailed $t$- and Wilcoxon signed-rank tests), and the misclassification frequency distributions for original and permuted analyses significantly differed for each measure (all $P < 0.0001$, FDR adjusted [3 tests] via two-tailed, two-sample Kolmogorov–Smirnov test). (c) *$P < 0.0001$, FDR adjusted (3 tests) via paired, one-tailed Wilcoxon signed-rank test (as in Fig. 2b).

(d) Given the small number of included measures, we present these results only for consistency with main analyses. As in Fig. 2c, different participants excluded for intermediate, missing, or outlier scores for each measure; number of correlated participants for each measure pair ranges from misclassification frequency: 103-138, measure: 162-163. (e) Results as presented in Fig. 4a. Covariate relationships presented if they were significantly related to misclassification frequency in low or high scorers ($P < 0.05$, adjusted), or if they were significantly related to misclassification frequency in Yale analyses (education, race) to demonstrate comparable trends. All $P$ values FDR adjusted (22 tests). For full results and relationship of covariates to mean score, as well as sample sizes, see Supplementary Tables 9 and 10. PAMret, paired associates memory task, retrieval; SST, stop signal task; MF, misclassification frequency.

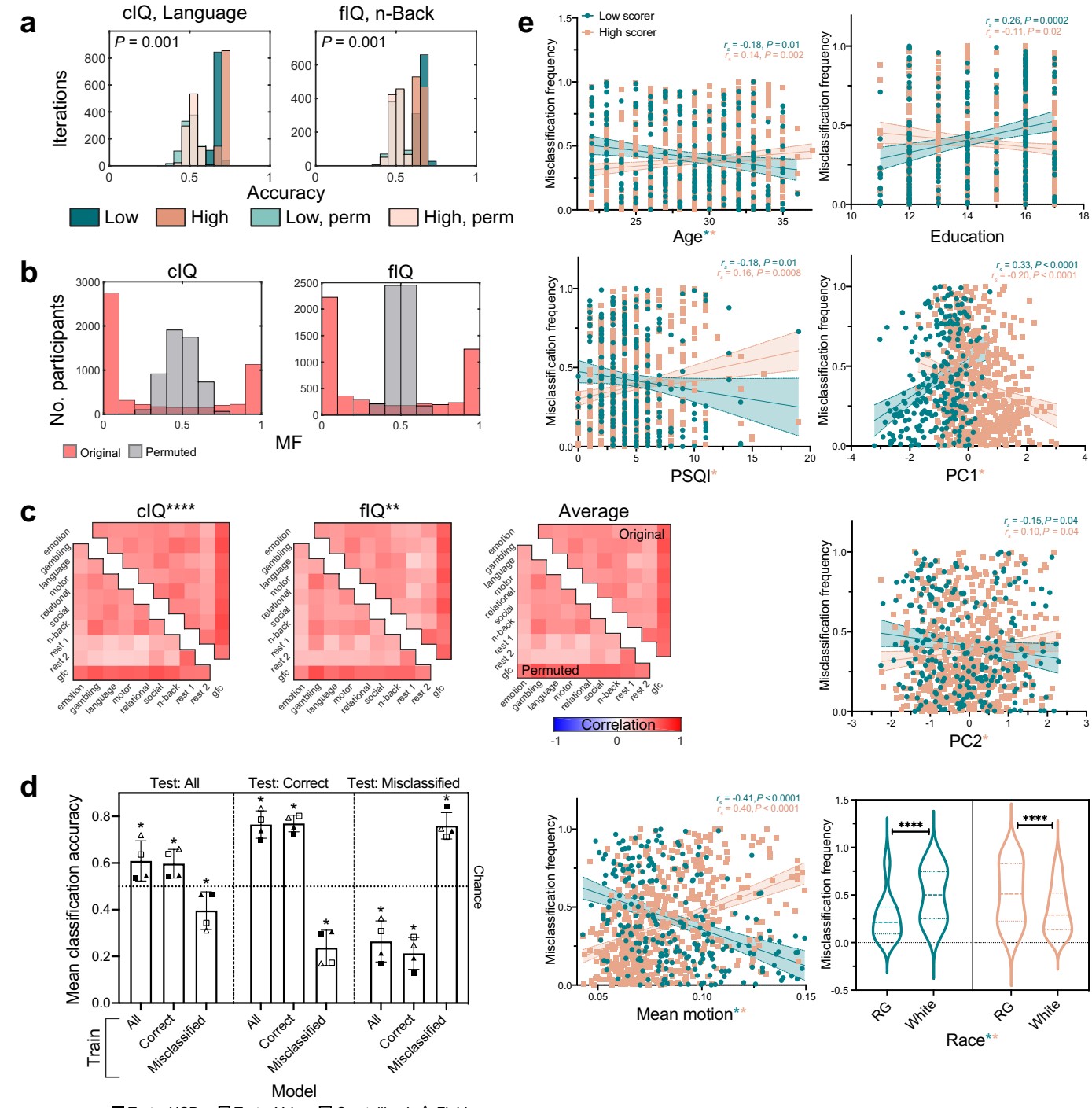

**Extended Data Fig. 3** | See next page for caption.

**Extended Data Fig. 3 | Replication of classification, internal validation and external validation results in the HCP dataset.** Results as presented in Fig. 1b, Fig. 2a, b, Fig. 3a, and Fig. 4a. Given the large HCP sample size, 10-fold cross-validation was used (20 partitions, 50 subsampling iterations each), with the requirement that family members be assigned to the same fold. Given that only two measures were classified, we omit measure versus misclassification frequency similarity and hierarchical linkage analyses. (a) Significance via one-tailed permutation testing (as in Fig. 1b); $P$ values FDR adjusted (2 tests). For sample sizes, see Supplementary Table 6. (b) Permuted distribution means significantly differed from 0.5 via two-tailed, one-sample $t$-test (cIQ mean = 0.491 [$P < 0.0001$], fIQ mean = 0.498 [$P = 0.04$], both FDR adjusted [2 tests]). All else as in Yale and UCLA analyses: mean and median of original data-based distribution significantly differed from 0.5 (all $P < 0.0001$, FDR adjusted [4 tests] via two-tailed $t$- and Wilcoxon signed-rank tests), and the misclassification frequency distributions for original and permuted analyses significantly differed for each measure (all $P < 0.0001$, FDR adjusted [2 tests] via

two-tailed, two-sample Kolmogorov–Smirnov test). MF, misclassification frequency. (c) **$P = 0.001$, ****$P < 0.0001$, FDR adjusted (2 tests) via paired, one-tailed Wilcoxon signed-rank test (as in Fig. 2b). (d) Results presented as in Fig. 3a. Bar height, grand mean; error bars, s.d. *$P < 0.0001$, FDR adjusted (9 tests) via two-tailed, nested ANOVA. For each classified measure (cIQ/vocabulary and fIQ/MR for HCP/Yale), six models were trained: 1 using all Yale participants, 1 using Yale CCP, 1 using Yale MCP (see Fig. 3 legend for training set sizes), 1 using all HCP participants (number of participants used for training after excluding intermediate and outlier scores and subsampling to balance classes: 230 and 350 for crystallized and fluid measures, respectively), 1 using HCP CCP (168, 216), and 1 using HCP MCP (62, 134). See Supplementary Tables 4 and 6 for test-set sizes. (e) Results as presented in Fig. 4a. Covariate relationships presented if they were significantly related to misclassification frequency in low or high scorers ($P < 0.05$, adjusted). For full results and relationship of covariates to mean score, as well as sample sizes, see Supplementary Tables 9 and 10. ****$P < 0.0001$; all $P$ values FDR adjusted (22 tests).

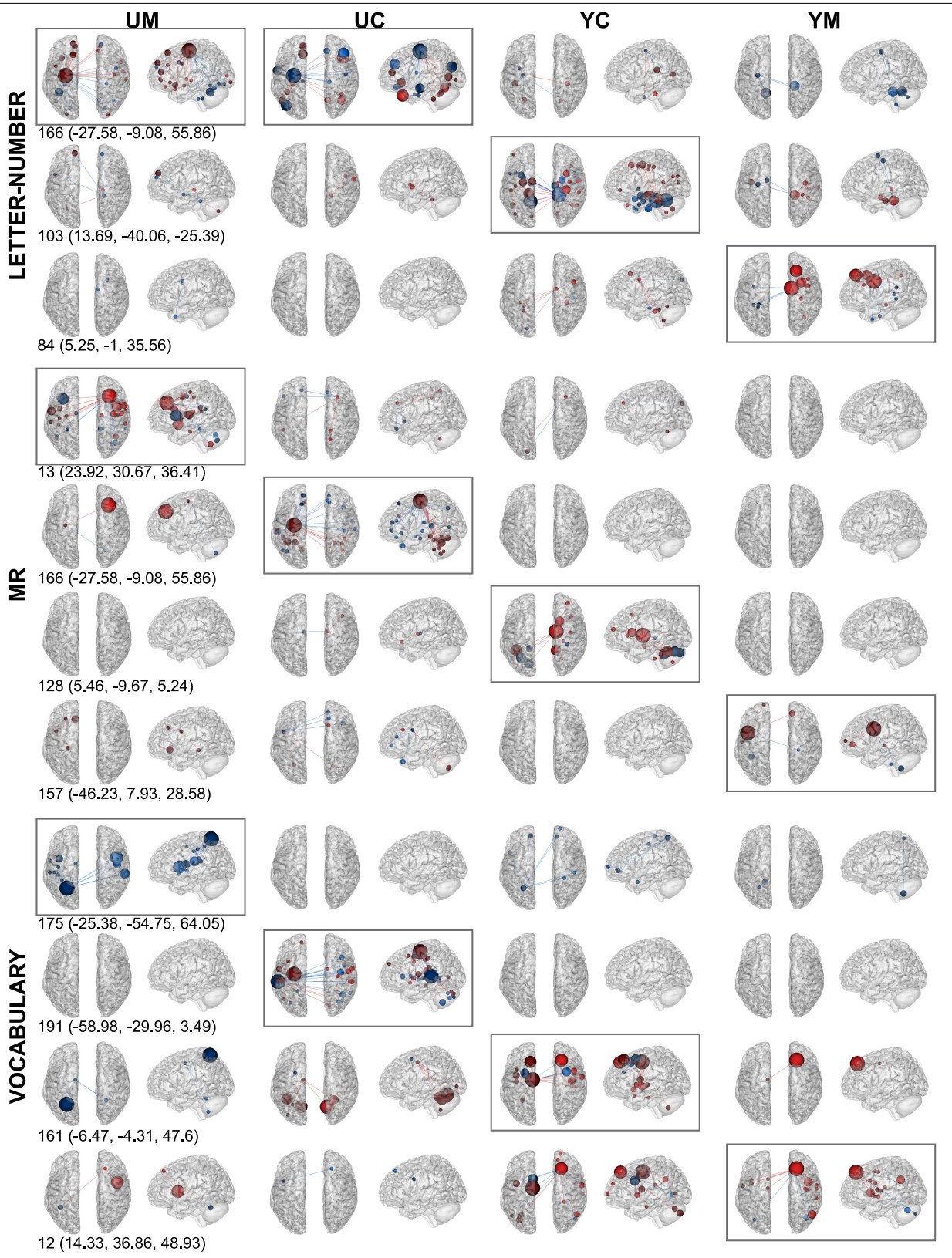

**Extended Data Fig. 4 | Selected edges for top-degree nodes in each Yale/UCLA model.** Results as presented in Fig. 3c. For MR, YM and Vocabulary, UC two nodes were tied for highest degree (MR, YM: 26 and 157; Vocabulary, UC: 166 and 191). Only one node for each model visualized for illustration.

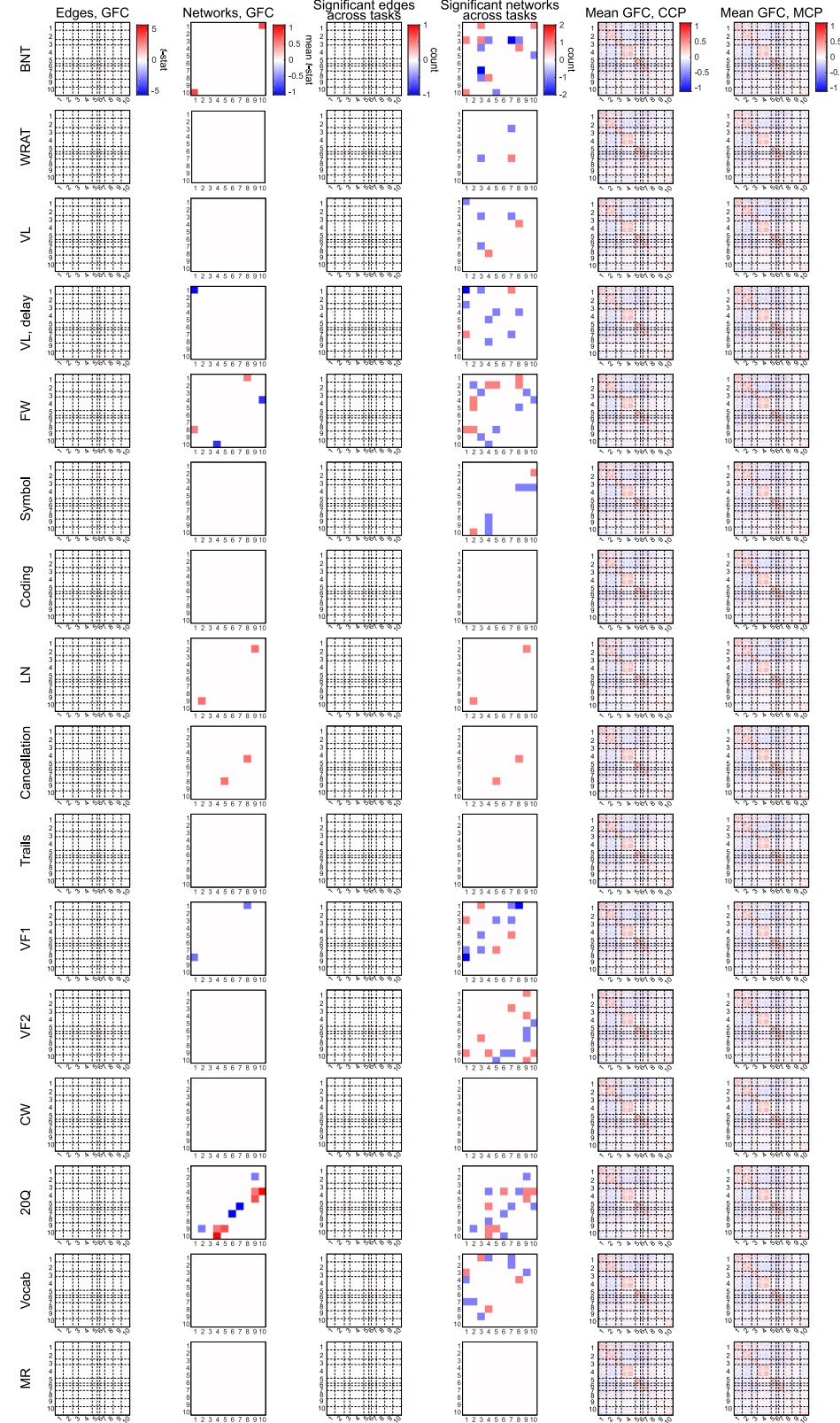

**Extended Data Fig. 5** | See next page for caption.

**Extended Data Fig. 5 | Comparison of FC between CCP and MCP groups at edge and network levels.** Edges, GFC: $t$ statistics for each GFC edge found to significantly differ (via two-sample $t$-test) between groups ($P < 0.05$, FDR adjusted), ordered by network. Red, CCP>MCP; Blue, MCP>CCP. Networks, GFC: mean $t$ statistics for each network pair (using GFC) found to significantly differ (via Constrained NBS[133]) between groups (one-tailed $P < 0.025$, FDR adjusted). Red, CCP>MCP; Blue, MCP>CCP. Significant edges across tasks and Significant networks across tasks: Number of times (i.e., tasks for which) edge (ordered by network) or network pair was significantly greater for CCP than MCP – number of times edge or network pair was significantly greater for MCP than CCP. Mean GFC, CCP and Mean GFC, MCP: GFC, averaged across participants within each group; main diagonal set to 0, and nodes ordered by network. Note that CCP and MCP groups differ for each phenotypic measure and in-scanner task (range of number of participants using GFC across phenotypic measures: CCP = 46-81, MCP = 23-63). Black dashed lines separate networks: 1 = medial frontal, 2 = frontoparietal, 3 = default mode, 4 = motor, 5 = visual A, 6 = visual B, 7 = visual association, 8 = salience, 9 = subcortical, 10 = cerebellum (for network visualization, see[140]).

**1 Pre-data collection**

- Collect expansive, inclusive, explanatory demographic data
- Increase enrollment of underrepresented groups
- Use validity, informed by psychometric literature, to guide phenotypic measure selection
- Formally evaluate risk of bias in study design

**2 Data collection**

- Use best practices for measure administration to minimize bias

**3 Data analysis**

- Consider appropriateness of norms
- Identify and correct for sample confounds, acknowledging the likely insufficiency of a unitary covariate-outcome relationship
- Characterize persistent confounds and use them to guide model interpretation

**Extended Data Fig. 6 | Future directions.** Schematic representation of recommended framework for study design and analysis to yield more precise, useful, and unbiased models.

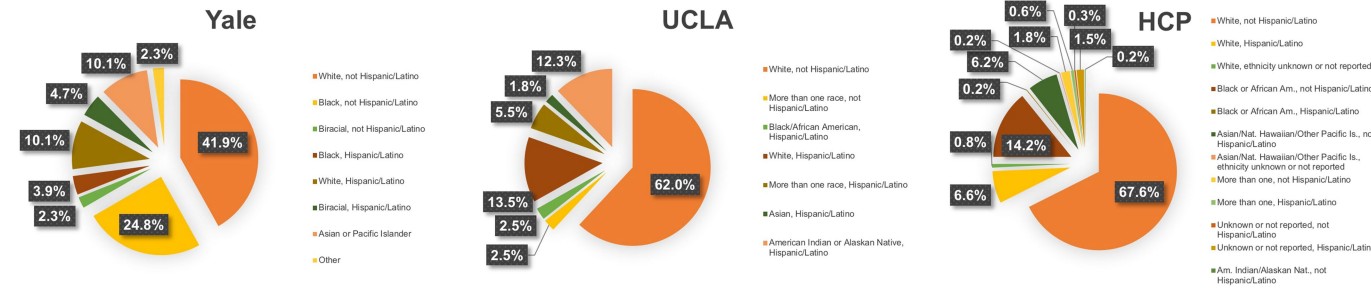

**Extended Data Fig. 7 | Race and ethnicity.** Reported racial and ethnic breakdowns of the Yale, UCLA and HCP samples.

**Extended Data Table 1 | Relationship of brain to phenotype after controlling for covariates**

| | | Score | Age | Race | Educ | Mot | Sex | Rx | PSQI |
|---|---|---|---|---|---|---|---|---|---|
| **Yale** | **LN** | n.s. | 2.7 (*P*=0.008) | 3.9 (*P*=0.0002) | 2.1 (*P*=0.038) | 3.2 (*P*=0.002) | 2.1 (*P*=0.038) | n.s. | n.s. |
| | **Vocab** | 2.0 (*P*=0.047) | 2.3 (*P*=0.024) | 3.1 (*P*=0.003) | n.s. | 2.3 (*P*=0.024) | 2.7 (*P*=0.009) | n.s. | n.s. |
| | **MR** | n.s. | 4.2 (*P*<0.0001) | 2.9 (*P*=0.004) | n.s. | 3.0 (*P*=0.003) | n.s. | n.s. | n.s. |
| **UCLA** | **LN** | 3.9 (*P*=0.0002) | 3.6 (*P*=0.0005) | n.s. | n.s. | 4.2 (*P*<0.0001) | n.s. | 2.6 (*P*=0.01) | |
| | **Vocab** | 3.0 (*P*=0.003) | 5.8 (*P*<0.0001) | n.s. | n.s. | 5.3 (*P*<0.0001) | n.s. | n.s. | |
| | **MR** | 5.8 (*P*<0.0001) | 4.4 (*P*<0.0001) | n.s. | n.s. | n.s. | n.s. | 3.8 (*P*=0.0002) | |
| **HCP** | **cIQ** | 4.9 (*P*<0.0001) | 5.3 (*P*<0.0001) | 5.6 (*P*<0.0001) | 2.8 (*P*=0.005) | 9.9 (*P*<0.0001) | n.s. | | 2.2 (*P*=0.027) |
| | **fIQ** | 4.5 (*P*<0.0001) | 4.8 (*P*<0.0001) | 6.9 (*P*<0.0001) | 3.7 (*P*=0.0002) | 9.3 (*P*<0.0001) | n.s. | | 2.8 (*P*=0.005) |

Multiple linear regression results for all models reported as |t| for parameter estimates. All covariates with coefficient *P*<0.05 for any model are reported (n.s.: *P*≥0.05; significant *P* values in parentheses, uncorrected). Educ, amount of education; Mot, in-scanner head motion; Rx, medication status; PSQI, Pittsburgh Sleep Quality Index. Outlier scores and missing data excluded, as in main analyses, but all (including intermediate) continuous scores included given application of previously developed models. Yale, *n*=94; UCLA, *n*=162-163; HCP, *n*=641–646.

# Reporting Summary

## Statistics

For all statistical analyses, confirm that the following items are present in the figure legend, table legend, main text, or Methods section.

| n/a | Confirmed | |
|---|---|---|
| ☐ | ☒ | The exact sample size (*n*) for each experimental group/condition, given as a discrete number and unit of measurement |
| ☐ | ☒ | A statement on whether measurements were taken from distinct samples or whether the same sample was measured repeatedly |
| ☐ | ☒ | The statistical test(s) used AND whether they are one- or two-sided<br>*Only common tests should be described solely by name; describe more complex techniques in the Methods section.* |
| ☐ | ☒ | A description of all covariates tested |
| ☐ | ☒ | A description of any assumptions or corrections, such as tests of normality and adjustment for multiple comparisons |
| ☐ | ☒ | A full description of the statistical parameters including central tendency (e.g. means) or other basic estimates (e.g. regression coefficient) AND variation (e.g. standard deviation) or associated estimates of uncertainty (e.g. confidence intervals) |
| ☐ | ☒ | For null hypothesis testing, the test statistic (e.g. *F*, *t*, *r*) with confidence intervals, effect sizes, degrees of freedom and *P* value noted<br>*Give P values as exact values whenever suitable.* |
| ☒ | ☐ | For Bayesian analysis, information on the choice of priors and Markov chain Monte Carlo settings |
| ☒ | ☐ | For hierarchical and complex designs, identification of the appropriate level for tests and full reporting of outcomes |
| ☐ | ☒ | Estimates of effect sizes (e.g. Cohen's *d*, Pearson's *r*), indicating how they were calculated |

*Our web collection on statistics for biologists contains articles on many of the points above.*

## Software and code

Policy information about availability of computer code

| | |
|---|---|
| Data collection | In-scanner tasks were presented using Psychtoolbox-3 running in Matlab (Mathworks; 2017b or later) |
| Data analysis | Yale and UCLA data were preprocessed using FSL, SPM, and BioImage Suite. HCP data were preprocessed using the HCP minimal preprocessing pipeline and BioImage Suite. Data analysis and result visualization were performed using BioImage Suite; Matlab (Mathworks) 2017a, 2018a, and 2021b; GraphPad Prism version 9.0.1; and R version 3.6.0 for macOS. MATLAB code to run classification, validation, and covariate analyses is available at https://github.com/abigailsgreene/modelFailure/. |

For manuscripts utilizing custom algorithms or software that are central to the research but not yet described in published literature, software must be made available to editors and reviewers. We strongly encourage code deposition in a community repository (e.g. GitHub). See the Nature Portfolio guidelines for submitting code & software for further information.

## Data

Policy information about availability of data

All manuscripts must include a data availability statement. This statement should provide the following information, where applicable:

- Accession codes, unique identifiers, or web links for publicly available datasets
- A description of any restrictions on data availability
- For clinical datasets or third party data, please ensure that the statement adheres to our policy

The primary dataset used in these analyses represents the first set of participants in an ongoing study. Data will be released in waves via the NIMH Data Archive, Collection 3276. The UCLA CNP data can be obtained from the OpenfMRI database (https://openfmri.org/dataset/ds000030/), accession number ds000030. The HCP data are publicly available on the ConnectomeDB database (https://db.humanconnectome.org/app/template/Login.vm). Data used to generate the parcellation can be found at http://fcon_1000.projects.nitrc.org/indi/retro/yale_hires.html.

# Field-specific reporting

Please select the one below that is the best fit for your research. If you are not sure, read the appropriate sections before making your selection.

☒ Life sciences ☐ Behavioural & social sciences ☐ Ecological, evolutionary & environmental sciences

For a reference copy of the document with all sections, see nature.com/documents/nr-reporting-summary-flat.pdf

# Life sciences study design

All studies must disclose on these points even when the disclosure is negative.

| | |
|---|---|
| Sample size | The Yale sample includes 129 participants, the UCLA sample includes 163 participants, and the HCP sample includes 664 participants after exclusions. These samples are of comparable size to or larger than samples used in previously published brain-based predictive modeling work (e.g., Finn, Shen, et al., 2015, Nature Neuroscience). |
| Data exclusions | Participants were excluded for excessive in-scanner head motion (pre-determined thresholds), missing data, and imaging coverage, anomalies, or other data quality issues. |
| Replication | The UCLA CNP and HCP datasets were used as external validation datasets, and all main results were replicated in them. Some differences in covariates that track misclassification were observed across datasets, reflecting the likely sample dependence of score stereotypes (see Discussion). |
| Randomization | N/A; there were no experimental groups in the study. |
| Blinding | N/A; there was no group allocation in the study. |

# Reporting for specific materials, systems and methods

We require information from authors about some types of materials, experimental systems and methods used in many studies. Here, indicate whether each material, system or method listed is relevant to your study. If you are not sure if a list item applies to your research, read the appropriate section before selecting a response.

## Materials & experimental systems

| n/a | Involved in the study |
|---|---|
| ☒ | Antibodies |
| ☒ | Eukaryotic cell lines |
| ☒ | Palaeontology and archaeology |
| ☒ | Animals and other organisms |
| ☐ | ☒ Human research participants |
| ☒ | Clinical data |
| ☒ | Dual use research of concern |

## Methods

| n/a | Involved in the study |
|---|---|
| ☒ | ChIP-seq |
| ☒ | Flow cytometry |
| ☐ | ☒ MRI-based neuroimaging |

# Human research participants

Policy information about studies involving human research participants

| | |
|---|---|
| Population characteristics | See Supplementary Table 3 and Extended Data Figure 7 for detailed demographic information for the Yale, UCLA, and HCP datasets. |
| Recruitment | The Yale sample was recruited via community advertisements and recruitment from several Yale clinics with which collaborations were established. UCLA participants were recruited via community advertisements, outreach to clinics, and online portals. HCP participants were recruited broadly from families containing twins in Missouri. |
| Ethics oversight | The protocol for the Yale study was approved by the Yale Institutional Review Board. The UCLA CNP study was approved by the Institutional Review Boards at UCLA and the Los Angeles County Department of Mental Health. The HCP study was approved by the Institutional Review Board at Washington University in St. Louis. |

Note that full information on the approval of the study protocol must also be provided in the manuscript.

# Magnetic resonance imaging

## Experimental design

| | |
|---|---|
| Design type | Both resting-state and task fMRI data were used. |
| Design specifications | Tasks used in each study have been described in previously published work (main text reference 78). For the purposes of this work, task design was not relevant, as functional connectivity was calculated from continuous BOLD timecourses regardless of task design. |
| Behavioral performance measures | In-scanner behavioral performance measures were not used for this work. |

## Acquisition

| | |
|---|---|
| Imaging type(s) | Functional and structural imaging was performed |
| Field strength | 3T |
| Sequence & imaging parameters | Yale: MPRAGE (208 slices acquired in the sagittal plane, repetition time (TR) = 2400 ms, echo time (TE) = 1.22 ms, flip angle = 8deg, slice thickness = 1 mm, in-plane resolution = 1 mm x 1 mm). EPI (75 slices acquired in the axial-oblique plane parallel to the AC-PC line, TR = 1000 ms, TE = 30 ms, flip angle = 55deg, slice thickness = 2 mm, multi-band acceleration factor = 5, in-plane resolution = 2 mm x 2 mm). UCLA: MPRAGE (176 slices acquired in the sagittal plane, TR = 1900 ms, TE = 2.26 ms, slice thickness = 1 mm, FOV = 250 mm, matrix size = 256 x 256). EPI (34 slices acquired in the oblique plane, TR = 2000 ms, TE = 30 ms, flip angle = 90deg, slice thickness = 4 mm, FOV = 192 mm, matrix size = 64 x 64). HCP: MPRAGE (256 slices acquired in the sagittal plane, TR = 2400 ms, TE = 2.14 ms, flip angle = 8deg, slice thickness = 0.7 mm, in-plane resolution = 0.7 mm x 0.7 mm). EPI (72 slices acquired in the axial-oblique plane, TR=720 ms, TE=33.1ms, flip angle=52deg, slice thickness = 2 mm, in-plane resolution = 2 mm x 2 mm, multiband factor=8). |
| Area of acquisition | Whole brain scans were acquired. |
| Diffusion MRI | ☐ Used    ☒ Not used |

## Preprocessing

| | |
|---|---|
| Preprocessing software | Structural scans were skull stripped using an optimized version of the FMRIB's Software Library (FSL) pipeline (optiBET). Motion correction was performed in SPM12. Nonlinear registration of the MPRAGE to the MNI template was performed in BioImage Suite, and linear registration of the functional to the structural images was performed using a combination of FSL and BioImage Suite linear registration tools to optimize registration quality. All additional preprocessing steps were performed in BioImage Suite, and included regression of mean time courses in white matter, cerebrospinal fluid, and gray matter; high-pass filtering to correct linear, quadratic, and cubic drift; regression of 24 motion parameters; and low-pass filtering (Gaussian filter, sigma=1.55). Preprocessing of the UCLA and HCP data was performed similarly, with minor modifications (including the use of the HCP minimal preprocessing pipeline; see Methods). |
| Normalization | Both nonlinear and linear registrations were performed (nonlinear registration of the MPRAGE to the MNI template, and linear registration of the functional data to the MPRAGE). |
| Normalization template | The MNI template was used for normalization. |
| Noise and artifact removal | See above. Additional confound analyses to control for motion are described in the Methods, with results reported in Supplementary Fig. 1 and Table 7. |
| Volume censoring | Censoring was not performed. |

## Statistical modeling & inference

| | |
|---|---|
| Model type and settings | Main analyses involved functional connectivity-based classification of binarized phenotypic scores. As described in the text (Methods: Phenotype classification), classification was performed in a leave-one-out cross-validated fashion, with the training set subsampled from non-test subjects to ensure balanced classes. Edges were selected based on their correlation with the modeled outcome, and summed to yield summary statistics that were normalized and submitted to a linear support vector machine. This process was repeated iteratively, with each subject held out, and this, in turn, was performed 100 times with unique training set subsampling. The entire analysis was repeated for every combination of in-scanner condition and phenotypic measure. Classification outcome and phenotypic scores were related to a range of covariates in a pairwise fashion (two-tailed Mann-Whitney U tests for binary covariates and rank correlation for continuous covariates), as well as via regression. Modifications of this analysis demonstrate the robustness of results to classification algorithm, phenotype binarization, parcellation, and cross-validation approach (see Methods, Extended Data Fig. 1, and Supplementary Table 11). One-tailed tests were only used in the case of a-priori directional hypotheses. |
| Effect(s) tested | The main effect of interest was whether individuals were correctly classified as high- or low-scoring. |
| Specify type of analysis: | ☒ Whole brain    ☐ ROI-based    ☐ Both |

| Statistic type for inference<br>(See <u>Eklund et al. 2016</u>) | Voxel-wise analyses were not performed. The constrained network-based statistic was used for comparison of FC at the network level across correctly classified and misclassified groups (Extended Data Fig. 5, Methods). |
|---|---|
| Correction | P values were FDR adjusted for multiple comparisons, where relevant and as noted throughout the text. |

## Models & analysis

| n/a | Involved in the study |
|---|---|
| ☐ | ☒ Functional and/or effective connectivity |
| ☐ | ☒ Graph analysis |
| ☐ | ☒ Multivariate modeling or predictive analysis |

| Functional and/or effective connectivity | Functional connectivity was measured via Fisher-transformed Pearson correlation. |
|---|---|
| Graph analysis | Binarized node degree was used for model visualizations in Fig. 3 and Extended Data Fig. 4. |
| Multivariate modeling and predictive analysis | In main analyses, edges were selected based on their correlation with the outcome (in the training set), and were summed to yield model features, which were normalized and submitted to a linear SVM to classify neurocognitive score (see above and Methods: Phenotype classification). In supplementary analyses, edges were used as features and submitted to an alternative classification algorithm (bagging or neural network; Methods). Classification accuracy was used to evaluate models, with significance evaluated via permutation tests. |

