## [Peer Review File · Nature]

Manuscript Title: Brain-phenotype models fail for individuals who defy sample stereotypes

Reviewer Comments & Author Rebuttals

Reviewer Reports on the Initial Version:

Referees' comments:

Referee #1 (Remarks to the Author):

1. Summary

Understanding brain-behavior relationships and using brain imaging metrics to predict behavior are common goals within cognitive neuroscience. However, though models are generalizable, they fail in subgroups of individuals. In the current work of Green et al ask the question "For what people do these models fail and why". To understand the relationship between brain and behavior, individual differences between functional connectivity measures and behavior are computed. The authors use a model assuming that the relationship between brain and behavior has a similar direction and mechanism across all individuals. However, when using this data-driven approach (linear support vector classification) the authors observed that there were consistent outliers when predicting neurocognitive measures in two independent samples. These outliers showed mostly inverse brain-behavior associations in the model, and post-hoc tests suggest that demographic factors such as age, motion, cognitive ability but also 'race' (white/non-white).

2. Originality and significance

As the results stand now I am not sure what the key findings are and how to interpret them with respect to the rest of the literature. Other approaches such as normative modelling or multivariate approaches may be better suited to study the role of individual differences and stereo/archetypes in brain-behavior relationship, rather than taking a prediction framework. Alternatively, the prediction framework is expanded to include also alternative models and parameter set ups may help to further understand whether the observed bias is a result of the computational approach or biological archetypes. Previous work that may be relevant is <https://doi.org/10.1016/j.neuroimage.2017.06.061> and <https://doi.org/10.1016/j.neuroimage.2018.06>. But also, more general for example the work studying bias in AI research on facial recognition (<https://arxiv.org/abs/1906.06439>).

3. Data and methodology

Overall, though it is an interesting question, I would expect a more thorough comparison of different models and approaches as well as sample composition to justify the conclusions. Moreover I believe the sample size in the current work is mostest, and it is unclear if some conclusions are not due to relatively small sample size. Last, contributors to misclassification seems to be a big point, but how were they selected, how independent they are is unclear. For example, I can imagine that self-reported stress and sleep disturbance have some correlation. It would help to further understand the interdependence of these factors and interplay with misclassifications.

4. Statistics and treatment of uncertainties

All the data is presented in a good way. However, as also written earlier in the review, I am not sure whether the prediction approach taken could not be compared to alternative approaches to be sure the misclassification is not accounted for by the approach. Specifically; 1. Though I understand that the key goal was to investigate why models fail for certain individuals, I am unsure why only one particular model was explored had various specific features that may have contributed to misclassification such as 1) binarization and 2) the approach of leave-one-out cross validation, but also 3) imaging parameters. First, binarization may lead to overinterpretation of differences and lacking training on subtle relationships between brain and behavior that are better caught by an individual difference approach that uses the categorical information. Second, though LOOCV is often used in the literature and has its value, it also has downsides with respect to variance and bias (<https://stats.stackexchange.com/questions/61783/bias-and-variance-in-leave-one-out-vs-k-fold-cross-validation>). This leads me to wonder about the generalizability of the findings. Third, also other options with respect to imaging markers (parcellation, summary of data) seem to be under explored, raising also here the question about generalisability.

E. Conclusions

Though in general I believe the conclusion makes sense, I am not sure if the alternative hypotheses about misclassification are tested thoroughly enough to really provide novel insights about the nature of misclassifications. Indeed, site-effects or difference between gender, or even ethnicity are often considered to be important factors in biopsychological research, so it seems a little bit of a too general conclusion to be wrong. In particular the notion of individuals 'who defy stereotypes' and relation to race (ethnicity?), education, and other demographic factors is tricky and not sufficiently addressed in the current work.

F. Suggested improvements

It depends a bit on the preferred scope of the work, but I would suggest to compare the current model to alternative models and more formally test misclassifications and also archetypes of certain datasets. Moreover, I would more carefully explain the selected variables that are thought to explain misclassification. Possibly, I'd also account of structure in the functional models, as this may also be of influence. Other comments, such as the question about the sample size, are also earlier in the review.

G. References

Though the references are appropriate. Possibly this work is also interesting: <https://www.sciencedirect.com/science/article/pii/S1053811918305081?via=ihub>. Possibly it may help to discuss 'ethical AI' in a broadly, as also suggested earlier.

H. Clarity

Overall the manuscript is clearly written however it seems like the question is broader than the introduction, approach, and discussion support.

I hope my comments are helpful.

Referee #2 (Remarks to the Author):

The study evaluated the generalizability of brain-phenotype prediction, and observed that a subset of individuals is consistently misclassified using a one-size-fits-all approach. Findings were observed across two different datasets.

These potentially important results argue for a more nuanced modelling of brain-phenotype relationships, which in particular speak against common one-size-fits-all approaches. The analyses are thoughtful and comprehensive. I would nevertheless have a series of suggestions to further strengthen the analytical/predictive learning component of the work, in order to solidify the main conclusions of the paper.

1) To enhance readability, the authors could provide more details on the datasets evaluated already in the main manuscript e.g., number of subjects, sites, age, healthy/diagnosed etc for both the Yale and UCLA CNP data. This could be presented in line 112 but also in part in the abstract/end of introduction already in my opinion.

2) The schema on figure 1 suggests that LOO cross validation was used for classifier training and performance evaluation. LOO may have limitations in testing classifier generalizability, and one may thus want to know whether results are consistent when a more conservative (e.g. 5 fold) cross validation scheme was used. Moreover, did the authors perform a nested cross validation in the training dataset for feature selection/validation (which one may recommend) prior to testing the performance in the outermost held out fold? In that respect, can the authors

3) The authors mention that linear SVMs were used to classify individuals as high/low scorers based on FC patterns. Given that many of the phenotypic scores are continuous, wouldn't a regression SVM framework be more adequate, as it may avoid unnecessary subsplits of the phenotypic variables and one would also not need to remove datapoints with inter-mediatry scores? Ideally, the authors could show consistency of their findings when using regression SVM, and could then assess residuals for the prediction instead of correct/incorrect classification results.

4) Have analytical strategies been used for cross-dataset homogenization e.g. linear regression and/or combat, for both the FC and/or cognitive phenotypes. If not, could the authors comment on the potential utility of such approaches and discuss whether their current findings might be affected or invariant to those (could also be a supplementary analysis).

5) Both the Yale and the UCLA sample include "clinically naturalistic and demographically diverse participants". While this transdiagnostic inclusion is great, would it be possible to show that overall findings hold even if only 'healthy' individuals without concurrent diagnosis were included, to show generalizability of the results to a purely non-clinical sample?

Minor:

I.330 - please report the corresponding test statistics, in addition to the significant p-values

I.393 - First sentence of the discussion is a bit odd. It might be worthwhile if the authors underscored the significance of brain-behavior prediction approaches a bit more specifically. Why is such an endeavour important?

I. 717 - please clarify whether the initial threshold is corrected or uncorrected.

Please provide the Matlab version in the methods

Referee #3 (Remarks to the Author):

The authors investigate the failure modes of a 2-dimensional linear support vector machine classifier for prediction of binarized behavioral variables with fMRI functional connectivity (FC) data. They use 9 different sources of FC data (two rest sessions and 7 tasks) and 16 different behavioral phenotypes, a 129 subject primary dataset and 163 subject validation dataset. For each behavioral variable they discard roughly 15% of subjects near the mean (within +/- 1/3 SD of mean) as well as lower (but not upper) outliers, before dichotomizing with the sign operator. They find that some subjects tend to be misclassified consistently over different types of FC data, and misclassification rates are correlated between different behavioral phenotypes.

I find the work overstated and suffering from at least one critical analytical flaw. The findings overstate the impact of results from one single and very simplistic prediction model, when the neuroimaging literature as a whole uses a vast array of different models that are all much more complex than this approach. The analytical flaw relates to the use of permutation techniques that appear to randomly shuffle data independently, when instead synchronized permutation is needed to ensure that the expected (within subject FC) correlation is accounted for.

Overall, I see this this work as a deep dive into the limitations of the 2D linear SVM approach to FC prediction. It is not an unuseful investigation, but I don't see how such a narrow methodological study would merit the attention of the readers of this journal.

Major Issues

The predictive method used here (independent selection of FC edges to compose two feature vectors based on selected edges, fed into a linear SVM) is but one method out of a constellation of machine learning methods available to conduct this exercise. This reviewer is not hinting that some sophisticated deep learning must be deployed. Schulz et al. (2020) have nicely shown that brain data don't benefit from the super-high dimensionality of DL methods, but best practice (as illustrated by Schulz et al) is to let the data dictate the dimensionality required... and in this reviewer's experience the data generally rarely indicate just 2 dimensions are needed.

With more dimensions, (or if fixed on 2D) nonlinear SVM, or ensemble methods that combine a battery of weak learners, the authors would likely find fewer and perhaps an unexceptional proportion of poorly predicted subjects. Thus, this work is severely limited by its use of this one very simplistic model.

The calculation of significance for 'Misclassification is generic to FC type' (Fig 2b) is using the wrong null hypothesis. It is inferred that null task-by-task correlations are obtained by correlating MF from independently shuffling the +/-1 label for each FC. This corresponds to a null hypothesis that the FC and the phenotype are independent *and* the different FC data are independent. Clearly, the same subjects give rise to the 9 FC measures and thus are dependent. Instead, the authors want to test a null of just "FC and the phenotype are independent" while preserving the dependence structure among the FC measures. Put another way, the lower diagonal correlations in Figure 2b must reflect the intrasubject dependence of the imaging measures. This is done by ensuring that the null MF correlations arise from the same permutation of phenotype labels for all imaging measures correlated (this may require a smaller dataset to ensure a common set of subjects for all 9 imaging measures; alternatively, the synchronized permutation could be done in a pairwise fashion, for each pair of the 9 measures).

It is likely that with use of synchronized permutation the null correlations will look more like the observed correlations, weakening or undermining the conclusions drawn from Fig 2b.

Other Serious Concerns

I find the filtering of the data troubling, i.e. the elimination of the central 2/3 SD of the subjects. I was expecting at least a sub-analysis to demonstrate what happens when no middling subjects are dropped. Or a comment on at least how the authors anticipate the results to change as this threshold is shrunk or expanded.

It seems arbitrary and worrying that only low scorers ($< -3sd$) were removed and not high scores ($> 3sd$). While this is clearly described no justification is given, nor any sensitivity analysis provided.

"FC from the best-performing condition for each measure classified scores on 14/16 phenotypic measures with above-chance performance." This is a crucial statement of results that clouds the actual multiplicity involved: I compute that there were $16 \times 9 = 144$ tests computed; was FDR computed over this set of 144 tests? Without further clarification, showing the best in each suggests the results are overly optimistic. While the accuracies are provided in Table S3 there are no p-values. This concern would be addressed if Table S3 was augmented by the uncorrected p-values for each combination of 16 measures and 8 FCs, along with the FDR p-value threshold in the caption so readers can judge the pattern and strength of effects.

Minor Points

For the breadth of the claims made, the scale of these datasets are disappointing (129 primary, 163 validation). Recent work suggests that both association and prediction cognitive outcomes with brain phenotypes requires at least 2000 subjects to have respectable out of sample performance (Marek et al., 2020; see results on prediction, Fig 3). That said, the authors are to be commended on collecting a demographically and clinically diverse sample.

Results state there were 16 non-imaging phenotypic measures considered, but Table S2 lists 17... while a reader can cross-reference figures and captions to deduce the 16 it would simply better of Table S2 clearly indicated the 16 measures used.

While the referenced GFC method is explicit about removing task variation explained by a usual task model, no mention of this is given about the treatment of the individual task data. It needs to be made clear whether the task data was used raw (after preprocessing) or after residualizing with respect to the task regressors.

Edges were selected on the basis of an intersubject correlation between edge fMRI data and -1/+1 phenotypic labels, significant at $P < 0.05$. However, correlation p-values are based on a bivariate Gaussian assumption that has no justification when X's are binary. If authors want to appeal to the meaningfulness of the 0.05 threshold (or more importantly the comparability with other work using continuous labels) they should replace the correlation with a two-sample t-test.

FDR correction was used a different points in this work, but it is not clear the family of tests to which it was applied. For example, at line 740 it isn't clear what set of p-values is submitted to FDR; in Figure 1, it isn't clear if $16 \times 9 = 144$ p-values were submitted to FDR or just 14. To avoid confusion, clearly state the set and count of p-value submitted to correction.

References

Marek et al. (2020). Towards Reproducible Brain-Wide Association Studies Affiliations. *BioRxiv*, 11, 15–18. <https://doi.org/10.1101/2020.08.21.257758>

Schulz et al (2020). Different scaling of linear models and deep learning in UKBiobank brain images versus machine-learning datasets. *Nature Communications*, 11(1), 4238. <https://doi.org/10.1038/s41467-020-18037-z>

Author Rebuttals to Initial Comments:

We thank the reviewers for their thoughtful comments. We believe that the additional analyses we have performed to address them have substantially strengthened the manuscript, demonstrating that results are robust to classification algorithm, cross-validation approach, parcellation, and phenotype binarization, and not simply the result of small sample sizes (as demonstrated through the addition of a third, much larger dataset comprised of healthy participants). These and other, more minor, revisions are described in greater detail below, with line numbers ('l.##') indicated as relevant.

Referees' comments:

Referee #1 (Remarks to the Author):

1. Summary

Understanding brain-behavior relationships and using brain imaging metrics to predict behavior are common goals within cognitive neuroscience. However, though models are generalizable, they fail in subgroups of individuals. In the current work of Green et al ask the question "For what people do these models fail and why". To understand the relationship between brain and behavior, individual differences between functional connectivity measures and behavior are computed. The authors use a model assuming that the relationship between brain and behavior has a similar direction and mechanism across all individuals. However, when using this data-driven approach (linear support vector classification) the authors observed that there were consisted outliers when predicting neurocognitive measures in two independent samples. These outliers showed mostly inverse brain-behavior associations in the model, and post-hoc tests suggest that demographic factors such as age, motion, cognitive ability but also 'race' (white/non-white).

2. Originality and significance

As the results stand now I am not sure what the key findings are and how to interpret them with respect to the rest of the literature. Other approaches such as normative modelling or multivariate approaches may be better suited to study the role of individual differences and stereo/archetypes in brain-behavior relationship, rather than taking a prediction framework. Alternatively, the prediction framework is expanded to include also alternative models and parameter set ups may help to further understand whether the observed bias is a result of the computational approach or biological archetypes. Previous work that may be relevant is <https://doi.org/10.1016/j.neuroimage.2017.06.061> and <https://doi.org/10.1016/j.neuroimage.2018.06>. But also, more general for example the work studying bias in AI research on facial recognition (<https://arxiv.org/abs/1906.06439>).

We thank the reviewer for pointing out that key findings could be clearer throughout the manuscript. We have substantially revised the introduction and discussion, and added several summary statements throughout the results section, to clarify key points. In particular, the key findings are:

- *There is no single brain-phenotype relationship that fits all individuals.*
- *Models fail systematically and consistently—across related measures and even independent datasets.*

- *Such model failure is particularly frequent in individuals who defy stereotypical profiles, profiles that models are detecting and using despite not having direct access to them.*
- *These results demonstrate that models are not predicting unitary neurocognitive constructs, but rather neurocognitive constructs intertwined with complex, sample-specific sociodemographic profiles.*
- *These profiles have the potential to limit model generalizability and propagate outcome biases into corresponding brain-based models. Acknowledging and accounting for such bias is crucial to develop useful brain-phenotype models, and to avoid misleading and potentially harmful model interpretation.*

To expand the prediction framework, we have conducted several additional analyses:

1. *We have employed an alternative classification algorithm, using an ensemble of weak learners with hyperparameters selected via cross-validated optimization, as described in Methods, 1.842-850:*

“To ensure that results are robust to classification algorithm, we repeated these analyses using an ensemble of weak learners. Specifically, we used a subset of phenotypic measures for hyperparameter optimization (ensemble aggregation method, number of learners, learning rate [where relevant], and minimum leaf size). We used all available high and low scorers (and their GFC) in a leave-one-out fashion for each optimization analysis, and used the consensus hyperparameters (method: bagging, number of learners = 150, minimum leaf size = 1) to classify all 16 phenotypic measures as in main analyses, using all selected edges (correlation $P < 0.05$, as in main analyses) as features. See Supplementary Fig. 1 for results.”

Classification accuracy and participant misclassification frequency were largely unchanged.

2. *To demonstrate that systematic misclassification was not simply the result of our selected method of cross-validation, given the bias-variance trade-off that this reviewer notes in comment 4, we have replicated main results using 10-fold, rather than leave-one-out, cross-validation (Supplementary Fig. 1 and Methods: Phenotype classification), and have used 10-fold cross-validation in HCP analyses (see comment 2 response, point 5). Classification accuracy and participant misclassification frequency were largely unchanged.*

3. *To demonstrate that systematic misclassification was not simply the result of the phenotypic binarization method, we have replicated main results using an alternative phenotypic binarization threshold (Methods 1.824-826 and Supplementary Fig. 1):*

“To ensure that results are robust to this thresholding choice, we repeated analyses using the normative mean as the cutoff (i.e., high if $>$ mean, low if $<$ mean). For results, see Supplementary Fig. 1 (“Phenotype mean split”).”

Classification accuracy and participant misclassification frequency were largely unchanged. For additional explanation of the binarization choice, see our response to Reviewer 1’s comment 4.

Further, we have replicated results using continuous prediction of all 16 phenotypic measures in the Yale dataset, and demonstrate that residualized model fit (i.e., difference between predicted and observed phenotypic scores) is related to the same covariates

that track misclassification frequency (Supplementary Table 11 [reproduced below]; Methods: Phenotype classification [1.874-882]).

Relationship to mean residualized prediction deviation	
Sex	$P > 0.1$
Age	$r_s = -0.38, P_{FDR} < 0.0001$
Race	median _{NW-W} = -0.78, $P_{FDR} = 0.0005$
Symptom severity	$P > 0.1$
Education	$r_s = 0.29, P_{FDR} = 0.003$
Stress	$P > 0.1$
PSQI	$P > 0.1$
Positive affect	$P > 0.1$
Negative affect	$P > 0.1$
Rx	$P > 0.1$
Dx	$P > 0.1$
PC1 score	$r_s = 0.30, P_{FDR} = 0.007$
PC2 score	$r_s = -0.20, P_{FDR} = 0.11$
PC3 score	$r_s = -0.17, P = 0.09$
Mean motion	$r_s = -0.39, P_{FDR} = 0.0001$

Supplementary Table 11. Relationship between continuous model fit and covariates in the Yale dataset. In this analysis, only low outliers and missing data, not intermediate scores, were excluded. Continuous phenotypic measures were predicted using FC (see Methods), and the difference between predicted and observed phenotypic score was regressed on phenotypic score. This residualized prediction deviation was averaged across in-scanner conditions, and then z-scored and averaged across phenotypic measures, yielding a single measure of model fit per participant. These mean residualized prediction deviation metrics were then related, via rank correlation for continuous covariates and Mann-Whitney U test for binary covariates, to the same covariates presented in Supplementary Table 9. As in Supplementary Table 9, FDR-adjusted P values presented for unadjusted $P < 0.05$; test statistics presented for all unadjusted $P < 0.1$; all else in gray.

4. *To demonstrate that systematic misclassification was not the result of the selected parcellation, we have replicated main results using an alternative, 368-node parcellation, described in Methods: Functional parcellation and network definition (1.809-813):*

“To ensure that results are robust to parcellation choice, we repeated analyses using the Shen 368-node atlas, derived using a combined approach: data-driven parcellation of cortical areas, anatomic delineation of subcortical regions, and a cerebellum parcellation based on the Yeo 17-network parcellation (Yeo et al., 2011) (Supplementary Fig. 1).”

Classification accuracy and participant misclassification frequency were largely unchanged.

For each analysis, we present classification accuracy for each phenotypic measure and in-scanner task, and the correlation between mean misclassification frequency and main result mean misclassification frequency. Across all analyses and phenotypic measures, classification accuracy and misclassification frequency are comparable to main results. That is, rates of misclassification are largely unchanged, and each participant’s misclassification frequency is robust to these alternative analysis approaches. See Supplementary Fig. 1, reproduced below:

Supplementary Figure 1. Model performance and misclassification frequency are robust to analysis approach. (a) Classification accuracy for each phenotypic measure using FC calculated from all in-scanner conditions in the Yale dataset, and four different analysis pipelines: an alternative, 368-node parcellation for FC matrix generation, an alternative classification algorithm (ensemble of weak learners), an alternative phenotypic binarization threshold (mean split), and an alternative (10-fold) cross-validation approach (see Methods for additional description of each analysis). Boxplot line and hinges represent median and quartiles, respectively; whiskers extend to most extreme non-outliers; outliers plotted individually (+). r1, rest 1; r2, rest 2; grad, gradual onset continuous performance task; sst, stop signal task; gfc, general functional connectivity. (b) MF, averaged across in-scanner conditions and phenotypic measures to derive a single value per participant, compared between each alternative analysis and main-text analyses. r_s , rank correlation, $n = 128-129$. Note that phenotype mean split is equivalent to $\text{mean} \pm 1/3 \times \text{s.d.}$ for scaled scores; mean split-based model accuracy is not reported for these measures, nor are they included in MF calculation. Given the limited mean split-based results, we repeated this analysis in the HCP data, with comparable results (mean MF $r_s = 0.86$, $P < 0.0001$). 10-fold results reflect 1000 analysis iterations per

phenotypic measure and in-scanner condition (50 per cross-validation partition); all other analyses reflect 100 iterations. In this and all subsequent figures: BNT, Boston Naming Test; WRAT, Wide Range Achievement Test; VL, verbal learning; FW, finger windows; LN, letter-number sequencing; Trails, trail making; VF, verbal fluency; CW, color-word interference; 20Q, 20 questions; Vocab, vocabulary; MR, matrix reasoning.

5. *Next, to mitigate concerns about mean and variance of predictive model accuracy in small samples (as described in (Varoquaux, 2018), discussed on manuscript l.623-624), we have replicated all main results, including cross-dataset analyses, in the Human Connectome Project dataset, bringing the total number of participants analyzed to nearly 1000. Results are presented in Supplementary Fig. 3 (reproduced below) as well as Supplementary Tables 6, 8, 9, and 10, and corresponding methods are discussed on l.749-798 (details about the dataset), l.858-859 (description of 10-fold cross-validation framework), and l.866-867 (family-related limits on exchangeability in permutation testing). We also note that our cross-dataset analyses further mitigate concerns about overfitting (Varoquaux, 2018).*

Supplementary Figure 3. Replication of classification, internal validation, and external validation results in the HCP dataset. Results as presented in Fig. 1b, Fig. 2a-b, Fig. 3a, and Fig. 4a. Note that, given the large HCP sample size, 10-fold cross-validation was used (20 partitions, 50 subsampling iterations each), with the requirement that family members be assigned to the same fold. Given that only two measures were classified, we omit measure versus MF similarity and hierarchical linkage analyses. (d) Results presented as in Fig. 3a. For each of the two classified measures (ciQ/vocabulary and fiQ/MR for HCP/Yale), six models were trained: 1 using all Yale participants, 1 using Yale CCP, 1 using Yale MCP (see Fig. 3 legend for training set sizes), 1 using all HCP participants (number of participants used for training after excluding intermediate and outlier scores and subsampling to balance classes: 230 and 350 for crystallized and fluid measures, respectively), 1 using HCP CCP (168, 216), and 1 using HCP MCP (62, 134). (e) Covariate relationships presented if they were significantly related to MF in low or high scorers ($P < 0.05$, adjusted). For full results and relationship of covariates to mean score, see Supplementary Tables 9 and 10. (a) *** P

< 0.001, FDR adjusted (2 tests), (c) $**P < 0.01$, $****P < 0.0001$, FDR adjusted (2 tests), (d) $*P < 0.0001$, FDR adjusted (9 tests) via nested ANOVA, (e) $****P < 0.0001$; all P values FDR adjusted (22 tests).

6. Finally, we have expanded our discussion of bias in AI and psychometrics, and draw more heavily from this literature (including the suggested reference on facial recognition; I.80) throughout the introduction and discussion. We regret that the second reference URL appears to be incomplete, but we welcome the opportunity to read it in a subsequent review round, should the reviewer still consider it relevant.

3. Data and methodology

Overall, though it is an interesting question, I would expect a more thorough comparison of different models and approaches as well as sample composition to justify the conclusions. Moreover I believe the sample size in the current work is mostest, and it is unclear if some conclusions are not due to relatively small sample size.

To conduct a more thorough comparison of different models and approaches, we have repeated analyses with an alternative classification algorithm, an alternative phenotypic binarization threshold, a continuous prediction approach, an alternative cross-validation approach, and an alternative parcellation (see response to Reviewer 1, comment 2). We have added information about sample composition earlier in the manuscript (beginning at I.106) and have added details to Methods about sample recruitment and composition. We have also expanded Supplementary Table 3, which includes information about sample composition, to include information about the HCP dataset and added Supplementary Fig. 9, which provides more detailed information on the racial and ethnic breakdowns of the three datasets.

We appreciate the reviewers' concerns about sample size in the Yale and UCLA datasets. To assuage concerns about mean and variance of model performance, as well as about potential dataset idiosyncrasies that may limit generalizability, we replicated analyses in the HCP dataset. Using fMRI and phenotypic data from over 650 HCP participants, we demonstrate that misclassification is reliable, generalizes across datasets, and is related to the same covariates (with the addition of sleep disturbance) as in the Yale dataset, despite the differences in sample size, in-scanner tasks, out-of-scanner phenotypic battery, and sample composition (e.g., HCP is a sample of healthy individuals, and has a very different age distribution). See Supplementary Table 3 and Supplementary Fig. 9, reproduced below:

	Yale	UCLA	HCP
Sex	F = 78, M = 51	F = 69, M = 94	F = 361, M = 303
Age	m = 29.5, s.d. = 8.9, n = 129	m = 31.8, s.d. = 9.1, n = 163	m = 28.7, s.d. = 3.8, n = 664
Race	NW = 62, W = 67	NW = 40, W = 123	NW = 154, W = 498
Symptom severity	m = 0.74, s.d. = 0.73, n = 129	m = 0.58, s.d. = 0.46, n = 163	m = 47.6, s.d. = 8.7, n = 663
Education	m = 15.6, s.d. = 3.1, n = 128	m = 14.5, s.d. = 1.8, n = 163	m = 15.0, s.d. = 1.7, n = 664
Stress	m = 24.0, s.d. = 9.6, n = 127		m = 47.8, s.d. = 9.2, n = 663

PSQI	m = 6.6, s.d. = 3.7, n = 125		m = 4.6, s.d. = 2.8, n = 664
Positive affect	m = 31.0, s.d. = 8.2, n = 128		m = 50.3, s.d. = 7.9, n = 663
Negative affect	m = 19.0, s.d. = 8.0, n = 129		
Rx	NRx = 69, Rx = 42	NRx = 101, Rx = 62	
Dx	NDx = 52, Dx = 66	NDx = 41, Dx = 122	

Supplementary Table 3. Demographic and clinical information for the Yale, UCLA, and HCP datasets. m, mean; s.d., standard deviation. For categorical variables, number of participants per group reported.

Supplementary Figure 9. Reported racial and ethnic breakdowns of the studied samples.

Last, contributors to misclassification seems to be a big point, but how were they selected, how independent they are is unclear. For example, I can imagine that self-reported stress and sleep disturbance have some correlation. It would help to further understand the interdependence of these factors and interplay with misclassifications.

We apologize for not including more information about covariate selection and interdependence in our initial submission. We have expanded discussion of covariates in Methods 1.962-967:

“These covariates reflect key demographic features of participants, as well as a range of clinically relevant information. For the latter, we complemented an extensively validated structured diagnostic interview for the major DSM-5 disorders (Sheehan et al., 1998) with self-report measures that reflect participants’ lived experiences with mental health concerns (e.g., sleep disturbance and perceived stress, both common experiences that transcend diagnostic boundaries (Catabay et al., 2019; Freeman et al., 2020; Hewitt et al., 1992; Riemann et al., 2020)).”

We have also created three figures (Supplementary Figs. 6-8, reproduced below) to visualize pairwise relationships between covariates in each dataset. To address the question about the interplay between covariate interdependence and misclassification, we tested the relationship between every pair of covariates that were significantly related to

misclassification frequency in a given dataset. For each such significant relationship, we re-tested the relationship between each covariate in the pair and misclassification frequency, while controlling for the other. Results are presented in Supplementary Figs. 6-8. We note that these analyses complement the regression analyses presented in Table 9 and Fig. 4a, Supplementary Fig. 2e, and Supplementary Fig. 3e, which present the results of a multilinear model of misclassification frequency using all pairwise significant covariates, with these covariates remaining significantly related to misclassification frequency in most cases.

Supplementary Figure 6. Relationships between each pair of covariates in the Yale dataset. Individual variable distributions on the main diagonal. For pairwise relationships, two continuous covariates presented as a scatterplot, one continuous and one categorical as a boxplot, and two categorical as a faceted bar plot. Boxplot line and hinges represent median and quartiles, respectively; whiskers extend to most extreme non-outliers. Lines on scatterplots reflect

smoothed conditional means and their 95% CI. Relationships between variables significantly related to MF enclosed in boxes. For each significant ($P < 0.05$, via Spearman correlation and Mann-Whitney U test) such relationship, we re-tested the relationship between each variable in the pair with MF while controlling for the other, either via partial Spearman correlation (continuous covariate of interest, continuous or binary control covariate) or via Mann-Whitney U test (binary covariate of interest and residualized MF [i.e., regressed on continuous control covariate]). *Significant ($P < 0.05$) and main result-consistent (i.e., sign/direction) relationship with low-scorer MF after control, *significant ($P < 0.05$) and consistent relationship with high-scorer MF after control. No significance test results (i.e., asterisks) reported for covariate pairs that are not significantly associated with each other. For related results, see regression analyses reported in Supplementary Tables 9 and 10. NW, non-white; W, white; NRx, not taking psychiatric medication; Rx, taking psychiatric medication; NDx, no diagnosis via interview; Dx, one or more diagnoses via interview; F, female; M, male; gsi, Brief Symptom Inventory global severity index.

Supplementary Figure 7. Relationships between each pair of covariates in the UCLA dataset, presented as in Supplementary Fig. 6. If only one covariate in pair remained significantly related to MF after control, that covariate's name is reported (e.g., *rx*rx for Hopkins/Rx indicates that medication status remains significantly associated with low-scoring and high-scoring MF after controlling for symptom severity, but symptom severity is not significantly associated with low-scoring or high-scoring MF after controlling for medication status).

Supplementary Figure 8. Relationships between each pair of covariates in the HCP dataset, presented as in Supplementary Figs. 6-7.

4. Statistics and treatment of uncertainties

All the data is presented in a good way. However, as also written earlier in the review, I am not sure whether the prediction approach taken could not be compared to alternative approaches to be sure the misclassification is not accounted for by the approach.

Specifically; 1. Though I understand that the key goal was to investigate why models fail for certain individuals, I am unsure why only one particular model was explored had various specific features that may have contributed to misclassification such as 1) binarization and 2) the approach of leave-one-out cross validation, but also 3) imaging parameters. First, binarization may lead to overinterpretation of differences and lacking training on subtle relationships between brain and behavior that are better caught by an individual difference approach that uses the categorical information. Second, though LOOCV is often used in the literature and has its value, it also has downsides with respect to variance and bias (<https://stats.stackexchange.com/questions/61783/bias-and-variance-in-leave-one-out-vs-k-fold-cross-validation>). This leads me to wonder about the generalizability of the findings. Third, also other options with respect to imaging markers (parcellation, summary of data) seem to be under explored, raising also here the question about generalisability.

We again thank the reviewer for pointing out the utility of replicating results with alternative analysis approaches. Specifically, to address these concerns, we performed the following additional analyses:

- 1. Binarization: To address this point, we present a supplementary analysis in which we replicate results using a continuous prediction framework, with observed phenotype regressed out of the difference between predicted and observed phenotypic measures, thus correcting for linear relationships between these terms (Le et al., 2018; Smith et al., 2019). We relate this residualized prediction deviation, averaged across in-scanner conditions and phenotypic measures, to the 15 utilized covariates in the Yale dataset and demonstrate that goodness of regression-based model fit is related to the same covariates that track misclassification frequency (e.g., increased education tracks overestimation of performance, as does lower age, self-reported white race, higher overall cognitive performance, and lower head motion; Supplementary Table 11, reproduced above).*

While we agree that the conversion of continuous to categorical variables requires information loss, we chose to binarize phenotypic measures in main analyses to avoid a methodological pitfall, detailed in (Smith et al., 2019) and alluded to above: the common underestimation of predictor coefficients moves outcome dependence into the deviation of predicted from observed outcome. That is, in the continuous case, the difference between predicted and observed phenotype is often confounded by the phenotype, itself. This problem is exacerbated by cross-validation, and may involve linear and non-linear relationships between residuals and the phenotype of interest, complicating efforts to evaluate residuals. To eliminate this issue entirely and derive unambiguous classification outcomes (i.e., correctly versus incorrectly classified labels), we chose to transform phenotypic measures into categorical variables. Binarization was the most sensible choice given sample sizes, and classification of binarized outcomes is a very common approach to relate brain to phenotype (e.g., diagnosis (Morgan et al., 2020; Saccà et al., 2019; Steardo et al., 2020), neurocognitive task performance (Cabral et al., 2017)). This motivation is summarized in Results, I.116-118. We note that while these choices may not maximize prediction performance, this is not our goal. Rather, we seek to demonstrate that at a given prediction accuracy, a stable and meaningful subset of individuals is poorly fit by the model. We see these results as proof-of-principle, to establish that models fit some

individuals better than others and that model failure is systematic and meaningful, and look forward to pursuing finer-grained questions in follow-up work using continuous phenotypic measures.

2. *Leave-one-out cross-validation: We thank the reviewer for raising this important point. We agree that the bias-variance tradeoff and the choice of k is a complicated and controversial issue, as described, for example, on the stackexchange thread to which the reviewer refers. We chose to use leave-one-out cross-validation due to our relatively small sample sizes (Scheinost et al., 2019), but have replicated main results using 10-fold cross-validation and have used 10-fold cross-validation in all HCP analyses, given the larger sample size. Additional details about these analyses, as well as reproductions of the figures that present corresponding results, can be found above (response to Reviewer 1's comment 2).*
3. *Imaging parameters: To demonstrate that results are not specific to the imaging parameters, we added a third dataset (HCP), such that results are replicated in three datasets that used distinct imaging protocols (Methods: Datasets, 1.618-798). Further, we repeated main analyses using a different parcellation (Methods: Functional parcellation and network definition, 1.809-813) and demonstrate that results are comparable (Supplementary Fig. 1). This analysis is further described, and the corresponding figure reproduced above, in response to Reviewer 1's comment 2.*
4. *Additional analyses to further explore the potential impact of modeling choices are described in response to Reviewer 1's comment 2.*

E. Conclusions

Though in general I believe the conclusion makes sense, I am not sure if the alternative hypotheses about misclassification are tested thoroughly enough to really provide novel insights about the nature of misclassifications. Indeed, site-effects or difference between gender, or even ethnicity are often considered to be important factors in biopsychological research, so it seems a little bit of a too general conclusion to be wrong. In particular the notion of individuals 'who defy stereotypes' and relation to race (ethnicity?), education, and other demographic factors is tricky and not sufficiently addressed in the current work.

We thank the reviewer for pointing out that our initial submission did not sufficiently explore artifactual explanations for our findings (e.g., small sample size, selected algorithm, and cross-validation approach). We now demonstrate that results hold across three, independent datasets that together comprise nearly 1000 participants, 24 different in-scanner conditions (of varying experimental design), and 21 phenotypic measures. Results are also robust to parcellation choice, cross-validation approach, phenotype thresholding, and classification algorithm.

We agree that the results are intuitive; indeed, there is an extensive literature documenting bias in neurocognitive test results, and we have expanded discussion of this literature in the revised manuscript's introduction and discussion. However, we believe that our work is novel and useful for two key reasons: 1. It is not required that bias in a given phenotypic measure be reflected in brain-based models of that measure. That is, we are not telling our models explicitly about any of these demographic or clinical covariates, and yet these covariates track classification outcomes, suggesting that models are predicting composite stereotypical profiles, not unitary neurocognitive measures. We feel that this

finding is itself important, and 2. Given that this is the case, we must ask ourselves what our brain-based models are truly predicting in our given sample. As it currently stands, the psychometric literature remains relatively siloed from the human neuroscience literature. Many human neuroscientists are relating brain activity to neurocognitive phenotypes without accounting for relationships between the phenotype of interest and demographic or clinical covariates. These relationships must guide the way we interpret these phenotypic measures and the brain models that are derived from them. Doing so will determine the generalizability and applications of these models and is thus a crucial step toward developing more precise and useful brain-phenotype models.

Finally, we appreciate that the notion of defying stereotypes and its relationship to the studied covariates is extremely complex and outside the scope of authors' primary expertise. To ensure that we are presenting and interpreting these findings in a manner that is productive and responsible, we have partnered with a clinical psychologist who specializes in culturally informed neuropsychological evaluations, Dr. Carmen Carrión. Dr. Carrión is now part of our authorship team, and has offered extensive feedback on the revised manuscript to this end, helping us expand related discussions throughout the manuscript.

F. Suggested improvements

It depends a bit on the preferred scope of the work, but I would suggest to compare the current model to alternative models and more formally test misclassifications and also archetypes of certain datasets. Moreover, I would more carefully explain the selected variables that are thought to explain misclassification. Possibly, I'd also account of structure in the functional models, as this may also be of influence. Other comments, such as the question about the sample size, are also earlier in the review.

See response to Reviewer 1, comment 2 (addition of analyses using an alternative classification algorithm, cross-validation approach, larger dataset, etc.). See response to Reviewer 1, comment 3 (addition of discussion regarding covariate selection and analyses to explore covariate relationships). See introduction and discussion for additional description of the existing literature on bias in AI and neurocognitive testing, and its relationship to misclassification. While we agree that the addition of brain structural measurements to models may provide interesting additional insight into brain-phenotype associations and relationships between brain measures and various covariates, we focus here on differences in patterns of brain activity that are associated with neurocognitive task performance. The relationship between fixed structural features and the flexible functional organization of the brain within this fixed infrastructure remains poorly understood, and methods are only beginning to emerge to combine these measures. As such work develops, we agree that incorporating structural brain features into brain-phenotype models will be an exciting area for future work.

G. References

Though the references are appropriate. Possibly this work is also interesting: <https://www.sciencedirect.com/science/article/pii/S1053811918305081?via=ihub>. Possibly it may help to discuss 'ethical AI' in a broadly, as also suggested earlier.

We thank the reviewer for this helpful suggestion. We have added a discussion of (Cui and Gong, 2018) to Methods (l.624) along with (Varoquaux, 2018), as suggested previously. We have also expanded our discussion of bias and ethics in AI (with the suggested reference cited on l.80) and neurocognitive testing in the introduction (l.67-84) and discussion (l.514-547).

H. Clarity

Overall the manuscript is clearly written however it seems like the question is broader than the introduction, approach, and discussion support.

We thank the reviewer for this supportive and helpful feedback. We have substantially revised the manuscript to further clarify the question of interest and key findings. We have, as described previously, more extensively tested the robustness of results to analysis choices and replicated results in a third, large dataset. These analyses suggest that results are not the product of small sample size, chosen classification algorithm, cross-validation approach, parcellation, or phenotype binarization. Given the robustness of key findings across analysis approaches and very different datasets—in size, scan protocol, in-scanner tasks, phenotypic measures, demographic features, and presence (Yale and UCLA) or absence (HCP) of participants with psychiatric disorders—we believe that the manuscript now more convincingly answers the broadly relevant question it poses. We thank Reviewer 1 for their helpful suggestions to this end.

I hope my comments are helpful.

Referee #2 (Remarks to the Author):

The study evaluated the generalizability of brain-phenotype prediction, and observed that a subset of individuals is consistently misclassified using a one-size-fits-all approach. Findings were observed across two different datasets.

These potentially important results argue for a more nuanced modelling of brain-phenotype relationships, which in particular speak against common one-size-fits-all approaches. The analyses are thoughtful and comprehensive. I would nevertheless have a series of suggestions to further strengthen the analytical/predictive learning component of the work, in order to solidify the main conclusions of the paper.

1) To enhance readability, the authors could provide more details on the datasets evaluated already in the main manuscript e.g., number of subjects, sites, age, healthy/diagnosed etc for both the Yale and UCLA CNP data. This could be presented in line 112 but also in part in the abstract/end of introduction already in my opinion.

Thank you for this helpful suggestion. We have added details on included datasets at the beginning of the results section, as suggested (l.103-112 in the revised manuscript):

“For primary analyses, we used a new dataset collected at Yale, which includes fMRI (6 in-scanner task runs [Supplementary Table 1], 2 resting-state runs [hereafter, “conditions”]) and out-of-scanner phenotypic (neuropsychological

tests and self-report measures; Supplementary Table 2) data ($n = 129$, 78 female, age mean (s.d.) = 29.5 (8.9), 66 with one or more psychiatric diagnoses; Supplementary Table 3). For cross-dataset and replication analyses, we used fMRI and comparable neurocognitive data from the UCLA Consortium for Neuropsychiatric Phenomics (Poldrack et al., 2016) and Human Connectome Project (HCP) (Van Essen et al., 2013) datasets (UCLA: $n = 163$, 69 female, age mean (s.d.) = 31.8 (9.1), 122 with one or more psychiatric diagnoses; HCP: $n = 664$, 361 female, age mean (s.d.) = 28.7 (3.8), healthy sample, broadly defined; Supplementary Table 3).”

Given the addition of analyses in the HCP dataset, we have included relevant details about this dataset in the same manuscript sections and have added them to the dataset information in the supplement (now Supplementary Table 3).

2) The schema on figure 1 suggests that LOO cross validation was used for classifier training and performance evaluation. LOO may have limitations in testing classifier generalizability, and one may thus want to know whether results are consistent when a more conservative (e.g 5 fold) cross validation scheme was used. Moreover, did the authors perform a nested cross validation in the training dataset for feature selection/validation (which one may recommend) prior to testing the performance in the outermost held out fold? In that respect, can the authors *[remaining text cut off]*

We agree that the bias-variance trade-off and selection of k in k-fold cross-validation is an important and complex topic. To explore this further and to balance bias and variance given sample sizes, we replicated the main results using 10-fold cross-validation, which we also used for the new HCP analyses. See responses to Reviewer 1, comment 2 and Reviewer 1, comment 4.

Feature selection was performed exclusively in the subset of participants selected for training (after undersampling to balance classes). The P value threshold for edge selection was selected based on prior work demonstrating stable model performance across a range of thresholds, including $P < 0.05$ and sparsity thresholds comparable to that used in cross-dataset analyses (Finn et al., 2015; Greene et al., 2018). For the analysis in which we used an ensemble of weak learners, rather than SVM, to classify outcome, we used an initial leave-one-out cross-validated analysis for hyperparameter optimization (ensemble aggregation method, number of learners, learning rate [where relevant], and minimum leaf size). Because we sought to apply a consistent approach across phenotypic measures, we conducted this optimization in a preliminary, leave-one-out analysis for a subset of measures and used the consensus hyperparameters to classify all 16 phenotypic measures (bagging, 150 learners, with minimum leaf size = 1). This is described in the revised manuscript (1.842-850).

3) The authors mention that linear SVMs were used to classify individuals as high/low scorers based on FC patterns. Given that many of the phenotypic scores are continuous, wouldn't a regression SVM framework be more adequate, as it may avoid unnecessary subsplits of the phenotypic variables and one would also not need to remove datapoints with inter-mediary scores? Ideally, the authors could show consistency of their findings when using regression SVM, and could then assess residuals for the prediction instead of correct/incorrect classification results.

To address this point, we conducted a supplementary analysis in which we use functional connectivity to predict continuous neurocognitive measures and demonstrate that the same covariates that track misclassification also track the residualized difference between predicted and observed phenotype (Supplementary Table 11, reproduced above). We agree that information is lost by transforming continuous scores to categorical scores, but we did so in primary analyses to avoid a methodological pitfall: residual dependence on the phenotype of interest (Smith et al., 2019). Further, while relating continuous outcomes to the brain may yield improved prediction accuracy, we note that this is not germane to our primary goal in this paper: to demonstrate, as proof-of-principle, that for a given model, a consistent and meaningful subset of individuals is poorly fit. In future work, we plan to develop methods to relate continuous outcomes to model fit in an unbiased fashion in order to study more specific questions about a given model's fit. Please see response to Reviewer 1, comment 4 for additional details about our choice to binarize phenotypic measures, and to Reviewer 1, comment 2 for results of supplementary analyses using an alternative binarization threshold and continuous phenotypic measures.

4) Have analytical strategies been used for cross-dataset homogenization e.g. linear regression and/or combat, for both the FC and/or cognitive phenotypes. If not, could the authors comment on the potential utility of such approaches and discuss whether their current findings might be affected or invariant to those (could also be a supplementary analysis).

In cross-dataset analyses, data from multiple datasets were never mixed. That is, a model was trained in one dataset, and tested in the other. Dataset-specific site effects would not be expected to generalize (consistent with the goal of dataset homogenization). Indeed, this in broad terms was our motivation for the cross-dataset analysis: one could imagine any number of dataset-specific confounds that may contribute to idiosyncratic results that fail to generalize across datasets (Alfaro-Almagro et al., 2020). The success and specificity of our cross-dataset results (i.e., the finding that correct and incorrect classification have comparable meanings across datasets) demonstrate that our results are not the product of dataset idiosyncrasies—indeed, lack of dataset homogeneity is a test of the robustness of results across different scanning protocols, related but distinct neurocognitive measures, and sample demographics (we discuss the latter, and its relationship to stereotypical profiles, in detail in the revised discussion, 1.468-476). That said, we recognize that identifying reliable brain-phenotype relationships requires large sample sizes (Marek et al., 2020), and that many researchers will seek to achieve them by aggregating data from multiple sites into a single sample. In this case, we recommend efforts be taken to homogenize the data to avoid the development of confounded models.

5) Both the Yale and the UCLA sample include "clinically naturalistic and demographically diverse participants". While this transdiagnostic inclusion is great, would it be possible to show that overall findings hold even if only 'healthy' individuals without concurrent diagnosis were included, to show generalizability of the results to a purely non-clinical sample?

To address this point, we have added the Human Connectome Project dataset to our analyses, which recruited healthy participants in a narrow age range. We replicate all main

analyses, including cross-dataset analyses, using the HCP dataset; see Supplementary Fig. 3, reproduced above, as well as Supplementary Tables 6, 8, 9, and 10. We include this as a motivation for the inclusion of the HCP dataset in Methods: Datasets, l.623-625. For additional detail on these analyses, see response to Reviewer 1, comments 2 and 3.

Minor:

I.330 - please report the corresponding test statistics, in addition to the significant p-values

We have added test statistics here (l.342-346 in the revised manuscript):

“Of these covariates, 5 were significantly related to MF in both low and high scorers (Fig. 4a): age (low $r_s = -0.38$, $P < 0.0001$; high $r_s = 0.32$, $P < 0.001$), race (low group median difference = -0.14 , $P < 0.05$; high group median difference = 0.19 , $P < 0.01$), education (low $r_s = 0.32$, $P < 0.01$; high $r_s = -0.33$, $P < 0.001$), overall cognitive ability (low $r_s = 0.31$, $P < 0.01$, high $r_s = -0.45$, $P < 0.0001$), and motion (low $r_s = -0.40$, high $r_s = 0.48$, both $P < 0.0001$; all P values FDR adjusted).”

I.393 - First sentence of the discussion is a bit odd. It might be worthwhile if the authors underscored the significance of brain-behavior prediction approaches a bit more specifically. Why is such an endeavour important?

We have revised the beginning of the discussion, which now reads:

“Relating individual differences in brain activity to behavior, traits, and clinical symptoms is a long-standing aim of human neuroscience, one that has been advanced by the application of machine learning algorithms to neuroimaging and phenotypic data. Resulting models, which have demonstrated high accuracy and generalizability (Barron et al., 2020; Greene et al., 2018; O’Connor et al., 2021; Rosenberg et al., 2015; Wager et al., 2013), have the potential to reveal the macroscale neural circuitry underlying a range of phenotypic measures, with relevance in both health and disease (Woo et al., 2017)” (l. 406-411).

I. 717 - please clarify whether the initial threshold is corrected or uncorrected.

Given the cross-validated nature of the analysis and consistent with prior work (e.g., Dubois et al., 2018; Finn et al., 2015; Rosenberg et al., 2015; Shen et al., 2017), the initial edge selection threshold does not need to be and is not corrected. We have clarified this in the text (l.834-835). Thank you for pointing out this omission.

Please provide the Matlab version in the methods

We have added this information to Methods: Statistical analysis, l.1028.

Referee #3 (Remarks to the Author):

The authors investigate the failure modes of a 2-dimensional linear support vector machine classifier for prediction of binarized behavioral variables with fMRI functional connectivity (FC) data. They use 9 different sources of FC data (two rest sessions and 7 tasks) and 16

different behavioral phenotypes, a 129 subject primary dataset and 163 subject validation dataset. For each behavioral variable they discard roughly 15% of subjects near the mean (within +/- 1/3 SD of mean) as well as lower (but not upper) outliers, before dichotomizing with the sign operator. They find that some subjects tend to be misclassified consistently over different types of FC data, and misclassification rates are correlated between different behavioral phenotypes.

I find the work overstated and suffering from at least one critical analytical flaw. The findings overstate the impact of results from one single and very simplistic prediction model, when the neuroimaging literature as a whole uses a vast array of different models that are all much more complex than this approach. The analytical flaw relates to the use of permutation techniques that appear to randomly shuffle data independently, when instead synchronized permutation is needed to ensure that the expected (within subject FC) correlation is accounted for.

Overall, I see this this work as a deep dive into the limitations of the 2D linear SVM approach to FC prediction. It is not an unuseful investigation, but I don't see how such a narrow methodological study would merit the attention of the readers of this journal.

We thank the reviewer for this feedback. We have substantially expanded supplementary analyses to demonstrate the generalizability of these results beyond a single methodological approach, as described throughout this response.

Major Issues

The predictive method used here (independent selection of FC edges to compose two feature vectors based on selected edges, fed into a linear SVM) is but one method out of a constellation of machine learning methods available to conduct this exercise. This reviewer is not hinting that some sophisticated deep learning must be deployed. Schulz et al. (2020) have nicely shown that brain data don't benefit from the super-high dimensionality of DL methods, but best practice (as illustrated by Schulz et al) is to let the data dictate the dimensionality required... and in this reviewer's experience the data generally rarely indicate just 2 dimensions are needed.

With more dimensions, (or if fixed on 2D) nonlinear SVM, or ensemble methods that combine a battery of weak learners, the authors would likely find fewer and perhaps an unexceptional proportion of poorly predicted subjects. Thus, this work is severely limited by its use of this one very simplistic model.

As suggested, we have implemented an alternative classification algorithm, using an ensemble of weak learners to classify score. Results with this approach are nearly indistinguishable (both in terms of classification accuracy and participant misclassification frequency) from SVM-based results. For additional details about the approach and results, see response to Reviewer 1, comment 2.

The calculation of significance for 'Misclassification is generic to FC type' (Fig 2b) is using

the wrong null hypothesis. It is inferred that null task-by-task correlations are obtained by correlating MF from independently shuffling the +/-1 label for each FC. This corresponds to a null hypothesis that the FC and the phenotype are independent *and* the different FC data are independent. Clearly, the same subjects give rise to the 9 FC measures and thus are dependent. Instead, the authors want to test a null of just "FC and the phenotype are independent" while preserving the dependence structure among the FC measures. Put another way, the lower diagonal correlations in Figure 2b must reflect the intrasubject dependence of the imaging measures. This is done by ensuring that the null MF correlations arise from the same permutation of phenotype labels for all imaging measures correlated (this may require a smaller dataset to ensure a common set of subjects for all 9 imaging measures; alternatively, the synchronized permutation could be done in a pairwise fashion, for each pair of the 9 measures).

It is likely that with use of synchronized permutation the null correlations will look more like the observed correlations, weakening or undermining the conclusions drawn from Fig 2b.

We agree entirely and apologize for not being clearer about this initially. We did use a synchronized permutation approach, such that permutations were held constant across in-scanner conditions for a given analysis iteration. We have clarified this point in the manuscript (1.865-866): "Permutations were fixed across in-scanner conditions...". We have also switched the comparison between original and permuted correlations to a paired test (1.892-893), given the dependence of FC across task conditions, as this reviewer points out.

Other Serious Concerns

I find the filtering of the data troubling, i.e. the elimination of the central 2/3 SD of the subjects. I was expecting at least a sub-analysis to demonstrate what happens when no middling subjects are dropped. Or a comment on at least how the authors anticipate the results to change as this threshold is shrunk or expanded.

To address this concern, we present a supplementary analysis in which the normative mean is used as a threshold for high and low scorers. Results were comparable to those in which the central 2/3 s.d. of the participants are excluded. Please see the response to Reviewer 1, comment 2 for additional details on and results of this analysis.

It seems arbitrary and worrying that only low scorers ($< -3sd$) were removed and not high scores ($> 3sd$). While this is clearly described no justification is given, nor any sensitivity analysis provided.

Thank you for pointing out this oversight. In the revised manuscript, we explain that extremely low scorers were excluded to ensure that participants understood and performed the tasks as intended, and include references to demonstrate precedent for this common data processing step (1.817-821):

*"To ensure that all participants understood and performed the neurocognitive tests as intended, we used normative means and standard deviations for each measure to exclude outlier extremely low scorers (score \leq mean $- 3*s.d.$) (Guilmette et al.,*

2020), as is common and recommended practice (e.g., Esterman et al., 2014; Karcher et al., 2019; Mackenzie and Wonders, 2016)."

We also note that very few participants were excluded for low scores: Yale: BNT (n=16), VL (n=1), VL delay (n=3), FW (n=1), Trail making (n=4), CW (n=5), remaining tests n=0; UCLA: LN (n=1), vocabulary and MR n=0; HCP: no low outliers.

"FC from the best-performing condition for each measure classified scores on 14/16 phenotypic measures with above-chance performance." This is a crucial statement of results that clouds the actual multiplicity involved: I compute that there were $16 \times 9 = 144$ tests computed; was FDR computed over this set of 144 tests? Without further clarification, showing the best in each suggests the results are overly optimistic. While the accuracies are provided in Table S3 there are no p-values. This concern would be addressed if Table S3 was augmented by the uncorrected p-values for each combination of 16 measures and 8 FCs, along with the FDR p-value threshold in the caption so readers can judge the pattern and strength of effects.

We apologize for not presenting these results more clearly. As suggested, we have added P values to the supplementary tables in which we present classification accuracy for all in-scanner conditions and phenotypic measures for each dataset (now Supplementary Tables 4-6 in revised materials). For ease of viewing given the size of the tables, we indicate values that are significantly greater than chance after FDR adjustment with '' or in the caption, and define the number of comparisons in each caption. We have also clarified throughout the text the number of P values submitted to each correction (see response to Reviewer 3, final comment).*

Minor Points

For the breadth of the claims made, the scale of these datasets are disappointing (129 primary, 163 validation). Recent work suggests that both association and prediction cognitive outcomes with brain phenotypes requires at least 2000 subjects to have respectable out of sample performance (Marek et al., 2020; see results on prediction, Fig 3). That said, the authors are to be commended on collecting a demographically and clinically diverse sample.

We thank the reviewer for this supportive comment. We agree that the Yale and UCLA sample sizes are relatively modest. To address related concerns, we have replicated results in a third dataset, the Human Connectome Project, adding over 650 participants. See response to Reviewer 1, comments 2 and 3 for additional details. Further, as noted in this comment, out-of-sample performance is a crucial test of result generalizability. It is for this reason that we performed cross-dataset analyses, which we extended in revised materials to include the HCP dataset, to demonstrate that misclassification generalizes across datasets (Fig. 3a and Supplementary Fig. 3d, reproduced above). Finally, identification of model-based subtypes and building subtype-specific models, as shown here, may in fact reduce the necessary sample size to identify reliable brain-phenotype relationships.

Results state there were 16 non-imaging phenotypic measures considered, but Table S2

lists 17... while a reader can cross-reference figures and captions to deduce the 16 it would simply better of Table S2 clearly indicated the 16 measures used.

We have added asterisks to Supplementary Table 2 to indicate the 16 phenotypic measures, and hashes to indicate covariates used in post-hoc analyses, with these symbols explained in the corresponding table caption.

While the referenced GFC method is explicit about removing task variation explained by a usual task model, no mention of this is given about the treatment of the individual task data. It needs to be made clear whether the task data was used raw (after preprocessing) or after residualizing with respect to the task regressors.

We have clarified on l.860-861 that, given the continuous design of the Yale tasks (purpose-built for functional connectivity analyses) and the comparable prediction performance of “standard” and residualized FC (Greene et al., 2020), the task data were used raw, without regression of task timing.

Edges were selected on the basis of an intersubject correlation between edge fMRI data and -1/+1 phenotypic labels, significant at $P < 0.05$. However, correlation p-values are based on a bivariate Gaussian assumption that has no justification when X's are binary. If authors want to appeal to the meaningfulness of the 0.05 threshold (or more importantly the comparability with other work using continuous labels) they should replace the correlation with a two-sample t-test.

We appreciate the opportunity to clarify this point. The point-biserial correlation (mathematically equivalent to the Pearson correlation) yields a test statistic that follows the Student's t-distribution, rendering P values derived from the point-biserial correlation identical to those derived from a two-sample t-test. We also confirmed this empirically, by ensuring that the same edges are selected via correlation and t-test, and via simple simulation in Matlab. For more, see for example (Kurtz and Mayo, 1979).

FDR correction was used a different points in this work, but it is not clear the family of tests to which it was applied. For example, at line 740 it isn't clear what set of p-values is submitted to FDR; in Figure 1, it isn't clear if $16 \times 9 = 144$ p-values were submitted to FDR or just 14. To avoid confusion, clearly state the set and count of p-value submitted to correction.

We thank the reviewer for pointing out this omission. We have clarified the set and count of P values submitted to FDR adjustment throughout the manuscript. Every time multiple comparison correction is mentioned in Methods and figure legends, and at several points throughout results where it may otherwise be unclear, we have indicated the relevant number of comparisons. This is also summarized in Methods: Statistical analysis, l.1034-1036. In particular, the adjustment to which the reviewer refers in this comment is described on l.871-873 and in the Fig. 1 caption (l.159).

References

Alfaro-Almagro, F., McCarthy, P., Afyouni, S., Andersson, J.L.R., Bastiani, M., Miller, K.L., Nichols, T., and Smith, S.M. (2020). Confound modelling in UK Biobank brain imaging. *BioRxiv* 2020.03.11.987693 doi:10.1101/2020.03.11.987693.

Barron, D.S., Gao, S., Dadashkarimi, J., Greene, A.S., Spann, M.N., Noble, S., Lake, E.M.R., Krystal, J.H., Constable, R.T., and Scheinost, D. (2020). Transdiagnostic, connectome-based prediction of memory constructs across psychiatric disorders. *Cereb. Cortex* 31, 2523–2533 doi:10.1093/cercor/bhaa371.

Cabral, J., Vidaurre, D., Marques, P., Magalhães, R., Silva Moreira, P., Miguel Soares, J., Deco, G., Sousa, N., and Kringelbach, M.L. (2017). Cognitive performance in healthy older adults relates to spontaneous switching between states of functional connectivity during rest. *Sci. Reports* 2017 7 1, 1–13 doi:10.1038/s41598-017-05425-7.

Catabay, C.J., Stockman, J.K., Campbell, J.C., and Tsuyuki, K. (2019). Perceived stress and mental health: The mediating roles of social support and resilience among black women exposed to sexual violence. *J. Affect. Disord.* 259, 143–149 doi:10.1016/J.JAD.2019.08.037.

Cui, Z., and Gong, G. (2018). The effect of machine learning regression algorithms and sample size on individualized behavioral prediction with functional connectivity features. *Neuroimage* 178, 622–637 doi:10.1016/J.NEUROIMAGE.2018.06.001.

Dubois, J., Galdi, P., Paul, L.K., and Adolphs, R. (2018). A distributed brain network predicts general intelligence from resting-state human neuroimaging data. *Philos. Trans. R. Soc. B Biol. Sci.* 373, 20170284 doi:10.1098/rstb.2017.0284.

Van Essen, D.C., Smith, S.M., Barch, D.M., Behrens, T.E.J., Yacoub, E., Ugurbil, K., and WU-Minn HCP Consortium, for the W.-M.H. (2013). The WU-Minn Human Connectome Project: an overview. *Neuroimage* 80, 62–79 doi:10.1016/j.neuroimage.2013.05.041.

Esterman, M., Rosenberg, M.D., and Noonan, S.K. (2014). Intrinsic fluctuations in sustained attention and distractor processing. *J. Neurosci.* 34, 1724 doi:10.1523/JNEUROSCI.2658-13.2014.

Finn, E.S., Shen, X., Scheinost, D., Rosenberg, M.D., Huang, J., Chun, M.M., Papademetris, X., and Constable, R.T. (2015). Functional connectome fingerprinting: identifying individuals using patterns of brain connectivity. *Nat. Neurosci.* 18, 1664–1671 doi:10.1038/nn.4135.

Freeman, D., Sheaves, B., Waite, F., Harvey, A.G., and Harrison, P.J. (2020). Sleep disturbance and psychiatric disorders. *The Lancet Psychiatry* 7, 628–637 doi:10.1016/S2215-0366(20)30136-X.

Greene, A.S., Gao, S., Scheinost, D., and Constable, R.T. (2018). Task-induced brain state manipulation improves prediction of individual traits. *Nat. Commun.* 9, 2807 doi:10.1038/s41467-018-04920-3.

Greene, A.S., Gao, S., Noble, S., Scheinost, D., and Constable, R.T. (2020). How tasks change whole-brain functional organization to reveal brain-phenotype relationships. *Cell Rep.* 32, 870287 doi:10.2139/ssrn.3471318.

Guilmette, T.J., Sweet, J.J., Hebben, N., Koltai, D., Mahone, E.M., Spiegler, B.J., Stucky, K., Westerveld, M., and Participants, C. (2020). American Academy of Clinical Neuropsychology consensus conference statement on uniform labeling of performance test scores. *Clin. Neuropsychol.* 34, 437–453 doi:10.1080/13854046.2020.1722244.

Hewitt, P.L., Flett, G.L., and Mosher, S.W. (1992). The Perceived Stress Scale: Factor structure and relation to depression symptoms in a psychiatric sample. *J. Psychopathol. Behav. Assess.* *14*, 247–257 doi:10.1007/BF00962631.

Karcher, N.R., O'Brien, K.J., Kandala, S., and Barch, D.M. (2019). Resting-state functional connectivity and psychotic-like experiences in childhood: Results from the Adolescent Brain Cognitive Development study. *Biol. Psychiatry* *86*, 7–15 doi:10.1016/J.BIOPSYCH.2019.01.013.

Kurtz, A.K., and Mayo, S.T. (1979). More Measures of Correlation. In *Statistical Methods in Education and Psychology*, (New York, NY: Springer New York), pp. 311–361 doi:10.1007/978-1-4612-6129-2_10.

Le, T.T., Kuplicki, R.T., McKinney, B.A., Yeh, H.-W., Thompson, W.K., Paulus, M.P., and Tulsa 1000 Investigators (2018). A nonlinear simulation framework supports adjusting for age when analyzing BrainAGE. *Front. Aging Neurosci.* *10*, 317 doi:10.3389/FNAGI.2018.00317/BIBTEX.

Mackenzie, G.B., and Wonders, E. (2016). Rethinking intelligence quotient exclusion criteria practices in the study of attention deficit hyperactivity disorder. *Front. Psychol.* *7*, 794 doi:10.3389/FPSYG.2016.00794.

Marek, A.S., Tervo-clemmens, B., Calabro, F.J., David, F., Uriarte, J., Snider, K., Tam, A., Chen, J., Dillan, J., Greene, D.J., et al. (2020). Towards reproducible brain-wide association studies. *BioRxiv*.

Morgan, S.E., Young, J., Patel, A.X., Whitaker, K.J., Scarpazza, C., van Amelsvoort, T., Marcelis, M., van Os, J., Donohoe, G., Mothersill, D., et al. (2020). Functional magnetic resonance imaging connectivity accurately distinguishes cases with psychotic disorders from healthy controls, based on cortical features associated with brain network development. *Biol. Psychiatry Cogn. Neurosci. Neuroimaging* doi:10.1016/J.BPSC.2020.05.013.

O'Connor, D., Lake, E.M.R., Scheinost, D., and Constable, R.T. (2021). Resample aggregating improves the generalizability of connectome predictive modeling. *Neuroimage* *236*, 118044 doi:10.1016/j.neuroimage.2021.118044.

Poldrack, R.A., Congdon, E., Triplett, W., Gorgolewski, K.J., Karlsgodt, K.H., Mumford, J.A., Sabb, F.W., Freimer, N.B., London, E.D., Cannon, T.D., et al. (2016). A phenome-wide examination of neural and cognitive function. *Sci. Data* *3*, 160110 doi:10.1038/sdata.2016.110.

Riemann, D., Krone, L.B., Wulff, K., and Nissen, C. (2020). Sleep, insomnia, and depression. *Neuropsychopharmacology* *45*, 74–89 doi:10.1038/s41386-019-0411-y.

Rosenberg, M.D., Finn, E.S., Scheinost, D., Papademetris, X., Shen, X., Constable, R.T., and Chun, M.M. (2015). A neuromarker of sustained attention from whole-brain functional connectivity. *Nat. Neurosci.* *19*, 165–171 doi:10.1038/nn.4179.

Saccà, V., Sarica, A., Novellino, F., Barone, S., Tallarico, T., Filippelli, E., Granata, A., Chiriaco, C., Bruno Bossio, R., Valentino, P., et al. (2019). Evaluation of machine learning algorithms performance for the prediction of early multiple sclerosis from resting-state fMRI connectivity data. *Brain Imaging Behav.* *13*, 1103–1114 doi:10.1007/S11682-018-9926-9/TABLES/3.

Scheinost, D., Noble, S., Horien, C., Greene, A.S., Lake, E.M., Salehi, M., Gao, S., Shen, X., O'Connor, D., Barron, D.S., et al. (2019). Ten simple rules for predictive modeling of individual differences in neuroimaging. *Neuroimage* *193*, 35–45 doi:10.1016/J.NEUROIMAGE.2019.02.057.

Sheehan, D. V, Lecrubier, Y., Sheehan, K.H., Amorim, P., Janavs, J., Weiller, E., Hergueta, T.,

Baker, R., and Dunbar, G.C. (1998). The Mini-International Neuropsychiatric Interview (M.I.N.I.): The development and validation of a structured diagnostic psychiatric interview for DSM-IV and ICD-10. *J. Clin. Psychiatry* *59 Suppl 20*, 22–33.

Shen, X., Finn, E.S., Scheinost, D., Rosenberg, M.D., Chun, M.M., Papademetris, X., and Constable, R.T. (2017). Using connectome-based predictive modeling to predict individual behavior from brain connectivity. *Nat. Protoc.* *12*, 506–518 doi:10.1038/nprot.2016.178.

Smith, S.M., Vidaurre, D., Alfaro-Almagro, F., Nichols, T.E., and Miller, K.L. (2019). Estimation of brain age delta from brain imaging. *Neuroimage* *200*, 528–539 doi:10.1016/j.neuroimage.2019.06.017.

Steardo, L., Carbone, E.A., Filippis, R. de, Pisanu, C., Segura-Garcia, C., Squassina, A., De Fazio, P., and Steardo, L. (2020). Application of support vector machine on fMRI data as biomarkers in schizophrenia diagnosis: A systematic review. *Front. Psychiatry* *11*, 588 doi:10.3389/FPSYT.2020.00588/BIBTEX.

Varoquaux, G. (2018). Cross-validation failure: Small sample sizes lead to large error bars. *Neuroimage* *180*, 68–77 doi:10.1016/J.NEUROIMAGE.2017.06.061.

Wager, T.D., Atlas, L.Y., Lindquist, M.A., Roy, M., Woo, C.-W., and Kross, E. (2013). An fMRI-based neurologic signature of physical pain. *N. Engl. J. Med.* *368*, 1388–1397 doi:10.1056/NEJMoa1204471.

Woo, C.-W., Chang, L.J., Lindquist, M.A., and Wager, T.D. (2017). Building better biomarkers: Brain models in translational neuroimaging. *Nat. Neurosci.* *20*, 365–377 doi:10.1038/nn.4478.

Yeo, B.T.T., Krienen, F.M., Sepulcre, J., Sabuncu, M.R., Lashkari, D., Hollinshead, M., Roffman, J.L., Smoller, J.W., Zollei, L., Polimeni, J.R., et al. (2011). The organization of the human cerebral cortex estimated by intrinsic functional connectivity. *J. Neurophysiol.* *106*, 1125–1165 doi:10.1152/JN.00338.2011.

Reviewer Reports on the First Revision:

Referees' comments:

Referee #1 (Remarks to the Author):

I want to thank the authors for the addressing of the various comments and concerns made by the Reviewers. Though overall I find the manuscript has improved, I still fail to see the broader relevance for the insights proven, beyond confirmatory information for the (functional) neuroimaging community.

In particular, current findings are analysed using one particular metric of in vivo brain variation; brain function in three moderately large datasets, one of which including twins. A lot of previous work has already shown modest relationships between brain structure, function and behaviour, for example Marek, biorXiv. Thus, it is intuitive that predictive models would not do better than this.

Further, I do not follow the logic that it is not tautological that bias in neurocognitive tests would not propagate to prediction models. Would any other result than that biased behaviour measures probing individual differences result in biased models in the brain be possible/expected as the models are based on the biased data? Second, I fail to see the key novelty of the message behind what brain-behaviour models are 'truly' predicting. From various studies evaluating brain-behaviour relationships, amongst other UKBiobank and ABCD, but also HCP, it has become clear that confounding factors (age, education, motion, race) but also SES, BMI, water intake, global signal, time of day of scan, may play key roles in driving brain-behaviour relationships. Indeed, such factors are already taken into account in neuroimaging papers investigating brain-behaviour associations. For example, usually specific age-ranges are investigated, effects of motion are evaluated, or (supplementary) analysis performed only in caucasian samples. Thus, I do see the break through insights gained in the current work.

Minor comments;

1. Though I appreciate the inclusion of the HCP dataset, I believe that this sample is biased in the sense that it would include twin-pairs, therefore showing stronger inter-subject similarity than the other datasets. Moreover, the individual datasets individually still are about half the size of the recommended sample size (N=2000) for brain-behavior predictions.
2. In the revision the authors have added one alternative classification algorithm, an ensemble of weak learners with hyperparameters selected via cross-validated optimisation, however, I am not sure if only two different models are sufficient to generalise over the wealth of models possible.
3. Taking for example figure 4, it is clear there is a strong relationship between not only education but also motion and prediction score. Does this not point to an overall bias of motion in the fMRI data? Or what would the mechanism here be?
4. It seems the in-text changes are not tracked, nor fully specified in the letter to the reviewers, making it challenging for me to evaluate alterations and edits in the current revision.

Referee #2 (Remarks to the Author):

The authors have been very responsive to the previous reviewers' comments and provided comprehensive replies. Several additional considerations would still be worthwhile to incorporate into this paper, however:

1) The goal of the work was to clarify in whom predictive brain-behavior models fail and why. The revision now shows such model failure in 3 independent datasets and across two machine learning architectures. I think the paper therefore now gives a broader support of the initial message.

However, I partially also agree with some of the other reviewers' previous comments on whether this is sufficient support for a generalizable conclusion, or whether these describe still algorithm-specific limitations. The authors would therefore be encouraged to either present additional support across a few additional architectures, or at a minimum to further emphasize this as a limitation of the current work in the paper's discussion.

2) The authors outline associations between misclassification and several socio-demographic covariates, incl age, education, etc. Wouldn't in many cases statistical correction and homogenization procedures pre-analysis, often done in the field, mitigate those effects. Further exploration of invariance of their findings to presence/absence of statistical control procedures, and/or additional discussion on this issue could be helpful.

3) In addition to above socio-demographic associations to MF, one issue that was somewhat underspecified was the question of whether there might be specific brain features associated to MF. The authors could present, in supplementary form, eg FC matrices of the CCP and MCP groups and their difference, and to map areas and connections in the brain that contribute to model failure. Alternatively, it may be possible to correlate (for example corresponding to Fig 2) MF frequency with FC measures. As MF misclassification appears to be consistent across conditions, one may further want to know whether FC features associated with MF are also consistent across conditions. I think such an analysis would add potentially meaningful neuroscientific insights to the paper, and should be relatively easy to incorporate.

Minor:

For the HCP dataset, the authors mention that HCP contains family associations so I assume they removed twins but further specification may be worthwhile.

Referee #3 (Remarks to the Author):

The authors are to be commended for a thorough and energetic response to the reviews.

This reviewer's primary concern was about the limited interpretability of the results due to a reliance on a single predictive framework. However, the authors have addressed this in a number of ways, using a completely different family of predictive methods, considering 10-fold instead of LOO cross validation, using a continuous instead of binarised outcome and altering the parcellation scheme used, while finding similar results.

As a result I have no further concerns about the publication of this work.

Author Rebuttals to First Revision:

We thank the reviewers for their suggestions, which we feel have substantially strengthened the manuscript. In particular, we have implemented a fourth predictive modeling algorithm (neural networks) with comparable results, explored differences in brain functional organization between correctly classified and misclassified groups, added a more comprehensive discussion of result implications and next steps, and clarified key points throughout the manuscript. All changes are tracked in the manuscript, and referenced lines of text are also embedded in this document.

Referee #1 (Remarks to the Author):

I want to thank the authors for the addressing of the various comments and concerns made by the Reviewers. Though overall I find the manuscript has improved, I still fail to see the broader relevance for the insights proven, beyond confirmatory information for the (functional) neuroimaging community.

In particular, current findings are analysed using one particular metric of in vivo brain variation; brain function in three moderately large datasets, one of which including twins. A lot of previous work has already shown modest relationships between brain structure, function and behaviour, for example Marek, biorXiv. Thus, it is intuitive that predictive models would not do better than this.

This point highlights precisely why we begin the paper with a presentation of classification performance. Our results demonstrate that a) predicting these phenotypes from fMRI activity patterns is possible (a prerequisite for the subsequent exploration of subgroup-specific model failure), and b) our models perform comparably to those previously reported (proof that our results are not driven by failure to optimize models in keeping with field standards).

That is, the reviewer correctly points out that there is a rapidly growing body of work, by our group and others, relating brain activity to phenotype via predictive models. We demonstrate here that such models consistently perform better in some groups than others, suggesting that performance of published models could be improved through the development of group-specific models, and that prior models may reflect not unitary constructs of interest, but rather complex stereotypic profiles that must be acknowledged and explored. In sum, our work is important both because it reveals the impact of biased input data on model generalizability and interpretation, and because it offers concrete recommendations to address this issue; notably, simply increasing sample size will not be sufficient to overcome the problems that we identify.

Further, I do not follow the logic that it is not tautological that bias in neurocognitive tests would not propagate to prediction models. Would any other result than that biased behaviour measures probing individual differences result in biased models in the brain be possible/expected as the models are based on the biased data?

We thank the reviewer for this important point and apologize for not presenting it more clearly. A phenotypic outcome may be biased by a covariate, but this shared variance between outcome and covariate need not overlap with the variance shared between outcome and brain. Alternatively, it is possible that a given outcome-covariate relationship holds across the entirety of a sample (i.e., nobody defies the stereotype) or that a sample is homogeneous in the given domain (e.g., same age); in these cases, the covariate would not influence misclassification frequency (but may of course still be worthy of exploration). Finally, if the outcome-covariate shared variance does overlap with the outcome-brain shared variance, it need not do so in all participants; those in whom the consensus outcome-covariate relationship does not hold will be reliably misclassified, as we demonstrate.

We have added text to the discussion (lines 526-534) to clarify the various ways in which covariates related to the phenotype of interest may be reflected in the brain: “*Further, relationships between covariates and the outcome of interest may be complex, and differentially impact brain-phenotype relationships. A phenotypic outcome may be related to a covariate, but this shared variance need not overlap with the variance shared between outcome and brain. Alternatively, it is possible that a given outcome-covariate relationship holds across the entirety of the sample, or that a sample is homogenous in the given domain (e.g., age); in these cases, the covariate would not influence misclassification frequency. Finally, if the outcome-covariate shared variance does overlap with the outcome-brain shared variance¹, it need not do so in the same manner in all participants.*”

Second, I fail to see the key novelty of the message behind what brain-behaviour models are 'truly' predicting. From various studies evaluating brain-behaviour relationships, amongst other UKBiobank and ABCD, but also HCP, it has become clear that confounding factors (age, education, motion, race) but also SES, BMI, water intake, global signal, time of day of scan, may play key roles in driving brain-behaviour relationships. Indeed, such factors are already taken into account in neuroimaging papers investigating brain-behaviour associations. For example, usually specific age-ranges are investigated, effects of motion are evaluated, or (supplementary) analysis performed only in caucasian samples. Thus, I do see the break through insights gained in the current work.

As this reviewer points out, there is increasing evidence that brain-phenotype relationships can be confounded by sociodemographic covariates. As discussed above (response to Reviewer 1, comment 2), however, there are various ways in which a covariate may be related to brain and phenotype. Confound correction is indeed common, but it usually addresses only one of these scenarios, relying on the assumption that a single, often linear covariate-outcome relationship holds for all individuals in the sample². If it is instead the case that, within a sample, the relationship between a given covariate and outcome differs across groups, as we demonstrate here, then such a correction will fail, and may in some cases even induce a confounding relationship where in truth none exists³. In other words, the modeled covariate-outcome relationship reflects a stereotype; as we demonstrate here, these stereotypes are sample specific and often violated, and when they are, models—and corrections—that rely on them will fail. This is the primary message of our work: models not only pick up on and use biases in input data, but also fail in the substantial number of people who do not fit these stereotypes. More sophisticated correction approaches (e.g., the use of crossed-term confounds [i.e., interactions] and confound-based sample splitting³), paired with a deeper interrogation of these sample-specific stereotypes to inform model interpretation, will be crucial next steps. We discuss these recommended steps in greater detail in the revised abstract (lines 54-56) and discussion (lines 663-692).

Lines 54-56: *We present a framework to address these issues so that such models may reveal neural circuits underlying specific phenotypes...*

Lines 663-692: *Where, then, do we go from here? Given the demonstrated importance of sociodemographic and clinical covariates to brain-phenotype modeling analyses, future work should further characterize score profiles, looking to best-practice guidelines to collect more expansive and inclusive demographic data⁴, increase enrollment of underrepresented groups, and exchange proxies such as race for more meaningful causal or explanatory variables⁵⁻¹⁰. In the service of result generalizability and as proof of principle, we present the covariates that are related to model failure across all studied phenotypic measures, but such future work will permit the identification of more precise and phenotype-specific stereotypic profiles. In parallel, phenotypic measures must be carefully selected and administered to maximize their validity^{11,12}.*

These choices may be guided by recently developed tools to evaluate the risk of bias in study design (e.g., PROBAST, Step 3¹³).

Then, once data are collected, they must be used. That is, modeling analyses must be adapted to ask: what combination of factors does our outcome of interest measure, and how can we interpret related patterns of brain activity? First, statistical tools may be used to isolate, to the extent possible, the phenotype of interest. When standardizing phenotypic measures, norms should be carefully considered to ensure appropriateness (e.g., for the NIH Toolbox^{14,15}, but see^{16,17}). Further, data may be corrected for identified confounds. Many approaches to confound correction rely on the assumption that a single covariate-outcome relationship holds for all individuals in the sample. If this is not the case, as we demonstrate here, then such correction will fail, and may even induce a confounding relationship where in truth none exists³. To address this issue, more sophisticated correction approaches that account for sample-specific stereotypes will be necessary (e.g., use of crossed-term confounds, confound-based sample splitting³, inverse probability weighting¹⁸, or post-hoc confound control²). Inevitably, however, confounds will remain. It is incumbent on the modeler to use the previously collected, comprehensive sociodemographic data to precisely characterize these persistent confounds and interpret resulting models accordingly: as group-specific neural representations of composite phenotypes.

Relatedly, Reviewer 1 has pointed out that our work is similar to previously published work, particularly that investigating “brain age.” Indeed, the brain age literature inspired this work, and is methodologically similar. However, our work diverges from the brain age literature in motivation, implementation, and implications. Brain age analyses seek to develop a single, normative model, usually in a sample of “control” individuals, with deviations from this normative trajectory assumed to be informative and driven by biology¹⁹. For example, a participant who is predicted to be older than their chronological age is understood to have “advanced brain age,” and this has been related to a range of clinical symptoms, such as mild cognitive impairment and Alzheimer’s disease²⁰, as well as hypertension, diabetes, stroke, and lower performance on neurocognitive tests²¹. Conversely, we developed models in the sort of representative, heterogeneous samples that are commonly used for brain-phenotype modeling (e.g., ²²), and found that there is no single, normative model of a cognitive phenotype; different groups require different models, in part because of biases in input data. Thus, in the brain age paradigm, model errors reflect concerns about (e.g., the health of) the test participant; in our framework, model errors reflect concerns about the training data and the resulting model. We consider this difference so fundamental that, while we reference the brain age literature for methodological guidance in the manuscript, we do not use it to contextualize our results, as we fear that doing so may confuse readers. We do, however, place our work in the context of the bias in AI and neuropsychology literatures, and we highlight and have expanded this discussion in the revised manuscript (lines 73-101, 520-525, 567-601, 663-674).

Lines 73-101: The existence of structured model failure—some individuals who are better fit by a model than others—would suggest that one brain-phenotype relationship does not fit all, and that systematic bias may determine who is fit and who is not. This, in turn, may engender imprecise, misleading, and in some cases harmful model interpretations. That is, a brain network found to be associated with a given phenotype may only apply to a specific subset of the population at large, limiting its practical utility²³⁻²⁵, or may not represent the phenotype of interest. Indeed, factors that interfere with adequate phenotypic characterization have been documented for many widely used neurocognitive tests^{11,26}, and may include the fallacy of universalism (construct bias), the application of inappropriate norms, discordance between primary and assessment language, and the presence of instrument, administration (method), or interpretation bias^{11,27-30}. Related concerns about the ethical implications of data and model bias are receiving increasing attention in the machine learning literature³¹ (e.g., racial disparities³² or unrelated attribute sensitivity³³ in

facial recognition, and the reflection of biased input data in algorithmic predictions, from criminal justice^{34,35} to healthcare^{36,37}).

Lines 520-525: Other effects of sample representation on phenotypic score profiles are suggested by many intersecting literatures. Cultural influences on task strategies and test performance are well documented (e.g.,^{38,39}), and neuropsychological test performance differs by such factors as life course epidemiology, education quality, acculturation, and physical health⁴⁰⁻⁴². Many tests are thus composite measures²⁸, and it is these composites that our models are predicting.

Lines 567-601: Equally important and not mutually exclusive is the opportunity to use such cases, and the profiles they defy, to explore sources of bias encoded in input data. That is, if test scores are themselves biased, the models may be as well. Such model bias has been described in applications of machine learning algorithms ranging from criminal justice^{34,35} to healthcare^{36,37}. Care must be taken to interpret results accordingly. For example, African Americans and Hispanic/Latino Americans tend to score lower than non-Hispanic/Latino white Americans on neuropsychological tests (e.g.,⁴¹). These group differences are complex, often reductionist, and non-causal; efforts to explain them have focused on differences in factors such as education quality⁴³, acculturation⁴², neighborhood disadvantage⁴⁴, and research methodology^{11,12}. While consensus causal explanations remain an open question, the pervasiveness of such bias in commonly used phenotypic measures^{11,28} is a call to action to carefully consider what brain-based models are truly predicting. Indeed, race tracked neuropsychological test performance in all three studied samples. And despite the fact that our models had no access to information about race, race was related to MF in the Yale and HCP samples, such that high-scoring non-white participants were frequently misclassified as low-scoring, and vice versa for white participants. This finding is reminiscent of the errors made by the Correctional Offender Management Profiling for Alternative Sanctions (COMPAS, now “equivant”) system³⁵ and of recent evidence for “prediction shift” in African Americans²⁵.

We seek to avoid overinterpretation of these findings and note again that race is a non-causal, non-biological proxy for unmeasured variables that obscures much heterogeneity in these samples (see Limitations and future directions). What our results do reveal is unintended and easily missed bias in both model inputs (i.e., phenotypic measures limited by available assessment tools^{11,28}, such as those comprising the NIH Toolbox^{16,17}) and outputs (i.e., the profiles to which models correspond). This bias matters for two reasons: 1) It may yield the right predictions for the wrong reasons; researchers may interpret the model as the neural representation of a unitary phenotypic construct or may acknowledge the role of covariates but wrongly assume causality, and 2) It determines the limits of model generalizability, which in turn guide the models’ practical applications.

Lines 663-674: Where, then, do we go from here? Given the demonstrated importance of sociodemographic and clinical covariates to brain-phenotype modeling analyses, future work should further characterize score profiles, looking to best-practice guidelines to collect more expansive and inclusive demographic data⁴, increase enrollment of underrepresented groups, and exchange proxies such as race for more meaningful causal or explanatory variables⁵⁻¹⁰. In the service of result generalizability and as proof of principle, we present the covariates that are related to model failure across all studied phenotypic measures, but such future work will permit the identification of more precise and phenotype-specific stereotypic profiles. In parallel, phenotypic measures must be carefully selected and administered to maximize their validity^{11,12}. These choices may be guided by recently developed tools to evaluate the risk of bias in study design (e.g., PROBAST, Step 3¹³).

We also note a more minor difference between our work and much of the brain age literature: in keeping with recent work highlighting the pitfalls of brain age modeling—and in particular the relationship between prediction deviation (i.e., predicted – observed outcome) and outcome, itself^{45,46}—we use binary classification for main analyses and present residualized prediction deviation in Supplementary Table 11, avoiding practices that would inflate model performance⁴⁶.

Finally, to the reviewer's point about impact and novelty, we underscore the key message of our work: brain-based models of phenotype recapitulate sample stereotypes, and individuals to whom these stereotypes do not apply will be consistently misclassified or poorly predicted, necessitating more sophisticated correction approaches and group-specific model development. We have clarified this message throughout the abstract and discussion. We agree that there is a growing literature on bias in neuropsychological testing and AI, and highlight this past work in our introduction (lines 93-101; see above). We also appreciate the reviewer bringing to our attention Falk et al., 2013, PNAS. This perspective piece concisely lays out the need for a science of individual differences that appreciates potential group-specific brain-phenotype relationships. The authors present several examples of brain-behavior relationships that are moderated by "experience, context, and culture"⁴⁷, and underscore the importance of relationship generalizability to populations of interest, both of which are key focuses of our work. In other words, this perspective is a call to action, but one that has not yet altered common practices in the field. Our work is the first to our knowledge to empirically demonstrate the need for this paradigm shift across all brain-phenotype modeling work, and outlines guidelines to achieve it (lines 663-692; see above). In sum, we agree with this reviewer that the bias encoded in phenotypic measures is a critical problem, and one that we find to be inextricably intertwined with brain-phenotype model performance and utility (lines 464-473). Developing methods to adequately address this problem will be necessary if brain-phenotype modeling is to yield useful insights in all individuals, not just those who fit sample stereotypes. We again direct the reviewer to lines 663-692 of the revised manuscript, where we discuss recommendations to guide this future work.

Lines 464-473: *Model failure is thus inextricably intertwined with biases in input data, and these issues must be jointly addressed if brain-phenotype models are to yield useful neuroscientific and clinical insights.*

Minor comments;

1. Though I appreciate the inclusion of the HCP dataset, I believe that this sample is biased in the sense that it would include twin-pairs, therefore showing stronger inter-subject similarity than the other datasets. Moreover, the individual datasets individually still are about half the size of the recommended sample size (N=2000) for brain-behavior predictions.

We thank the reviewer for these important points. To account for potential family-related biases, we assigned family members to the same fold and used multi-level block permutation, as implemented in PALM^{48,49}. We have added a description of this analytic approach to Results (lines 155-156) and describe it in Methods (lines 981-983) and in the caption of Supplementary Fig. 3.

Lines 155-156: *...(with family members assigned to the same fold, and permutations respecting family-related limits on exchangeability^{48,49}; Supplementary Fig. 3a; Supplementary Table 6).*

Lines 981-983: *Permutations were fixed across in-scanner conditions, and respected family-related limits on exchangeability for the HCP dataset^{48,49}.*

In particular, in the context of our work, we had two concerns about family-related biases, both addressed by our approach:

1. Inflated model performance due to inclusion of an individual's sibling (and, in particular, twin) in the training set: We avoid this issue by ensuring that family members are always assigned to the same fold. Further, by only shuffling data within families and across families of comparable structure, we maintain any family-related structure in phenotypic scores when we run permutation tests. With this structure intact, we still see a drop in model performance (mean accuracy for cIQ across tasks and iterations = 0.509, mean accuracy for fIQ across tasks and iterations = 0.502), with unpermuted model performance significantly better than permuted model performance.
2. Relationship between having a twin/sibling in the sample and misclassification frequency: Again, respecting family-related limits on exchangeability addresses this concern. Holding family structure constant, we still see the characteristic distribution shape of permuted misclassification frequency, with a significant difference between permuted and unpermuted misclassification frequency distributions ($P < 0.0001$, FDR adjusted, via two-tailed, two-sample Kolmogorov-Smirnov test; Supplementary Fig. 3b, revised legend). To further address this concern, we compared misclassification frequencies for individuals with siblings in the sample and those without, finding no significant difference via Wilcoxon rank sum test (cIQ $P = 0.77$, fIQ $P = 0.36$). This result further demonstrates that related individuals do not have higher or lower misclassification frequencies than singletons. This result is now reported in the manuscript on lines 230-232: "*In the HCP sample, MF did not significantly differ for individuals with and without siblings in the sample (cIQ $P = 0.77$, fIQ $P = 0.36$, uncorrected, via Mann-Whitney U test).*"

To the reviewer's point about sample size, we have closely followed the work of Marek et al. and share the concerns of those authors about non-generalizable brain-phenotype relationships⁵⁰. That said, the primary focus of Marek et al. is on within-sample brain-phenotype associations, not on prediction, which by definition requires such associations to generalize to unseen data (and which is the approach we take here). Indeed, the importance of such validation is a key theme of Marek et al.; they use split-half analyses as a benchmark to assess effect size inflation in the training set in both univariate (Supplementary Fig. 3, Extended Data Fig. 8) and multivariate (Fig. 4, Supplementary Fig. 16) analyses, finding that a large training set is necessary to meaningfully study *within-sample* brain-phenotype associations using their phenotypes and brain measures of choice.

Importantly, our work is in complete agreement with that of Marek et al. in that we only analyze brain-phenotype relationships that generalize to unseen data. In fact, our work offers a more stringent test of generalizability than cross-validation by demonstrating that results (model performance and misclassification) hold across independent datasets—datasets that vary in terms of study design, as well as participant demographic and clinical characteristics.

In the revised manuscript, we have clarified the distinction between prediction and association, as well as the importance of cross-dataset validation (lines 39-40, 128-129, and 138-139). We have also summarized this discussion in Methods (lines 721-725). For these reasons, we feel that the results of Marek et al. are not directly relevant to our work, and are confident that our chosen methods strongly support result generalizability.

Lines 39-40: *...we related brain activity to phenotype via predictive models—trained and tested on independent data to ensure generalizability—and explored model failure.*

Lines 128-129: *For cross-dataset and replication analyses, designed to ensure the generalizability of our results...*

Lines 138-139: *...which crucially tests models on previously unseen data (unlike explanatory models^{51,52}*

Lines 721-725: *while it is still not as large as samples called for by recent work⁵⁰, we note that the concerns of Marek and colleagues are not directly relevant to our work, as the former focuses on within-sample brain-phenotype associations, whereas our analyses rely exclusively on prediction, presenting brain-phenotype relationships that generalize to unseen data*

2. In the revision the authors have added one alternative classification algorithm, an ensemble of weak learners with hyperparameters selected via cross-validated optimisation, however, I am not sure if only two different models are sufficient to generalise over the wealth of models possible.

To address this concern, we have added a neural network-based analysis to the manuscript, described in Methods (lines 948-966) with results presented in Supplementary Fig. 1 (reproduced below). With this addition, we now present four, commonly used algorithmic approaches to supervised learning (classification: SVM, random forest, neural network; prediction: linear regression) that reflect the ranges of model interpretability and complexity (see, for example, Fig. 1 of ⁵³), as described on lines 948-951. In addition, we demonstrate result generalizability across two different brain parcellations, two different cross-validation approaches, and alternative phenotype binarization thresholds. At each of these decision points, we have sought methods that are common and representative of their class. Across all tested algorithms, both parcellations, both cross-validation approaches, and both binarization thresholds, model performance and misclassification frequency are essentially unchanged, evidencing the robustness of our results.

Lines 948-966: *To ensure that results are robust to classification algorithm, we repeated these analyses using two additional, commonly used algorithms for supervised learning that, together with linear SVM and linear regression (see below), reflect the full range of model interpretability and complexity⁵³: an ensemble of weak learners and a fully-connected neural network, both implemented in Matlab (Mathworks). In both cases, we used a subset of phenotypic measures for hyperparameter optimization (ensemble learners: ensemble aggregation method, number of learners, learning rate [where relevant], and minimum leaf size; neural network: activation functions, standardization, regularization term strength, and layer sizes). We used all available high and low scorers' GFC and phenotypic data, undersampled to balance class size, in a leave-one-out fashion for each optimization analysis, and used the best consensus hyperparameters (ensemble learners: method = bagging, number of learners = 150, minimum leaf size = 1; neural network: activation function = none; standardization = true; lambda = 1.34E-5; layer sizes = 8, 200, and 8) to classify all 16 phenotypic measures as in main analyses, using all selected edges (correlation $P < 0.05$, as in main analyses) as features. See Supplementary Fig. 1 for results.*

Supplementary Figure 1. Model performance and misclassification frequency are robust to analysis approach. (a) Classification accuracy for each phenotypic measure using FC calculated from all in-scanner conditions in the Yale dataset, and five different analysis pipelines: an alternative, 368-node parcellation for FC matrix generation, two alternative classification algorithms (ensemble of weak learners and neural network), an alternative phenotypic binarization threshold (mean split), and an alternative (10-fold) cross-validation approach (see Methods for additional description of each analysis). Boxplot line and hinges represent median and quartiles, respectively; whiskers extend to most extreme non-outliers; outliers plotted individually (+). r1, rest 1; r2, rest 2; grad, gradual onset continuous performance task; sst, stop signal task; gfc, general functional connectivity. (b) MF, averaged across in-scanner conditions and phenotypic measures to derive a single value per participant, compared between each alternative analysis and main-text analyses. r_s , rank correlation, $n = 128-129$. Note that phenotype mean split is equivalent to mean $\pm 1/3$ *s.d. for scaled scores; mean split-based model accuracy is not reported for these measures, nor are they included in MF calculation. Given the limited mean split-based results, we repeated this analysis in the HCP data, with comparable results (mean MF $r_s = 0.86$, $P < 0.0001$). 10-fold results reflect 1000 analysis

iterations per phenotypic measure and in-scanner condition (50 per cross-validation partition); all other analyses reflect 100 iterations. In this and all subsequent figures: BNT, Boston Naming Test; WRAT, Wide Range Achievement Test; VL, verbal learning; FW, finger windows; LN, letter-number sequencing; Trails, trail making; VF, verbal fluency; CW, color-word interference; 20Q, 20 questions; Vocab, vocabulary; MR, matrix reasoning.

3. Taking for example figure 4, it is clear there is a strong relationship between not only education but also motion and prediction score. Does this not point to an overall bias of motion in the fMRI data? Or what would the mechanism here be?

As this reviewer points out, there is a well-documented relationship between motion- and other health-related variables^{3,54}, and motion and functional connectivity (e.g.,⁵⁵). We find this to be an example of the case in which the shared variance between brain and phenotype, and between phenotype and motion, overlaps, such that the models are in part using motion to generate their predictions. This of course requires that model interpretation incorporate motion, and/or that modelers correct brain and phenotypic data for head motion. Correction is complicated, however, by one of the key findings of the paper, described in greater detail above: the relationship between motion and phenotype differs across groups. This relationship is therefore crucially not an “overall” bias. While motion generally tracks worse performance on phenotypic measures in all three datasets, it is either unrelated (Yale) or positively related (UCLA, HCP) to cognitive performance in individuals who are frequently misclassified. This is yet another example of a stereotype that does not hold in all individuals; models that use this stereotype will fail in those individuals who defy it.

4. It seems the in-text changes are not tracked, nor fully specified in the letter the reviewers, making it challenging for me to evaluate alterations and edits in the current revision. We apologize for this omission. All changes are tracked in this submission. In our previous submission, we substantially revised the introduction and discussion to clarify key points of the manuscript, as suggested in reviewers’ comments, and chose not to track these changes in the final PDF for the sake of readability. However, all major changes (added analyses and results) are referenced with line numbers in our point-by-point response.

Referee #2 (Remarks to the Author):

The authors have been very responsive to the previous reviewers' comments and provided comprehensive replies. Several additional considerations would still be worthwhile to incorporate into this paper, however:

1) The goal of the work was to clarify in whom predictive brain-behavior models fail and why. The revision now shows such model failure in 3 independent datasets and across two machine learning architectures. I think the paper therefore now gives a broader support of the initial message.

However, I partially also agree with some of the other reviewers' previous comments on whether this is sufficient support for a generalizable conclusion, or whether these describe still algorithm-specific limitations. The authors would therefore be encouraged to either present additional support across a few additional architectures, or at a minimum to further emphasize this as a limitation of the current work in the paper's discussion.

We thank the reviewer for bringing up this important point. As described in our response to Reviewer 1's minor comment 2, we have added a fourth algorithm to the manuscript: a fully-connected neural network. Methods and motivation for using each algorithm are described on lines 948-966 (see above) and results are summarized in Supplementary Fig. 1 (reproduced

above) and Results (lines 232-235: “MF and overall classification performance were comparable with additional motion controls (Supplementary Fig. 4; Supplementary Table 7), as well as with different supervised learning algorithms, brain parcellation, cross-validation approach, and phenotype binarization threshold (Supplementary Fig. 1)”). In brief, we used three phenotypic measures to optimize hyperparameters (activation function, layer number and sizes, standardization, and regularization term strength), as for the ensemble learner analysis. We then used the single best combination of hyperparameters for classification of each phenotypic measure. Results were comparable to results of the SVM and ensemble learner analyses, and are presented in the same fashion (classification accuracy and misclassification frequency) in Supplementary Fig. 1. As discussed above and on lines 948-951 (see above), the four algorithms we present—linear SVM, ensemble of weak learners, and neural network for classification, and linear regression for continuous prediction—represent the spectrum of model interpretability and complexity, as described for example by Bzdok and Ioannidis⁵³ and others⁵⁶⁻⁵⁸. Our results hold across these very different approaches, demonstrating their robustness to algorithm and the generalizability of the key findings of this paper: model failure is structured and reliable, and reflects biases that must be acknowledged.

2) The authors outline associations between misclassification and several socio-demographic covariates, incl age, education, etc. Wouldn't in many cases statistical correction and homogenization procedures pre-analysis, often done in the field, mitigate those effects. Further exploration of invariance of their findings to presence/absence of statistical control procedures, and/or additional discussion on this issue could be helpful.

Please see our response to Reviewer 1's second and third comments, above. In short, these covariates are related to misclassification because their relationship with phenotype differs across groups. Further complicating standard correction approaches, these group-specific relationships will in many cases also be sample specific. This points to two key future directions, as described on lines 675-692. First, as this reviewer points out, the development of more sophisticated, *sample-specific* correction approaches will identify biases and remove them from analyses when possible. Second, biases that cannot be eliminated must be acknowledged in model interpretations.

Lines 675-692: *Then, once data are collected, they must be used. That is, modeling analyses must be adapted to ask: what combination of factors does our outcome of interest measure, and how can we interpret related patterns of brain activity? First, statistical tools may be used to isolate, to the extent possible, the phenotype of interest. When standardizing phenotypic measures, norms should be carefully considered to ensure appropriateness (e.g., for the NIH Toolbox^{14,15}, but see^{16,17}). Further, data may be corrected for identified confounds. Many approaches to confound correction rely on the assumption that a single covariate-outcome relationship holds for all individuals in the sample. If this is not the case, as we demonstrate here, then such correction will fail, and may even induce a confounding relationship where in truth none exists³. To address this issue, more sophisticated correction approaches that account for sample-specific stereotypes will be necessary (e.g., use of crossed-term confounds, confound-based sample splitting³, inverse probability weighting¹⁸, or post-hoc confound control²). Inevitably, however, confounds will remain. It is incumbent on the modeler to use the previously collected, comprehensive sociodemographic data to precisely characterize these persistent confounds and interpret resulting models accordingly: as group-specific neural representations of composite phenotypes.*

3) In addition to above socio-demographic associations to MF, one issue that was somewhat underspecified was the question of whether there might be specific brain features associated to MF. The authors could present, in supplementary form, eg FC matrices of the CCP and MCP

groups and their difference, and to map areas and connections in the brain that contribute to model failure. Alternatively, it may be possible to correlate (for example corresponding to Fig 2) MF frequency with FC measures. As MF misclassification appears to be consistent across conditions, one may further want to know whether FC features associated with MF are also consistent across conditions. I think such an analysis would add potentially meaningful neuroscientific insights to the paper, and should be relatively easy to incorporate. We thank the reviewer for these interesting suggestions. We have performed several additional analyses to explore any differences in the functional organization of CCP and MCP individuals' brains. First, we identified CCP and MCP groups for each phenotypic measure and in-scanner task using the median accuracy iteration (as for the cross-dataset analyses). Then, we performed a two-sample t -test at every edge, FDR adjusted resulting P values, and plotted all edges found to significantly differ between the two groups, with nodes ordered by canonical brain network. Next, we sought brain networks that significantly differ between groups, using the constrained network-based statistic approach⁵⁹, again with FDR adjustment of resulting P values, and again plotted these networks. Results for GFC are plotted in Supplementary Fig. 9 (reproduced below), and demonstrate that there are very few edges and network pairs that significantly differ across groups. To further assess the consistency of these group differences across in-scanner tasks, as this reviewer suggested, we counted the number of times (i.e., tasks for which) an edge or network pair was significantly greater for CCP than MCP, and subtracted the number of times the edge or network pair was significantly greater for MCP than CCP. Resulting matrices are plotted in Supplementary Fig. 9, and demonstrate that there is very little consistency in group differences across in-scanner tasks. Finally, we present GFC for CCP and MCP, averaged across participants, with nodes ordered by network (Supplementary Fig. 9), again highlighting the very similar patterns of functional brain organization between groups. Overall, these results highlight that CCP and MCP groups do not have consistently different brain organization. Rather, it is the relationship between brain and phenotype that differs across groups. We have added a discussion of these analyses and findings to lines 418-421 and 1061-1064, and highlight this distinction between brain differences and brain-phenotype relationship differences on lines 608-610.

Lines 418-421: *Finally, we explored two additional questions raised by these findings. First, we compared FC patterns of CCP and MCP groups to identify any group differences in functional brain organization that may explain misclassification. We found no consistent differences between groups at either the edge or network levels (Supplementary Fig. 9).*

Lines 1061-1064: *The above approach for identifying CCP and MCP groups was also used to explore between-group differences in FC in the Yale sample at the edge (via two-sample t -test) and network (via the constrained network-based statistic⁵⁹) levels (Supplementary Fig. 9).*

Lines 608-610: *Intriguingly, we demonstrate that misclassified individuals—those who defy the consensus score profile—do not have distinct brain organization; rather, it is the relationship between brain and phenotype that differs between CCP and MCP.*

Supplementary Figure 9. Comparison of FC between CCP and MCP groups at edge and network levels. Edges, GFC: t statistics for each GFC edge found to significantly differ (via two-sample t -test) between groups ($P < 0.05$, FDR adjusted), ordered by network. Red, CCP>MCP; Blue, MCP>CCP. Networks, GFC: mean t statistics for each network pair (using GFC) found to significantly differ (via Constrained NBS⁵⁹) between groups (one-tailed $P < 0.025$, FDR adjusted). Red, CCP>MCP; Blue, MCP>CCP. Significant edges across tasks and Significant networks across tasks: Number of times (i.e., tasks for which) edge (ordered by network) or network pair was significantly greater for CCP than MCP – number of times edge or network pair was significantly greater for MCP than CCP. Mean GFC, CCP and Mean GFC, MCP: GFC, averaged across participants within each group; main diagonal set to 0, and nodes ordered by network. Note that CCP and MCP groups differ for each phenotypic measure and in-scanner task (range of number of participants for GFC across phenotypic measures: CCP = 46-81, MCP = 23-63). Black dashed lines separate networks; networks: 1 = medial frontal, 2 = frontoparietal, 3 = default mode, 4 = motor, 5 = visual A, 6 = visual B, 7 = visual association, 8 = salience, 9 = subcortical, 10 = cerebellum (for network visualization, see ⁶⁰).

Minor:

For the HCP dataset, the authors mention that HCP contains family associations so I assume they removed twins but further specification may be worthwhile.

Please see response to Reviewer 1's minor comment 1. In brief, we ensured that family members were always assigned to the same fold and accounted for family-related limits on exchangeability using multi-level block permutation, an approach that is well suited to datasets with family members and extensively validated, including in this dataset^{48,49}. We have added a description of this analytic approach to Results (lines 155-156) and describe it in Methods (lines 981-983) and Supplementary Fig. 3.

Lines 155-156: *...(with family members assigned to the same fold, and permutations respecting family-related limits on exchangeability^{48,49}; Supplementary Fig. 3a; Supplementary Table 6).*

Lines 981-983: *Permutations were fixed across in-scanner conditions, and respected family-related limits on exchangeability for the HCP dataset^{48,49}.*

Referee #3 (Remarks to the Author):

The authors are to be commended for a thorough and energetic response to the reviews.

This reviewer's primary concern was about the limited interpretability of the results due to a reliance on a single predictive framework. However, the authors have addressed this in a number of ways, using a completely different family of predictive methods, considering 10-fold instead of LOO cross validation, using a continuous instead of binarised outcome and altering the parcellation scheme used, while finding similar results.

As a result I have no further concerns about the publication of this work.

We thank the reviewer for these supportive comments.

References

1. Dadi, K. *et al.* Population modeling with machine learning can enhance measures of mental health. *bioRxiv* 2020.08.25.266536 (2021). doi:10.1101/2020.08.25.266536
2. Dinga, R., Schmaal, L., Penninx, B. W. J. H., Veltman, D. J. & Marquand, A. F. Controlling for effects of confounding variables on machine learning predictions. *bioRxiv* 2020.08.17.255034 (2020). doi:10.1101/2020.08.17.255034

3. Alfaro-Almagro, F. *et al.* Confound modelling in UK Biobank brain imaging. *Neuroimage* **224**, 117002 (2021).
4. Hughes, J. L., Camden, A. A., Yangchen, T. & Colledge, A. S. Rethinking and updating demographic questions: Guidance to improve descriptions of research samples. *Psi Chi J. Psychol. Res.* **21**, 138–151 (2016).
5. Williams, D. R. The concept of race in Health Services Research: 1966 to 1990. *Health Serv. Res.* **29**, 267–274 (1994).
6. Kaplan, J. B. & Bennett, T. Use of race and ethnicity in biomedical publication. *J. Am. Med. Assoc.* **289**, 2709–2716 (2003).
7. Fullilove, M. T. Comment: Abandoning ‘race’ as a variable in public health research—an idea whose time has come. *Am. J. Public Health* **88**, 1297–1298 (1998).
8. Wang, L.-I. Race as proxy: Situational racism and self-fulfilling stereotypes. *DePaul Law Rev.* **53**, 1013–1110 (2004).
9. Corbie-Smith, G., Henderson, G., Blumenthal, C., Dorrance, J. & Estroff, S. Conceptualizing race in research. *J. Natl. Med. Assoc.* **100**, 1235–1243 (2008).
10. Ioannidis, J. P. A., Powe, N. R. & Yancy, C. Recalibrating the use of race in medical research. *JAMA* **325**, 623–624 (2021).
11. Fernández, A. L. & Abe, J. Bias in cross-cultural neuropsychological testing: problems and possible solutions. *Cult. Brain* **6**, 1–35 (2018).
12. Rivera Mindt, M., Byrd, D., Saez, P. & Manly, J. Increasing culturally competent neuropsychological services for ethnic minority populations: a call to action. *Clin. Neuropsychol.* **24**, 429–453 (2010).
13. Wolff, R. F. *et al.* PROBAST: A tool to assess the risk of bias and applicability of prediction model studies. *Ann. Intern. Med.* **170**, 51–58 (2019).
14. Casaletto, K. B. *et al.* Demographically corrected normative standards for the English version of the NIH Toolbox Cognition Battery. *J Int Neuropsychol Soc* **21**, 378–391 (2015).
15. Nitsch, K. P. *et al.* Uncorrected versus demographically-corrected scores on the NIH Toolbox Cognition Battery in persons with traumatic brain injury and stroke. *Rehabil. Psychol.* **62**, 485–495 (2017).
16. Karr, J. E., Mindt, M. R. & Iverson, G. L. A multivariate interpretation of the Spanish-language NIH Toolbox Cognition Battery: The normal frequency of low scores. *Arch. Clin. Neuropsychol.* **00**, 1–14 (2021).
17. MacAulay, R. K., Boeve, A. & Halpin, A. Comparing psychometric properties of the NIH Toolbox Cognition Battery to gold-standard measures in socioeconomically diverse older adults. *Arch. Clin. Neuropsychol.* **36**, 1523–1534 (2021).
18. Linn, K. A., Gaonkar, B., Doshi, J., Davatzikos, C. & Shinohara, R. T. Addressing confounding in predictive models with an application to neuroimaging. *Int. J. Biostat.* **12**, 31–44 (2016).
19. Baecker, L., Garcia-Dias, R., Vieira, S., Scarpazza, C. & Mechelli, A. Machine learning for brain age prediction: Introduction to methods and clinical applications. *EBioMedicine* **72**, (2021).
20. Varikuti, D. P. *et al.* Evaluation of non-negative matrix factorization of grey matter in age prediction. *Neuroimage* **173**, 394 (2018).
21. Cole, J. H. Multimodality neuroimaging brain-age in UK biobank: relationship to biomedical, lifestyle, and cognitive factors. *Neurobiol. Aging* **92**, 34–42 (2020).
22. Van Essen, D. C. *et al.* The Human Connectome Project: A data acquisition perspective. *Neuroimage* **62**, 2222 (2012).
23. Lanka, P. *et al.* Supervised machine learning for diagnostic classification from large-scale neuroimaging datasets. *Brain Imaging Behav.* **14**, 2378–2416 (2020).
24. Benkarim, O. *et al.* The cost of untracked diversity in brain-imaging prediction. *bioRxiv*

- (2021). doi:10.1101/2021.06.16.448764
25. Li, J. *et al.* Cross-ethnicity/race generalization failure of behavioral prediction from resting-state functional connectivity. *Sci. Adv.* **8**, 1812 (2022).
 26. Statucka, M. & Cohn, M. Origins matter: Culture impacts cognitive testing in Parkinson's disease. *Front. Hum. Neurosci.* **13**, 269 (2019).
 27. Manly, J. J. Critical issues in cultural neuropsychology: profit from diversity. *Neuropsychol. Rev.* **18**, 179 (2008).
 28. Casaletto, K. B. & Heaton, R. K. Neuropsychological assessment: past and future. *J. Int. Neuropsychol. Soc.* **23**, 778–790 (2017).
 29. Whaley, A. L. Stereotype threat and neuropsychological test performance in the U.S. African American population. *Arch. Clin. Neuropsychol.* **36**, 1361–1366 (2021).
 30. Thames, A. D. *et al.* Effects of stereotype threat, perceived discrimination, and examiner race on neuropsychological performance: Simple as black and white? *J. Int. Neuropsychol. Soc.* **19**, 583–593 (2013).
 31. Chouldechova, A. & Roth, A. A snapshot of the frontiers of fairness in machine learning. *Commun. ACM* **63**, 82–89 (2020).
 32. Klare, B. F., Burge, M. J., Klontz, J. C., Vorder Bruegge, R. W. & Jain, A. K. Face recognition performance: Role of demographic information. *IEEE Trans. Inf. Forensics Secur.* **7**, 1789–1801 (2012).
 33. Denton, E., Hutchinson, B., Mitchell, M., Gebru, T. & Zaldivar, A. Image counterfactual sensitivity analysis for detecting unintended bias. *arXiv* (2020).
 34. Dressel, J. & Farid, H. The accuracy, fairness, and limits of predicting recidivism. *Sci. Adv.* **4**, eaao5580 (2018).
 35. Machine Bias — ProPublica. doi:<https://www.propublica.org/article/machine-bias-risk-assessments-in-criminal-sentencing>
 36. Roberts, M. *et al.* Common pitfalls and recommendations for using machine learning to detect and prognosticate for COVID-19 using chest radiographs and CT scans. *Nat. Mach. Intell.* **3**, 199–217 (2021).
 37. Obermeyer, Z., Powers, B., Vogeli, C. & Mullainathan, S. Dissecting racial bias in an algorithm used to manage the health of populations. *Science* **366**, 447–453 (2019).
 38. Hedden, T., Ketay, S., Aron, A., Markus, H. R. & Gabrieli, J. D. E. Cultural influences on neural substrates of attentional control. *Psychol. Sci.* **19**, 12–17 (2008).
 39. Pérez-Arce, P. The influence of culture on cognition. *Arch. Clin. Neuropsychol.* **14**, 581–592 (1999).
 40. Werry, A. E., Daniel, M. & Bergström, B. Group differences in normal neuropsychological test performance for older non-Hispanic White and Black/African American adults. *Neuropsychology* **33**, 1089–1100 (2019).
 41. Gasquoine, P. G. Race-norming of neuropsychological tests. *Neuropsychol Rev* **19**, 250–262 (2009).
 42. Manly, J. J. *et al.* The effect of African-American acculturation on neuropsychological test performance in normal and HIV-positive individuals. The HIV Neurobehavioral Research Center (HNRC) Group. *J. Int. Neuropsychol. Soc.* **4**, 291–302 (1998).
 43. Manly, J. J., Jacobs, D. M., Touradji, P., Small, S. A. & Stern, Y. Reading level attenuates differences in neuropsychological test performance between African American and White elders. *J. Int. Neuropsychol. Soc.* **8**, 341–348 (2002).
 44. Vinopal, K. & Morrissey, T. W. Neighborhood disadvantage and children's cognitive skill trajectories. *Child. Youth Serv. Rev.* 105231 (2020).
 45. Smith, S. M., Vidaurre, D., Alfaro-Almagro, F., Nichols, T. E. & Miller, K. L. Estimation of brain age delta from brain imaging. *Neuroimage* **200**, 528–539 (2019).
 46. Butler, E. R. *et al.* Pitfalls in brain age analyses. *Hum. Brain Mapp.* **42**, 4092–4101 (2021).

47. Falk, E. B. *et al.* What is a representative brain? Neuroscience meets population science. *Proc. Natl. Acad. Sci. U. S. A.* **110**, 17615 (2013).
48. Winkler, A. M., Ridgway, G. R., Webster, M. A., Smith, S. M. & Nichols, T. E. Permutation inference for the general linear model. *Neuroimage* **92**, 381–397 (2014).
49. Winkler, A. M., Webster, M. A., Vidaurre, D., Nichols, T. E. & Smith, S. M. Multi-level block permutation. *Neuroimage* **123**, 253–268 (2015).
50. Marek, S. *et al.* Reproducible brain-wide association studies require thousands of individuals. *Nat.* **11**, 1–7 (2022).
51. Yarkoni, T. & Westfall, J. Choosing prediction over explanation in psychology: lessons from machine learning. *Perspect. Psychol. Sci.* **12**, 1100–1122 (2017).
52. Scheinost, D. *et al.* Ten simple rules for predictive modeling of individual differences in neuroimaging. *Neuroimage* **193**, 35–45 (2019).
53. Bzdok, D. & Ioannidis, J. P. A. Exploration, inference, and prediction in neuroscience and biomedicine. *Trends in Neurosciences* **42**, 251–262 (2019).
54. Bijsterbosch, J. D. *et al.* The relationship between spatial configuration and functional connectivity of brain regions. *Elife* **7**, DOI: 10.7554/eLife.32992 (2018).
55. Power, J. D., Barnes, K. A., Snyder, A. Z., Schlaggar, B. L. & Petersen, S. E. Spurious but systematic correlations in functional connectivity MRI networks arise from subject motion. *Neuroimage* **59**, 2142–2154 (2012).
56. Smith, S. M. & Nichols, T. E. Statistical challenges in “big data” human neuroimaging. *Neuron* **97**, 263–268 (2018).
57. Schulz, M.-A. *et al.* Different scaling of linear models and deep learning in UKBiobank brain images versus machine-learning datasets. *Nat. Commun.* **11**, 1–15 (2020).
58. Bzdok, D., Nichols, T. E. & Smith, S. M. Towards algorithmic analytics for large-scale datasets. *Nat. Mach. Intell.* **1**, 296–306 (2019).
59. Noble, S. & Scheinost, D. The Constrained Network-Based Statistic: A new level of inference for neuroimaging. *Lect. Notes Comput. Sci. (including Subser. Lect. Notes Artif. Intell. Lect. Notes Bioinformatics)* **12267 LNCS**, 458–468 (2020).
60. Noble, S. *et al.* Influences on the test–retest reliability of functional connectivity MRI and its relationship with behavioral utility. *Cereb. Cortex* **27**, 5415–5429 (2017).

Reviewer Reports on the Second Revision:

Referees' comments:

Referee #1 (Remarks to the Author):

I thank the authors for the hard work and taking the time to respond to my concerns,

best wishes.

Referee #2 (Remarks to the Author):

I thank the authors for their careful attention to mine and the other Reviewer's comments and support publication of the current work.